# IMPACT GNN: IMPOSING INVARIANCE WITH MESSAGE PASSING IN CHRONOLOGICAL SPLIT TEMPORAL GRAPHS

## ABSTRACT

This paper addresses domain adaptation challenges in graph data resulting from chronological splits. In a transductive graph learning setting, where each node is associated with a timestamp, we focus on the task of Semi-Supervised Node Classification (SSNC), aiming to classify recent nodes using labels of past nodes. Temporal dependencies in node connections create domain shifts, causing significant performance degradation when applying models trained on historical data into recent data. Given the practical relevance of this scenario, addressing domain adaptation in chronological split data is crucial, yet underexplored. We propose Imposing invariance with Message Passing in Chronological split Temporal Graphs (IMPaCT), a method that imposes invariant properties based on realistic assumptions derived from temporal graph structures. Unlike traditional domain adaptation approaches which rely on unverifiable assumptions, IMPaCT explicitly accounts for the characteristics of chronological splits. The IMPaCT is further supported by rigorous mathematical analysis, including a derivation of an upper bound of the generalization error. Experimentally, IMPaCT achieves a 3.8% performance improvement over current SOTA method on the ogbn-mag graph dataset. Additionally, we introduce the Temporal Stochastic Block Model (TSBM), which replicates temporal graphs under varying conditions, demonstrating the applicability of our methods to general spatial GNNs.

## 1 INTRODUCTION

The task of Semi-supervised Node Classification (SSNC) on graph often involves nodes with temporal information. For instance, in academic citation network, each paper node may contain information regarding the year of its publication. The focus of this study lies within such graph data, particularly on datasets where the train and test splits are arranged in chronological order. In other words, the separation between nodes available for training and those targeted for inference occurs temporally, requiring the classification of the labels of nodes with the most recent timestamp based on the labels of nodes with historical timestamp. While leveraging GNNs trained on historical data to classify newly added nodes is a common scenario in industrial and research settings (Liu et al., 2016; Bai et al., 2020; Pareja et al., 2020), systematic research on effectively utilizing temporal information within chronological split graphs remains scarce.

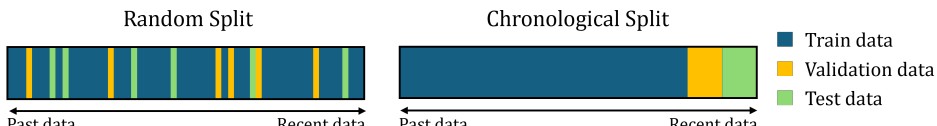

Figure 1: Illustrative explanation of chronological split dataset.

Failure to appropriately utilize temporal information can lead to significant performance degradation when the model attempts to classify labels of recent data. We conducted a toy experiment on the ogbn-mag dataset, an academic graph dataset having features with chronological split, to confirm the existence of such distribution shifts. The specific settings of this experiment can be found in Appendix A.2. Table 1 presents results of the toy experiment.

Table 1: Accuracy of SeHGNN (Yang et al., 2023) on ogbn-mag for chronological and random split.

| Split | wo/time emb | w/time emb |
|---|---|---|
| Chronological | $0.5682 \pm 0.001$ | $0.5580 \pm 0.0009$ |
| Random | $0.6302 \pm 0.0011$ | $0.6387 \pm 0.0011$ |

The substantial difference in accuracy, 5.2%, between the chronological split and random split settings clearly demonstrates the presence of distribution shift induced by the chronological split. Time positional encoding contributes to obtain better test accuracy only in random split setting. This discrepancy arises because in the chronological split setting, the inference process of test nodes encounters time positional encodings not seen during training. The distribution for recent nodes may exhibit extrapolation properties that diverge from those of past nodes, thereby adding to the challenging nature of this problem. In our work, we presented robust and realistic assumptions on temporal graph, and proposed message passing methods, IMPaCT, to impose invariant representation.

**Contributions** Our research contributes in the following ways:
(a) We present robust and realistic assumptions that rooted in properties observable in real-world graphs, to effectively analyze and address the domain adaptation problem in graph datasets with chronological splits. (b) We propose the scalable imposing invariance with message passing methods, IMPaCT, and established a theoretical upper bound of the generalization error when our methods were used. (c) We propose Temporal Stochastic Block Model (TSBM) to generate realistic temporal graph, and systematically demonstrate the robustness and applicability to general spatial GNNs of IMPaCT. (d) We showcase significant performance improvements on the real-world citation graph ogbn-mag, yielding significant margin of 3.8% over current SOTA method.

## 2 RELATED WORK

### 2.1 GRAPH NEURAL NETWORKS

Graph Neural Networks (GNNs) have gained significant attention across various domains, including recommender systems (Ying et al., 2018; Gao et al., 2022), biology (Barabasi & Oltvai, 2004), and chemistry (Wu et al., 2018). Spatial GNNs, such as GCN (Kipf & Welling, 2017), GraphSAGE (Hamilton et al., 2017), GAT (Velickovic et al., 2017) and HGNN (Feng et al., 2019), derive topological information by aggregating information from neighboring nodes through message passing.

$$M_v^{(k+1)} \leftarrow \text{AGG}(\{X_u^{(k)}, \forall u \in \mathcal{N}_v\}) \tag{1}$$

$$X_v^{(k+1)} \leftarrow \text{COMBINE}(\{X_v^{(k)}, M_v^{(k+1)}\}), \ \forall v \in \mathbf{V}, k < K \tag{2}$$

Here, $K$ is the number of GNN layers, $X_v^0 = X_v$ is initial feature vector of each node, and the final representation $X_v^K = Z_v$ serves as the input to the node-wise classifier. The AGG function performs topological aggregation by collecting information from neighboring nodes, while the COMBINE function performs semantic aggregation through processing the collected message for each node. Scalability is a crucial issue when applying GNNs to massive graphs. Ego graph, which defines the scope of information influencing the classification of a single node, exponentially increases with the number of GNN layers. Therefore, to ensure scalability, algorithms must be meticulously designed to efficiently utilize computation and memory resources (Hamilton et al., 2017; Shi et al., 2022; Zeng et al., 2019). Decoupled GNNs, whose process of collecting topological information occurs solely during preprocessing and is parameter-free, such as SGC (Wu et al., 2019), SIGN (Rossi et al., 2020), and GAMLP (Zhang et al., 2022), have demonstrated outstanding performance and scalability on many real-world datasets. Furthermore, SeHGNN (Yang et al., 2023) and RpHGNN (Hu et al., 2023) propose decoupled GNNs that efficiently apply to heterogeneous graphs by constructing separate embedding spaces for each metapath based on HGNN (Feng et al., 2019).

### 2.2 DOMAIN ADAPTATION

A machine will learn from a train domain in order to perform on a test domain. Domain adaptation is needed due to the discrepancy between train and test domains. That is, we can not guarantee that a model which performed well on the train domain, will perform well on the test domain. The performance on the test domain is known to depend on the performance of the train domain and the similarity between two domains (Ben-David et al., 2006; 2010; Germain et al., 2013; 2016).

For feature space $\mathcal{X}$ and label space $\mathcal{Y}$, the goal is to train a predictor function $f : \mathcal{X} \to \mathcal{Y}$ to minimize the risk $R_{tr}(f) = \mathbb{E}_{(X,Y) \sim P_{tr}} [L(f(X), Y)]$ where $P_{tr}$ is the distribution of the train

feature-label pairs, and $L$ is a loss function $L : \mathcal{Y} \times \mathcal{Y} \to \mathbb{R}$. We are interested in minimizing $R_{te}(f) = \mathbb{E}_{(X,Y) \sim P_{te}} [L(f(X), Y)]$ where $P_{te}$ is the distribution of the test feature-label pairs.

The domain adaptation bound, or upper bound of generalization error was firstly proposed for a binary classification task by defining the set of all trainable functions $\mathcal{F}$, symmetric hypothesis class $\mathcal{F} \Delta \mathcal{F}$ (Ben-David et al., 2006), and a metric for distributions, namely $d_{\mathcal{F} \Delta \mathcal{F}}$. $d_{\mathcal{F} \Delta \mathcal{F}}$ is the factor which represents the similarity between two distributions. Recently, theories and applications as setting up the metric between two distributions as the Wasserstein-1 distance, $W_1$ instead of $d_{\mathcal{F} \Delta \mathcal{F}}$ have been developed (Lee et al., 2019; Shen et al., 2018; Arjovsky et al., 2017; Redko et al., 2017). For brevity, we omit the assumptions introduced in (Redko et al., 2017) and simply state the theoretical domain adaptation bound below.

$$R_{te}(f) \leq R_{tr}(f) + W_1(\mathcal{D}_{tr}, \mathcal{D}_{te}) \tag{3}$$

where $\mathcal{D}_{tr}$ and $\mathcal{D}_{te}$ are marginal distributions on $\mathcal{X}$ of $\mathcal{P}_{tr}$ and $\mathcal{P}_{te}$, respectively.

### 2.3 Prior Studies

Despite its significance, studies on domain adaptation in GNNs are relatively scarce. Notably, to our knowledge, no studies propose invariant learning applicable to large graphs. For example, EERM (Wu et al., 2022) defines a graph editor that modifies the graph to obtain invariant features through reinforcement learning, which cannot be applied to decoupled GNNs. SR-GNN (Zhu et al., 2021) adjusts the distribution distance of representations using a regularizer, with computational complexity proportional to the square of the number of nodes, making it challenging to apply to large graphs. This scarcity is attributed by several factors: data from different environments may have interdependencies, and the extrapolating nature of environments complicates the problem.

## 3 Method Explanation

### 3.1 Motivation of Our Method: Imposing Invariance with Message Passing

The distribution of node connections depends on both timestamps and labels. As a result, even if features from previous layers are invariant, features after the message passing layers belong to different distributions: $\mathcal{D}_{tr}$ for training and $\mathcal{D}_{te}$ for testing. Imposing invariance here means aligning the mean and variance of $\mathcal{D}_{tr}$ and $\mathcal{D}_{te}$. While it may seem straightforward to compute and align the mean and variance for each label, this is impractical in real settings since test labels are unknown during prediction. To overcome this, we analyzed real-world temporal graphs and identified practical assumptions about connection distributions. Based on this, we propose a message passing method that corrects the discrepancy between $\mathcal{D}_{tr}$ and $\mathcal{D}_{te}$, ensuring feature invariance at each layer.

### 3.2 Problem Setting

Denote the possible temporal information as $\mathbf{T} = \{\ldots, t_{max} - 1, t_{max}\}$, $\mathbf{Y}$ as the set of labels, and $P_{tr}$ and $P_{te}$ as the joint probability distribution of feature-label pairs in train data and test data. The training data will be historical labels, that is, nodes with timestamp smaller than $t_{max}$. The test data will be recent labels, that is, nodes with timestamp $t_{max}$. Therefore, labels of nodes with time $t_{max}$ are unknown. We say that a variable is *invariant* if and only if it does not depend on $t$.

Here are the 3 assumptions introduced in this study.

$$\textit{Assumption 1} : P_{te}(Y) = P_{tr}(Y) \tag{4}$$

$$\textit{Assumption 2} : P_{te}(X|Y) = P_{tr}(X|Y) \tag{5}$$

$$\textit{Assumption 3} : \mathcal{P}_{yt}\left(\tilde{y}, \tilde{t}\right) = f(y, t)g\left(y, \tilde{y}, |\tilde{t} - t|\right), \ \forall y, \tilde{y} \in \mathbf{Y}, \forall t, \tilde{t} \in \mathbf{T} \tag{6}$$

From now on, we use $y$ and $t$ as the label and time of the target node, and $\tilde{y}$ and $\tilde{t}$ as the label and time of neighboring nodes, unless specified otherwise. Relative connectivity $\mathcal{P}_{yt}\left(\tilde{y}, \tilde{t}\right)$ denotes the probability distribution of label and time pairs of neighboring nodes. Hence, $\sum_{\tilde{y} \in \mathbf{Y}} \sum_{\tilde{t} \in \mathbf{T}} \mathcal{P}_{yt}\left(\tilde{y}, \tilde{t}\right) = 1$.

Assumptions 1 and 2 posit that the initial features and labels allocated to each node originate from same distributions. Assumption 3 assumes separability in the distribution of neighboring nodes. It is based on the observation that the proportion of nodes at time $\tilde{t}$ within the set of neighboring nodes of the target node at time $t$ decreases as the time difference $|\tilde{t} - t|$ increases. $g\left(y, \tilde{y}, |\tilde{t} - t|\right)$ is the function representing the proportion of neighboring nodes as a function on $|\tilde{t} - t|$. $f(y, t)$ is a

function to adjust relative proportion value $g\left(y, \tilde{y}, |\tilde{t} - t|\right)$ to construct $\mathcal{P}_{yt}\left(\tilde{y}, \tilde{t}\right)$ as a probability density function. Figure 2 graphically illustrates these properties. These assumptions are rooted in properties observable in real-world graphs. The motivation and analysis on real-world temporal graphs are provided in Appendix A.1.

### 3.3 OUTLINE OF IMPACT METHODS

In the analysis of IMPaCT methods, we will later define and use the first and second moment of distributions, which are simply the approximations of mean and variance. Occasionally, first moment and second moment are written as approximate expectation and approximate variance, respectively.

Figure 2: Graphical representation of functions $f$ and $g$. The shaded bars denote relative connectivity. Target node has label $y$, and only consider cases neighboring nodes with a labels $\tilde{y}$. The function $g(y, \tilde{y}, |\tilde{t} - t|)$ determines extent to which relative connectivity varies, and its scale is adjusted by the function $f(y, t)$.

Section 4 introduces the 1st moment alignment methods, MMP and PMP. These methods impose the invariance of the 1st moment among layers by modifying the original graph data. Formally, MMP and PMP ensures the aggregated message $M_v^{(k+1)}$ to approximately satisfy $P_{tr}(M_v^{(k+1)}|Y) = P_{te}(M_v^{(k+1)}|Y)$ when the representations $X_v^{(k)}$ at the $k$-th layer satisfies $P_{tr}(X_v^{(k)}|Y) = P_{te}(X_v^{(k)}|Y), \forall v \in \mathbf{V}$.

Section 5 introduces the 2nd moment alignment methods, PNY and JJNORM, which impose the invariance of the 2nd moment. These methods are not graph-modifying methods and should be applied over 1st moment alignment methods. Specifically, JJNORM algebraically alters the distribution of the final layer to impose 2nd moment invariance, without changing the 1st moment invariance property.

## 4 FIRST MOMENT ALIGNMENT METHODS

Message passing refers to the process of aggregating representations from neighboring nodes in the previous layer. Here, we assume the commonly used averaging message passing procedure. For any arbitrary target node $v \in \mathbf{V}$ with label $y$ and time $t$,

$$M_v^{(k+1)} = \frac{\sum_{\tilde{y} \in \mathbf{Y}} \sum_{\tilde{t} \in \mathbf{T}} \sum_{w \in \mathcal{N}_v\left(\tilde{y}, \tilde{t}\right)} X_w^{(k)}}{\sum_{\tilde{y} \in \mathbf{Y}} \sum_{\tilde{t} \in \mathbf{T}} |\mathcal{N}_v\left(\tilde{y}, \tilde{t}\right)|}, X_w^{(k)} \sim P_w^{(k)} \tag{7}$$

where $\mathcal{N}_v\left(\tilde{y}, \tilde{t}\right) = \{w \in \mathbf{V} \mid w$ is a neighbor of $v$ with $\tilde{y}$ and time $\tilde{t}\}$, $P_w^{(k)}$ is the distribution of random variable $X_w^{(k)}$, and $M_v^{(k+1)}$ is the aggregated message at node $v$ in the $k + 1$-th layer.

***The first moment of aggregated message.*** If the representations from the previous layer have means which are consistent across time, i.e., $\mathbb{E}[X_w^{(k)}] = \mu_X^{(k)}(\tilde{y})$ for $\forall w \in \mathcal{N}_v\left(\tilde{y}, \tilde{t}\right)$, we can calculate the approximate expectation, defined in Appendix A.4, as $\hat{\mathbb{E}}[M_v^{(k+1)}] = \sum_{\tilde{y} \in \mathbf{Y}} \sum_{\tilde{t} \in \mathbf{T}} \mathcal{P}_{yt}\left(\tilde{y}, \tilde{t}\right) \mu_X^{(k)}(\tilde{y})$. Here, we can observe that $\hat{\mathbb{E}}[M_v^{(k+1)}]$ depends on the target node's time $t$ due to $\mathcal{P}_{yt}\left(\tilde{y}, \tilde{t}\right)$. Our objective is to modify the spatial aggregation method to ensure invariance of the 1st moment and preserve it among layers.

### 4.1 PERSISTENT MESSAGE PASSING: PMP

We propose Persistent Message Passing (PMP) as one approach to achieve 1st moment invariance. For the target node $v$ with time $t$, consider the time $\tilde{t}$ of some neighboring node. For $\Delta = |\tilde{t} - t|$ where $0 < \Delta \leq |t_{max} - t|$, both $t + \Delta$ and $t - \Delta$ neighbor nodes can exist. However, nodes with $\Delta > |t_{max} - t|$ or $\Delta = 0$ are only possible when $\tilde{t} = t - \Delta$. Let $\mathbf{T}_t^{\text{double}} = \{\tilde{t} \in \mathbf{T} \mid 0 < |\tilde{t} - t| \leq |t_{max} - t|\}$ and $\mathbf{T}_t^{\text{single}} = \{\tilde{t} \in \mathbf{T} \mid |\tilde{t} - t| > |t_{max} - t| \text{ or } \tilde{t} = t\}$. The target node receives twice the weight from $\tilde{t} \in \mathbf{T}_t^{\text{double}}$ against $\tilde{t} \in \mathbf{T}_t^{\text{single}}$. Motivation behind PMP is to correct this by double weighting the neighbor nodes with time in $\mathbf{T}_t^{\text{single}}$.
In the case of figure 3, the target node's time is $t = 2018$, and by definition, $\mathbf{T}_t^{\text{double}} = \{2019, 2017\}, \mathbf{T}_t^{\text{single}} = \{2018, 2016, 2015, 2014, ...\}$. The neighbor nodes with time 2017 and

2019 have the same $\Delta = 1$, and hence by assumption 3, contribute equally when message passing to the target node. However, neighbor nodes with time in $\mathbf{T}_t^{single}$ do not have the "symmetric pairs", unlike 2017 having a "symmetric pair" 2019. Therefore, double nodes contribute twice more than single nodes when message passing. Hence, by multiplying 2 to the weight of single nodes, every node will contribute equally when message passing, regardless of the target node's time.

**Definition 4.1.** *The PMP from the $k$-th layer to the $k + 1$-th layer of target node $v$ is defined as:*

$$M_v^{pmp(k+1)} = \frac{\sum_{\tilde{y} \in \mathbf{Y}} \sum_{\tilde{t} \in \mathbf{T}_t^{single}} \sum_{w \in \mathcal{N}_v(\tilde{y},\tilde{t})} 2X_w^{(k)} + \sum_{\tilde{y} \in \mathbf{Y}} \sum_{\tilde{t} \in \mathbf{T}_t^{double}} \sum_{w \in \mathcal{N}_v(\tilde{y},\tilde{t})} X_w^{(k)}}{\sum_{\tilde{y} \in \mathbf{Y}} \sum_{\tilde{t} \in \mathbf{T}_t^{single}} 2|\mathcal{N}_v(\tilde{y},\tilde{t})| + \sum_{\tilde{y} \in \mathbf{Y}} \sum_{\tilde{t} \in \mathbf{T}_t^{double}} |\mathcal{N}_v(\tilde{y},\tilde{t})|} \tag{8}$$

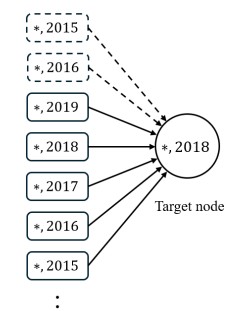

As noted, PMP is a graph modifying method. Neighbor nodes in $\mathbf{T}_t^{single}$ are duplicated in order to contribute equally with nodes in $\mathbf{T}_t^{double}$.

**Theorem 4.1.** *The 1st moment of aggregated message obtained by PMP layer is invariant, if the 1st moment of previous representation is invariant.*

**Sketch of proof** Let $\mathbb{E}[X_w^{(k)}] = \mu_X^{(k)}(\tilde{y})$ for $\forall w \in \mathcal{N}_v(\tilde{y},\tilde{t})$ as a function invariant with $\tilde{t}$. Then,

$$\hat{\mathbb{E}}\left[M_v^{pmp(k+1)}\right] = \frac{\sum_{\tilde{y} \in \mathbf{Y}} \sum_{\tau \geq 0} g(y,\tilde{y},\tau)\mu_X^{(k)}(\tilde{y})}{\sum_{\tilde{y} \in \mathbf{Y}} \sum_{\tau \geq 0} g(y,\tilde{y},\tau)} \tag{9}$$

*which is invariant with $t$. See Appendix A.5 for details and implementation.*

Figure 3: Graphical explanation of PMP

We provide a final remark that the initial layer of features must be experimentally ensured to have a time-invariant mean in order to apply PMP. Once the time invariance of the first moment in the initial layer is confirmed, the first moments of all subsequent layers are also time-invariant, as guaranteed by the previous theorem.

### 4.2 MONO-DIRECTIONAL MESSAGE PASSING: MMP

Besides PMP, there are numerous ways to adjust the 1st moment of train and test distributions to be invariant. We introduce Mono-directional Message Passing (MMP) as one such approach. MMP aggregates information only from neighboring nodes with time less or equal than the target node.

**Definition 4.2.** *The MMP from the $k$-th layer to the $k + 1$-th layer of target node $v$ is defined as:*

$$M_v^{mmp(k+1)} = \frac{\sum_{\tilde{y} \in \mathbf{Y}} \sum_{\tilde{t} \leq t} \sum_{w \in \mathcal{N}_v(\tilde{y},\tilde{t})} X_w^{(k)}}{\sum_{\tilde{y} \in \mathbf{Y}} \sum_{\tilde{t} \leq t} |\mathcal{N}_v(\tilde{y},\tilde{t})|} \tag{10}$$

**Theorem 4.2.** *The 1st moment of aggregated message obtained by MMP layer is invariant, if the 1st moment of previous representation is invariant.*

**Sketch of proof** Let $\mathbb{E}[X_w^{(k)}] = \mu_X^{(k)}(\tilde{y})$ for $\forall w \in \mathcal{N}_v(\tilde{y},\tilde{t})$ as a function invariant with $\tilde{t}$. Then,

$$\hat{\mathbb{E}}\left[M_v^{mmp(k+1)}\right] = \frac{\sum_{\tilde{y} \in \mathbf{Y}} \sum_{\tau \geq 0} g(y,\tilde{y},\tau)\mu_X^{(k)}(\tilde{y})}{\sum_{y \in \mathbf{Y}} \sum_{\tau \geq 0} g(y,\tilde{y},\tau)} \tag{11}$$

*which is also invariant with $t$. See Appendix A.6 for details and implementation.*

**Comparison between PMP and MMP.** Both PMP and MMP adjust the weights of messages collected from neighboring nodes that meet certain conditions, either doubling or ignoring their impact. They can be implemented easily by reconstructing the graph according to the conditions without altering the existing code. However, MMP collects less information since it only gathers information only from the past, resulting a smaller ego-graph. Therefore, PMP will be used as the 1st moment alignment method in the subsequent discussions. Furthermore, from Theorem 4.1, we will denote $\hat{\mathbb{E}}[M_v^{pmp(k+1)}]$ as $\mu_M^{pmp(k+1)}(y)$ for target node $v$ with label $y$ in the following discussions.

### 4.3 THEORETICAL ANALYSIS OF PMP WHEN APPLIED IN MULTI-LAYER GNNS.

We will assume the messages and representations to be scalar in this discussion. Now suppose that (i) $|M_v^{(k)}| \leq C$ almost surely for $\forall v \in \mathbf{V}$, $M_v^{(k)} \sim Q_v^{(k)}$, and (ii) $\mathrm{var}(M_v^{(k)}) \leq V$ for $\forall v \in \mathbf{V}$. Since we are considering 1st moment alignment method PMP, we may assume $\mathbb{E}[M_v^{(k)}] = \hat{\mathbb{E}}[M_v^{(k)}] = \mu_M^{(k)}(y)$ for $M_v^{(k)} \sim Q_v^{(k)}$, $\forall y \in \mathbf{Y}, \forall t \in \mathbf{T}$. Here, $W_1$ is the Wasserstein-1 metric of probability measures. We also assume G-Lipschitz condition for semantic aggregation functions $f^{(k)}$ for $\forall k \in \{1, 2, \ldots, K\}$. Detailed modelling of PMP with probability measures are in Appendix A.7, and

proofs of the following theorems are in Appendix A.8. For now on, we will omit the details and only state the theorems and provide interpretations of the theoretical results. $v$ and $v'$ in this section are nodes both having label $y$, but different times $t$ and $t'$, respectively.

**Theorem 4.3.** $W_1(Q_v^{(k)}, Q_{v'}^{(k)}) \leq \mathcal{O}(C^{1/3}V^{1/3})$

**Theorem 4.4.** $W_1(Q_v^{(k)}, Q_{v'}^{(k)}) \leq \mathcal{O}(\tau\sqrt{\log C})$ *if $Q_v^{(k)}, Q_{v'}^{(k)}$ are sub-Gaussians with constant $\tau$.*

Without PMP, we can only guarantee $W_1(Q_v^{(k)}, Q_{v'}^{(k)}) \leq 2C$, or $\mathcal{O}(C)$. However, PMP gives a tighter upper bound $\mathcal{O}(C^{1/3}V^{1/3})$. Furthermore, with additional assumption of sub-Gaussians, PMP gives a more significant upper bound $\mathcal{O}(\tau\sqrt{\log C})$.

**Theorem 4.5.** *If $W_1(Q_v^{(k)}, Q_{v'}^{(k)}) \leq W$ for $\forall v$ with label $y$ and time $t$, $\forall v'$ with label $y$ and time $t'$, then $W_1(Q_v^{(k+1)}, Q_{v'}^{(k+1)}) \leq \frac{G}{G^{(k)}}W$ where $G^{(k)} > 1$ is a constant only depending on the layer $k$.*

This theorem involves two steps. First, $W_1(Q_v^{(k)}, Q_{v'}^{(k)}) \leq W$ gives $W_1(P_v^{(k)}, P_{v'}^{(k)}) \leq GW$. Second, $W_1(P_v^{(k)}, P_{v'}^{(k)}) \leq GW$ gives $W_1(Q_v^{(k+1)}, Q_{v'}^{(k+1)}) \leq \frac{G}{G^{(k)}}W$ for $G^{(k)} > 1$, a constant only depending on the layer $k$. The strength of this inequality is that the denominator $G^{(k)}$ is larger than 1. For example, if we assume 1-Lipschitz property of aggregation functions, the upper bound of $W_1$ distance decreases layer by layer. The following corollary formulates this interpretation.

**Corollary 4.5.1.** $W_1(Q_v^{(k)}, Q_{v'}^{(k)}) \leq \frac{G^{K-1}}{G^{(1)}G^{(2)}...G^{(K-1)}}\mathcal{O}(\min\{C^{1/3}V^{1/3}, \tau\sqrt{\log C}\})$

Therefore, we ensured that the $W_1$ distance between train and test distributions of final representations are bounded when PMP is applied in multi-layer GNNs. In Section 2.2, we have previously introduced that the generalization error can be upper bounded by the $W_1$ distance. Hence, we have established a theoretical upper bound of the generalization error when PMP method is applied.

## 4.4 GENERALIZED PMP: GENPMP

Here, we note that the first moment is a good approximation for the mean only when the number of nodes with specific time label are similar to each other. Hence, if the dataset has a substantial difference among the number of nodes with specific time label, duplicating the single nodes as PMP will still adjust the 1st moment, but this will not be a good approximation for the mean. Therefore, we propose the Generalized PMP (GenPMP) for such datasets.

For the target node $v$ with time $t$, consider the time $\tilde{t}$ of some neighboring node. Instead of $\mathbf{T}_t^{\text{double}}$ and $\mathbf{T}_t^{\text{single}}$, we define $\mathbf{T}_t^{\Delta} = \{\tilde{t} \in \mathbf{T} \mid |\tilde{t} - t| = \Delta\}$ for $0 \leq \Delta \leq |t_{\max} - t|$. By collecting the nodes, we can get a discrete probability distribution $P_s$, where $P_s(\tau)$ is attained by adding $|\mathbf{T}_s^{\tau}|$ for all nodes with time label $s$, and then normalizing so that $\sum_{\tau \geq 0} P_s(\tau) = 1$.

**Definition 4.3.** *The generalized probabilistic message passing (GenPMP) from the $k$-th layer to the $(k+1)$-th layer of target node $v$ is defined as:*

$$M_v^{gpmp(k+1)} = \sum_{\tilde{y} \in \mathbf{Y}} \sum_{\Delta \geq 0} \sum_{\tilde{t} \in \mathbf{T}_t^{\Delta}} \sum_{w \in \mathcal{N}_v(\tilde{y}, \tilde{t})} \frac{P_{t_{\max}}(\Delta)}{P_{\tilde{t}}(\Delta)} X_w^{(k)} \tag{12}$$

Here, we are giving a relative weight to nodes $w$ in $\mathcal{N}_v(\tilde{y}, \tilde{t})$ by generating nodes with a ratio of $P_{t_{\max}}(\Delta)/P_{\tilde{t}}(\Delta)$. Unlike PMP which distinguishes neighbor nodes into only two classes, this method explicitly counts the nodes and adjusts the shape of distributions among train and test data. However, GenPMP has reduced adaptability. Unlike PMP, which can be implemented by simply adding or removing edges in the graph, GenPMP requires modifying the model to reflect real-valued edge weights during the message passing process. Moreover, when the number of nodes per times-tamp is equal, GenPMP behaves similarly to PMP. Theoretically, if the ratio $P_{t_{\max}}(\Delta)/P_{\tilde{t}}(\Delta)$ is too large for a fixed $\Delta$, the variance of aggregated message will increase by a factor roughly proportional to the square of the ratio by definition. Therefore, $V$ defined in Section 4.3 will increase substantially, and hence the theoretical upper bound of generalization error will also increase substantially since there is a factor $V^{1/3}$ in the bound. In conclusion, GenPMP is a method for exceptional usage on graph data which shows large differences of node numbers with specific timestamps.

## 5 SECOND MOMENT ALIGNMENT METHODS

While 1st order alignment methods like PMP and MMP preserve the invariance of the 1st moment of the aggregated message, they do not guarantee such property for the 2nd moment. Let's suppose that the 1st moment of the previous layer's representation $X$ is invariant with node's time $t$, and

2nd moment of the initial feature is invariant. That is, $\forall w \in \mathcal{N}\left(\tilde{y}, \tilde{t}\right)$, $\mathbb{E}[X_w^{(k)}] = \mu_X^{pmp(k)}(\tilde{y})$ for $X_w^{(k)} \sim P_w^{(k)}$, and $\Sigma_X^{pmp(0)}\left(\tilde{y}, \tilde{t}\right) = \Sigma_X^{pmp(0)}\left(\tilde{y}, t_{max}\right)$ where $\Sigma_X^{pmp(k)}\left(\tilde{y}, \tilde{t}\right) = \text{var}(X_w^{(k)}) = \mathbb{E}\left[(X_w^{(k)} - \mu_X^{pmp(k)}\left(\tilde{y}, \tilde{t}\right))(X_w^{(k)} - \mu_X^{pmp(0)}\left(\tilde{y}, \tilde{t}\right))^\top\right]$ for $X_w^{(k)} \sim P_w^{(k)}$. Given that the invariance of 1st moment is preserved after message passing by PMP or MMP, one naive idea for aligning the 2nd moment is to calculate the covariance matrix of the aggregated message $M_v^{pmp(k+1)}$ for each time $t$ of node $v$ and adjust for the differences. However, when $t = t_{max}$, we cannot directly estimate $\text{var}(M_v^{pmp(k+1)})$ since the labels are unknown for nodes in the test set. We introduce PNY and JJNORM, the methods for adjusting the aggregated message obtained using the PMP to achieve invariant property even for the 2nd moment, when the invariance for 1st moment is preserved.

***The second moment of aggregated message***. The *approximate* variance of $M_v^{pmp(k+1)}$ can also be calculated rigorously by using the definition of approximate variance in Appendix A.5, as:

$$\text{v\^ar}(M_v^{pmp(k+1)}) = \frac{\sum_{\tilde{y} \in \mathbf{Y}}\left(\sum_{\tilde{t} \in \mathbf{T}_t^{\text{single}}} 4\mathcal{P}_{yt}\left(\tilde{y}, \tilde{t}\right) + \sum_{\tilde{t} \in \mathbf{T}_t^{\text{double}}} \mathcal{P}_{yt}\left(\tilde{y}, \tilde{t}\right)\right) \Sigma_X^{pmp(k)}\left(\tilde{y}, \tilde{t}\right)}{\left(\sum_{\tilde{y} \in \mathbf{Y}} \sum_{\tilde{t} \in \mathbf{T}_t^{\text{single}}} 2\mathcal{P}_{yt}\left(\tilde{y}, \tilde{t}\right) + \sum_{\tilde{y} \in \mathbf{Y}} \sum_{\tilde{t} \in \mathbf{T}_t^{\text{double}}} \mathcal{P}_{yt}\left(\tilde{y}, \tilde{t}\right)\right)^2 |\mathcal{N}_{yt}|}$$ (13)

Hence, we can write $\text{v\^ar}(M_v^{pmp(k+1)}) = \Sigma_M^{pmp(k+1)}(y, t)$. Since $\Sigma_M^{pmp(k+1)}(y, t)$ is a covariance matrix, it is positive semi-definite, orthogonally diagonalized as $\Sigma_M^{pmp(k+1)}(y, t) = U_{yt} \Lambda_{yt} U_{yt}^{-1}$.

## 5.1 PERSISTENT NUMERICAL YIELD: PNY

If we can specify $\mathcal{P}_{yt}(\tilde{y}, \tilde{t})$ for $\forall y, \tilde{y} \in \mathbf{Y}, \forall t, \tilde{t} \in \mathbf{T}$, transformation of covariance matrix during the PMP process could be calculated. PNY numerically estimates the transformation of the covariance matrix during the PMP process, and determines an affine transformation to correct this variation.

**Definition 5.1.** *The PNY from the $k$-th layer to the $k + 1$-th layer of target node $v$ is defined as:*

*For affine transformation matrix $A_t = U_{yt_{max}} \Lambda_{yt_{max}}^{1/2} \Lambda_{yt}^{-1/2} U_{yt}^\top$,*

$$M_v^{PNY(k+1)} = A_t(M_v^{pmp(k+1)} - \mu_M^{pmp(k+1)}(y)) + \mu_M^{pmp(k+1)}(y)$$ (14)

Note that $M_v^{pmp(k+1)}$ is a random vector defined as 8, so $M_v^{PNY(k+1)}$ is also a random vector.

**Theorem 5.1.** *The 1st and 2nd moments of aggregated message after PNY transform is invariant, if the 1st and 2nd moments of previous representations are invariant.*

***Sketch of proof*** $\hat{\mathbb{E}}[M_v^{PNY(k+1)}] = \mu_M^{pmp(k+1)}(y)$, $\text{v\^ar}(M_v^{pmp(k+1)}) = \Sigma_M^{pmp(k+1)}(y, t_{max})$ holds,

*so the 1st and 2nd moments of representations are invariant with $t$. See Appendix A.10 for details.*

## 5.2 JUNCTION AND JUNCTION NORMALIZATION: JJNORM

A drawback of PNY is its complexity in handling covariance matrices, requiring computation of covariance matrices and diagonalization for each label and time of nodes, leading to high computational overhead. Additionally, estimation of $\mathcal{P}_{yt}\left(\tilde{y}, \tilde{t}\right)$ when $t$ or $\tilde{t}$ is $t_{max}$, necessitates solving overdetermined nonlinear systems of equations as Appendix A.9, making it difficult to analyze.

Assuming the function $g\left(y, \tilde{y}, |\tilde{t} - t|\right)$ to be consistent to $y$ and $\tilde{y}$ significantly simplifies the alignment of the 2nd moment. Here, we introduce JJNORM as a practical implementation of this idea.

*Assumption 4* : $g\left(y, \tilde{y}, \Delta\right) = g\left(y', \tilde{y}', \Delta\right), \forall y, \tilde{y}, y', \tilde{y}' \in \mathbf{Y}, \forall \Delta \in \{|t_2 - t_1| \mid t_1, t_2 \in \mathbf{T}\}$ (15)

Moreover, we will only consider GNNs with linear semantic aggregation functions. Formally,

$$M_v^{pmp(k+1)} \leftarrow \text{PMP}(X_w^{pmp(k)}, w \in \mathcal{N}_v)$$ (16)

$$X_v^{pmp(k+1)} \leftarrow A^{(k+1)} M_v^{pmp(k+1)}, \forall k < K, v \in \mathbf{V}$$ (17)

**Lemma 1.** *$\forall t \in \mathbf{T}$, there exists a constant $\alpha_t^{(k+1)} > 0$ only depending on $t$ and layer $k + 1$ such that*

$$\left(\alpha_t^{(k+1)}\right)^2 \Sigma_M^{pmp(k+1)}(y, t) = \Sigma_M^{pmp(k+1)}(y, t_{max}), \forall y \in \mathbf{Y}$$ (18)

*The covariance matrix of the aggregated message differs only by a constant factor depending on the layer $k$ and time $t$. Proof of this lemma is in Appendix A.11.2.*

**Definition 5.2.** *We define the constant of the final layer $\alpha_t = \alpha_t^{(K)} > 0$ as the JJ constant of node $v$.*

**Definition 5.3.** *The* JJNORM *is a normalization of the aggregated message to node* $v \in \mathbf{V} \setminus \mathbf{V}_{\cdot,t_{max}}$ *after the final layer of PMP defined as:*

$$M_v^{JJ} = \alpha_t \left( M_v^{pmp(K)} - \mu_M^{JJ}(y,t) \right) + \mu_M^{JJ}(y,t) \tag{19}$$

*where* $\mathbf{V}_{y,t} = \{u \in \mathbf{V} \mid u \text{ has label } y \text{ and time } t\}$, $\mathbf{V}_{\cdot,t} = \{u \in \mathbf{V} \mid u \text{ has time } t\}$, $\mu_M^{JJ}(y,t) = \frac{1}{|\mathbf{V}_{y,t}|} \sum_{w \in \mathbf{V}_{y,t}} M_w^{pmp(K)}$, $\mu_M^{JJ}(\cdot,t) = \frac{1}{|\mathbf{V}_{\cdot,t}|} \sum_{w \in \mathbf{V}_{\cdot,t}} M_w^{pmp(K)}$, *and* $\alpha_t$ *is the JJ constant.*

The 1st and 2nd moments of aggregated message processed by JJNORM, $\hat{\mathbb{E}}\left[M_v^{JJ}\right]$ and $\hat{\text{var}}\left(M_v^{JJ}\right)$, are defined differently from the definition above. Refer to Appendix A.11.1 for details.

**Theorem 5.2.** *The 1st and 2nd moments of aggregated message processed by* JJNORM *is invariant.*

***Sketch of proof*** *We can calculate* $\hat{\mathbb{E}}\left[M_v^{JJ}\right] = \mu_M^{pmp(K)}(y)$, *and* $\hat{\text{var}}\left(M_v^{JJ}\right) = \Sigma_M^{pmp(K)}(y,t_{max})$.

*So the 1st and 2nd moments of aggregated messages are invariant. See Appendix A.11.1 for details.*

We further present an unbiased estimator $\hat{\alpha}_t$ of $\alpha_t$. Refer to Appendix A.11.3 for derivation.

$$\hat{\alpha}_t = \frac{\frac{1}{|\mathbf{V}_{\cdot,t_{max}}|-1} \sum_{i \in \mathbf{V}_{\cdot,t_{max}}} (M_v^{pmp(K)} - \mu_M^{JJ}(\cdot,t_{max}))^2 - \frac{1}{|\mathbf{V}_{\cdot,t}|-1} \sum_{y \in \mathbf{Y}} \sum_{i \in \mathbf{V}_{y,t}} (\mu_M^{JJ}(y,t) - \mu_M^{JJ}(\cdot,t))^2}{\frac{1}{|\mathbf{V}_{\cdot,t}|-1} \sum_{y \in \mathbf{Y}} \sum_{i \in \mathbf{V}_{y,t}} (M_v^{JJ} - \mu_M^{JJ}(y,t))^2} \tag{20}$$

# 6 EXPERIMENTS

## 6.1 SYNTHETIC CHRONOLOGICAL SPLIT DATASET

**Temporal Stochastic Block Model(TSBM).** To assess the robustness and generalizability of proposed IMPaCT methods on graphs satisfying assumptions 1, 2, and 3, we conducted experiments on synthetic graphs. In order to create repeatable and realistic chronological graphs, we defined the Temporal Stochastic Block Model(TSBM) as our graph generation algorithm. TSBM can be regarded as a special case of the Stochastic Block Model(SBM) that incorporates temporal information of nodes (Holland et al., 1983; Deshpande et al., 2018). In the SBM, the probability matrix $\mathbf{P}_{y\tilde{y}}$ represents the probability of a connection between two nodes $i$ and $j$, where $y$ and $\tilde{y}$ denote the communities to which the nodes belong. Our study extends this concept to account for temporal information, differentiating communities based on both node labels and time. In the TSBM, the connection probability is represented by a 4-dimensional tensor $\mathbf{P}_{t\tilde{t}y\tilde{y}}$. We ensured that assumptions 1, 2, and 3 were satisfied. Specifically, the feature assigned to each node $\mathbf{x} \in \mathbb{R}^f$ was sampled from distributions depending solely on the label, defined as $\mathbf{x}_i = \mu(y) + k_y Z_i$. Here, $Z_i \in \mathbb{R}^f$ is an IID standard normal noise and $k_y$ represents the variance of features. To satisfy assumption 2, the time and label of each node were determined independently. To satisfy assumption 3, we first considered the possible forms of $g(y, \tilde{y}, |t - \tilde{t}|)$ and then determined $\mathbf{P}_{t\tilde{t}y\tilde{y}}$ accordingly. We used an exponentially decaying function with decay factor $\gamma_{y,\tilde{y}}$, defined as:

$$g(y, \tilde{y}, |t - \tilde{t}|) = \gamma_{y,\tilde{y}}^{|t-\tilde{t}|} g(y, \tilde{y}, 0), \ \forall |t - \tilde{t}| > 0 \tag{21}$$

**Experimental Setup.** For our experiments, we employed the fundamental decoupled GNN, Simple Graph Convolution (SGC) (Wu et al., 2019), as the baseline model. Additionally, we investigated whether the methods proposed in this study could improve the performance of general spatial GNNs. Hence, we used a 2-layer GCN(Kipf & Welling, 2017) that performs averaging message passing as another baseline model. We applied the MMP, PMP, PMP +PNY, and PMP +JJNORM methods to the baselines. Since the semantic aggregation of GCN is nonlinear, layer-wise JJNORM was applied, i.e. JJNORM could not be applied only to the aggregated message in the last layer but was applied to the aggregated message in each layer. To test the generalizability of JJNORM which is based on assumption 4, we experimented on graphs that both satisfy and do not satisfy assumption 4. Furthermore, for cases where Assumption 4 was satisfied, common decay factor $\gamma$ can be defined. A smaller $\gamma$ corresponds to a graph where the connection probability decreases drastically. We also compared the trends in the performance of each IMPaCT method by varying the value of $\gamma$. Detailed settings are provided in Appendix A.12. The results are presented in Table 3 and Figure 4.

## 6.2 REAL WORLD CHRONOLOGICAL SPLIT DATASET

To evaluate the performance of IMPaCT on real-world data, we used the ogbn-mag, ogbn-arxiv, and ogbn-papers100m datasets from the OGB benchmark (Hu et al., 2020), which include temporal

Table 2: Experimental results on real-world datasets. Bolded values represent SOTA.

| Dataset | Model | Test Acc. $\pm$ Std. | Valid Acc. $\pm$ Std. |
|---|---|---|---|
| ogbn-mag | LDHGNN (baseline) | $0.8789 \pm 0.0024$ | $0.8836 \pm 0.0028$ |
| | LDHGNN+MMP | $0.8945 \pm 0.0018$ | $0.8996 \pm 0.0015$ |
| | LDHGNN+PMP | $0.9093 \pm 0.0040$ | $0.9173 \pm 0.0027$ |
| | **LDHGNN+PMP+JJnorm** | $\mathbf{0.9178 \pm 0.0019}$ | $\mathbf{0.9236 \pm 0.0030}$ |
| ogbn-arxiv | Linear-RevGAT+GIANT (baseline) | $0.7065 \pm 0.0010$ | $0.7287 \pm 0.0009$ |
| | Linear-RevGAT+GIANT+GenPMP | $0.7468 \pm 0.0006$ | $0.7568 \pm 0.0010$ |
| | **RevGAT+SimTeG+TAPE** | $\mathbf{0.7803 \pm 0.0007}$ | $\mathbf{0.7846 \pm 0.0004}$ |
| ogbn-papers100m | GAMLP+GIANT+RLU (baseline) | $0.6967 \pm 0.0006$ | $0.7305 \pm 0.0004$ |
| | GAMLP+GIANT+RLU+GenPMP | $0.6976 \pm 0.0010$ | $0.7330 \pm 0.0006$ |
| | **GAMLP+GLEM+GIANT** | $\mathbf{0.7037 \pm 0.0002}$ | $\mathbf{0.7354 \pm 0.0001}$ |

information with chronological splits. The datasets used are summarized in Appendix A.3. While these datasets cover different tasks, they share the common goal of predicting the labels of the most recent papers in graphs with paper-type nodes, all framed as multi-class classification problems.

**ogbn-mag** The ogbn-mag dataset, which contains a balanced number of nodes from different time periods, is well-suited for applying PMP and JJNORM. Given the large scale of the graph, scalable approaches are essential, and many studies have utilized decoupled GNNs with linear semantic aggregation to tackle this challenge (Yang et al., 2023; Hu et al., 2023; Wong et al., 2024). This makes the IMPaCT method directly applicable. Consequently, ogbn-mag is a primary focus in our study. More details on this dataset can be found in Appendix A.2. We adopted LDHGNN (Wang, 2024) as the baseline model, which is built on RpHGNN (Hu et al., 2023) and incorporates a curriculum learning approach. RpHGNN, a decoupled GNN, balances the trade-off between information retention and computational efficiency in message passing by utilizing a random projection squashing technique. Since RpHGNN's overall semantic aggregation is linear, PMP and JJNORM can be seamlessly integrated. However, due to the graph's immense size, PNY was not applicable. Therefore, we applied MMP, PMP, and PMP +JJNORM to the baseline and compared their performance. Each experiment was repeated 8 times, with hyperparameters set according to Wong et al. (Wang, 2024). The details of the computing resources used for these experiments are described in Appendix A.3.

**ogbn-arxiv and ogbn-papers100m:** Both of these graph datasets exhibit significant variability in node counts across different time periods, making it challenging to directly improve performance using IMPaCT methods. Instead, we applied GenPMP to assess the generalizability of our approach. For ogbn-arxiv, since most studies utilize non-linear semantic aggregation, we modified the model to support GenPMP. Specifically, we utilized a linearized version of RevGAT (Li et al., 2021), replacing the GAT convolution layer with a Graph Convolution layer that performs linear topological aggregation. In the case of ogbn-papers100m, we selected the decoupled GNN, GAMLP+RLU, as our baseline. Additionally, in both experiments, we used GIANT embeddings (Chien et al., 2022) as initial features. We then applied GenPMP to these baselines to observe changes in test accuracy.

## 6.3 RESULTS

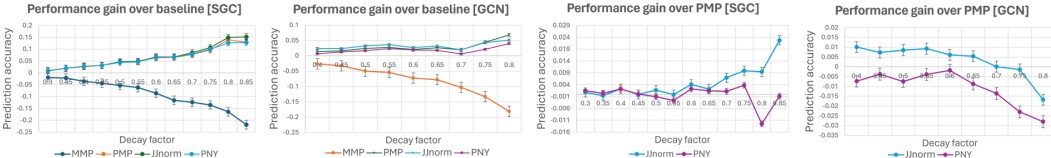

Figure 4: The left graphs show the performance gain of IMPaCT over the baseline. The right graphs illustrate the gain of 2nd moment alignment methods over the 1st moment alignment method PMP.

Table 3: Prediction accuracy and training time on synthetic graphs generated by TSBM. "Fixed $\gamma_{y_i y_j}$" represents scenarios satisfying Assumption 4, while "Random $\gamma_{y_i y_j}$" represents scenarios that do not. Time is reported for the entire training process over 200 epochs.

| | SGC | | | | | GCN | | | | |
|---|---|---|---|---|---|---|---|---|---|---|
| | Baseline | MMP | PMP | PNY | JJnorm | Baseline | MMP | PMP | PNY | JJnorm |
| Fixed $\gamma_{y_i y_j}$ | 0.2243 | 0.15 | 0.2653 | 0.2758 | **0.2777** | 0.2035 | 0.1439 | 0.2245 | 0.2178 | **0.2311** |
| Random $\gamma_{y_i y_j}$ | 0.1331 | 0.1063 | 0.1854 | 0.1832 | **0.1862** | 0.1298 | 0.1022 | 0.1565 | **0.1613** | 0.1609 |
| Time (sec) | 0.325 | 0.315 | 0.303 | 1.178 | 0.538 | 0.771 | 0.728 | 0.773 | 268.5 | 448.81 |

**Experimental results.** IMPaCT methods on both real and synthetic graphs showed superior performances. In the synthetic graph experiments, PMP provided a performance gain of 4.7% with

Table 4: Scalability of IMPaCT methods. "Graph." indicates method is applied by graph reconstruction. $N_c = |\mathbf{Y}||\mathbf{T}|$, $R$ is number of training epochs, and JJNORM †indicates layer-wise JJNORM.

| Method | Decoupled GNN | | General spatial GNN | |
|---|---|---|---|---|
| | Graph. | Message passing | Graph. | Message passing |
| MMP | $\mathcal{O}(|E|)$ | $\mathcal{O}(|E|fK)$ | $\mathcal{O}(|E|)$ | $\mathcal{O}(R|E|fK)$ |
| PMP | $\mathcal{O}(|E|)$ | $\mathcal{O}(|E|fK)$ | $\mathcal{O}(|E|)$ | $\mathcal{O}(R|E|fK)$ |
| GenPMP | $\mathcal{O}(|E|)$ | $\mathcal{O}(|E|fK)$ | $\mathcal{O}(|E|)$ | $\mathcal{O}(R|E|fK)$ |
| PNY | - | $\mathcal{O}(Kf^2(N_cf + N_c^2 + N) + |E|f)$ | - | $\mathcal{O}(RKf^2(N_cf + N_c^2 + N) + R|E|f)$ |
| JJNORM | - | $\mathcal{O}(Nf)$ | - | $\mathcal{O}(RNf)$ |
| JJNORM † | - | $\mathcal{O}(NfK)$ | - | $\mathcal{O}(RNfK)$ |

SGC and 2.4% with GCN over their respective baselines. 2nd moment alignment methods generally performed better than 1st moment alignment methods alone. JJNORM mostly performed better than PNY, except in cases where Assumption 4 did not hold and the baseline model was GCN. Experimental results further support the generalizability of our methods. Even with general spatial GNNs as the baseline, IMPaCT yielded significant performance gains. Furthermore, JJNORM improved performance over PMP even when Assumption 4 was violated.

On the real-world ogbn-mag dataset, applying PMP +JJNORM to LDHGNN showed a significant 3.8% performance improvement over state-of-the-art methods. Further experiments on the ogbn-arxiv and ogbn-papers100m datasets confirmed the generality of our assumptions across different datasets. On ogbn-arxiv, applying GenPMP to Linear-RevGAT+GIANT improved test accuracy by 4.0%, while on ogbn-papers100m, applying GenPMP to baseline resulted in a 0.09% improvement. All improvements were statistically significant. Based on these results, we offer guidelines for selecting IMPaCT models for chronological split datasets in Appendix A.13.

**Scalability.** The time complexity of IMPaCT methods is summarized in Table 4. Detailed analyses can be found in Appendix A.13. All methods exhibit linear complexity with respect to the number of nodes and edges. In particular, 1st order alignment methods can be implemented solely through graph modification, with no additional computational cost during the training process. Both MMP and PMP can be realized by simply adding or removing edges in the graph, while GenPMP is implemented by assigning weights to individual edges. Furthermore, when used in decoupled GNNs, PNY can be applied by adjusting the aggregated feature at each layer, and JJNORM can correct the final representation. Hence, all IMPaCT operations occur during preprocessing, offering not only high scalability but also adaptability.

However, when applied to general spatial GNNs, operations of 2nd order alignment methods are multiplied by the number of epochs, making it challenging to maintain scalability for PNY and JJNORM. While parallelization could speed up the computations, PNY requires eigenvalue decomposition of the covariance matrix, making parallelization difficult. In contrast, JJNORM can be parallelized by precomputing the mean feature values for each community (based on labels and time), allowing for efficient calculation of the scale factor $\alpha_t$ and feature updates. With efficient implementations, computation times can be reduced to an affordable level.

# 7 CONCLUSIONS

In this study, we addressed the domain adaptation challenges in graph data induced by chronological splits by proposing invariant message passing functions, IMPaCT. We analyzed and tackled the domain adaptation problem in graph datasets with chronological splits, presenting robust and realistic assumptions based on observable properties in real-world graphs. Based on these assumptions, we proposed IMPaCT, which preserves the invariance of both the 1st and 2nd moments of aggregated messages during the message passing step. We demonstrated its adaptability and scalability through experiments on both real-world citation graphs and synthetic graphs. Notably, on the ogbn-mag citation graph, we achieved substantial performance improvements over previous state-of-the-art methods, with a 3.0% increase in accuracy using the 1st moment alignment method and a 3.8% improvement when combining 1st and 2nd moment alignment. Furthermore, we validated the generality of our assumptions through experiments on the ogbn-arxiv and ogbn-papers100m.

**Limitations.** From an experimental standpoint, further investigation is needed to demonstrate the robustness and generalizability of IMPaCT across a wider variety of baseline models. Additionally, exploring the parallel implementation of JJNORM to assess how efficiently it can be applied to general spatial GNNs could be a topic for future work. From a theoretical perspective, our discussions were limited to spatial GNNs and assumed that the semantic aggregation satisfies the G-Lipschitz condition. Future work could focus on deriving tighter bounds under more realistic constraints.

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

## A APPENDIX

### A.1 MOTIVATION FOR ASSUMPTIONS

In this study, we make three assumptions regarding temporal graphs.

$$\textit{Assumption 1} : P_{te}(Y) = P_{tr}(Y) \tag{22}$$

$$\textit{Assumption 2} : P_{te}(X \mid Y) = P_{tr}(X \mid Y) \tag{23}$$

$$\textit{Assumption 3} : \mathcal{P}_{yt}(\tilde{y}, \tilde{t}) = f(y,t)g(y,\tilde{y}, \mid \tilde{t} - t \mid), \ \forall t, \tilde{t} \in \mathbf{T}, y, \tilde{y} \in \mathbf{Y} \tag{24}$$

Combining Assumption 1 and Assumption 2, we derive $P_{te}(X,Y) = P_{tr}(X,Y)$. This is a fundamental assumption in machine learning problems, implying that the initial feature distribution does not undergo significant shifts. However, in the real world graph dataset such as ogbn-mag dataset, features are embeddings derived from the abstracts of papers using a language model. It is crucial to verify whether these features remain constant over time, as per our assumption. Assumption 3 is a specific assumption derived from a close observation of the characteristics of temporal graphs, which requires justification based on actual data. To address these questions, we conducted a visual analysis based on the real-world temporal graph data from the ogbn-mag dataset. Statistics for ogbn-mag are provided in Appendix A.2.

#### A.1.1 INVARIANCE OF INITIAL FEATURES

First, to verify whether the distribution of features change over time, we calculated the average node features for each community, i.e., for each unique (label, time) pair. Our objective was to demonstrate that the distance between mean features of nodes with the same label but different times is significantly smaller than the distance between mean features of nodes with different labels. Given that the features are high-dimensional embeddings, using simple norms as distances might be inappropriate. Therefore, we employed the unsupervised learning method t-SNE (Van der Maaten & Hinton, 2008) to project these points onto a 2D plane, verifying that nodes with the same label but different times form distinct clusters. For the t-SNE analysis, we set the maximum number of iterations to 1000, perplexity to 30, and learning rate to 50.

We computed the mean feature for each community defined by the same (label, time) pair. Points corresponding to communities with the same label are represented in the same color. Thus, there are $|\mathbf{T}|$ points for each color, resulting in a total of $|\mathbf{Y}||\mathbf{T}|$ points in the left part of figure 5.

Given that the number of labels is $|\mathbf{Y}| = 349$, it is challenging to discern trends in a single graph displaying all points. The figure on the right considers only the 15 labels with the most nodes, redrawing the graph for clarity.

The clusters of nodes with the same color are clearly identifiable. While this analysis only consider 1st moment of initial feature of nodes, and does not confirm invariance for statistics other than the mean, it does show that the distance between mean features of nodes with the same label but different times is much smaller than the distance between mean features of nodes with different labels.

#### A.1.2 MOTIVATION FOR ASSUMPTION 3

Assumption 3 posits the separability of relative connectivity. Verifying this hypothesis numerically without additional assumptions about the connection distribution is challenging. Therefore, we aim to motivate Assumption 3 through a visualization of relative connectivity.

Consider fixing $y$ and $\tilde{y}$, and then examining the estimated relative connectivity $\mathcal{P}_{y,t}(\tilde{y}, \tilde{t})$ as a function of $t$ and $\tilde{t}$. Since $\mathcal{P}_{y,t}(\tilde{y}, \tilde{t}) = f(y,t)g(y,\tilde{y}, \mid \tilde{t} - t \mid)$, the graph of $\mathcal{P}_{y,t}(\tilde{y}, \tilde{t})$ for different $t$

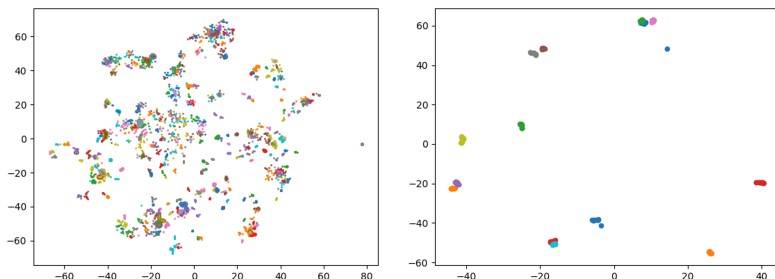

Figure 5: 2D projection of each community's mean feature by t-SNE. Points corresponding to communities with the same label are represented in the same color. [Left] Plot for all 349 labels, [Right] Plot for 15 labels with the most nodes.

should have similar shapes, differing only by a scaling factor determined by $f(y, t)$. In other words, by appropriately adjusting the scale, graphs for different $t$ should overlap.

Given $|\mathbf{Y}| = 349$, plotting this for all label pairs $y, \tilde{y}$ is impractical. Therefore we plotted graphs for few labels connected by a largest number of edges, plotting their relative connectivity. Although the ogbn-mag dataset is a directed graph, we treated it as undirected for defining neighboring nodes.

Graphs in different colors represent different target node times $t$, with the X-axis showing the relative time $\tilde{t} - t$ for neighboring nodes. Nodes with times $t = 2018$ and $t = 2019$ were excluded since they belong to the validation and test datasets, respectively. The data presented in graph 6 are the unscaled relative connectivity.

While plotting these graphs for all label pairs is infeasible, we calculated the weighted average relative connectivity for cases where $y = \tilde{y}$ and $y \neq \tilde{y}$ to understand the overall distribution. Specifically, for each $t$, we plotted the following values:

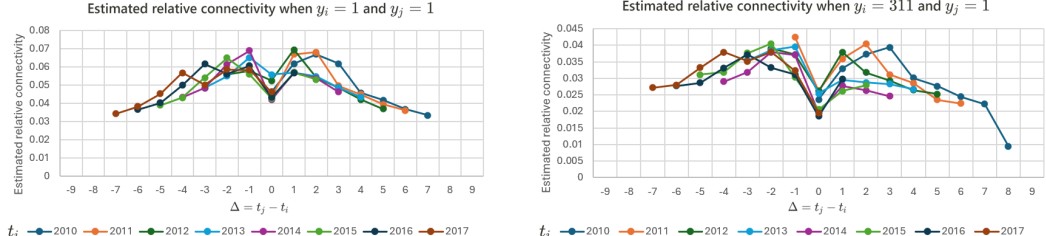

Figure 6: Estimated relative connectivity. [Left] when $y = 1$ and $\tilde{y} = 1$, [Right] when $y = 311$ and $\tilde{y} = 1$.

$$P_{\text{Same label}}(t, \tilde{t}) = \frac{|\{(u, v) \in E \mid u, v \text{ have same label}, u \text{ has time } t, v \text{ has time } \tilde{t}\}|}{|\{(u, v) \in E \mid u, v \text{ have same label}\}|} \quad (25)$$

$$P_{\text{Diff label}}(t, \tilde{t}) = \frac{|\{(u, v) \in E \mid u, v \text{ have different label}, u \text{ has time } t, v \text{ has time } \tilde{t}\}|}{|\{(u, v) \in E \mid u, v \text{ have different label}\}|} \quad (26)$$

These statistics represent the weighted average relative connectivity for each $y, t$ pair, weighted by the number of communities defined by each $(y, t)$ pair. Data for $t = 2018$ and $t = 2019$ were excluded, and no scaling corrections were applied.

The graphs 6, 7 reveal that the shape of graphs for different $t$ are similar and symmetric, supporting Assumption 3. Although this analysis is not a formal proof, it serves as a necessary condition that supports the validity of the separability assumption.

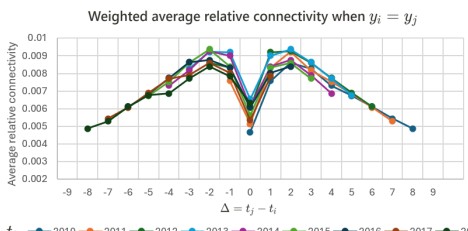 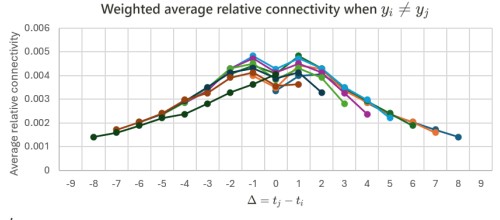

Figure 7: Weighted average relative connectivity. [Left] when $y = \tilde{y}$, [Right] when $y \neq \tilde{y}$.

## A.2 TOY EXPERIMENT

The purpose of toy experiment was to compare test accuracy obtained when dataset was split chronologically and split randomly regardless of time information. We further investigated whether incorporating temporal information in the form of time positional encoding significantly influences the distribution of neighboring nodes.

We conduct this toy experiment on ogbn-mag, a chronological heterogeneous graph within the Open Graph Benchmark (Hu et al., 2020), comprising Paper, Author, Institution, and Fields of study nodes. Only paper nodes feature temporal information. Detailed node and edge statistics of ogbn-mag dataset are provided in table 6 and 5. In this graph, paper nodes are divided into train, validation, and test nodes based on publication year, with the objective of classifying test and validation nodes into one of 349 labels. The performance metric is accuracy, representing the proportion of correctly labeled nodes among all test nodes.

Initial features were assigned only to paper nodes. In the chronological split dataset, nodes from the year 2019 were designated as the test set, while nodes from years earlier than 2018 were assigned to the training set. Time positional embedding was implemented using sinusoidal signals with 20 floating-point numbers, and these embeddings were concatenated with the initial features.

Table 5: Type and number of nodes in ogbn-mag.

| Node type | #Train nodes | #Validation nodes | #Test nodes |
|---|---|---|---|
| Paper | 59,965 | 64,879 | 41,939 |
| Author | 1,134,649 | 0 | 0 |
| Institution | 8,740 | 0 | 0 |
| Field of study | 59,965 | 0 | 0 |

Table 6: Type and number of edges in obgn-mag. * indicates the type of edges connect nodes with temporal information.

| Source type | Edges type | Destination type | #Edges |
|---|---|---|---|
| Author | affiliated with | Institution | 1,043,998 |
| Author | writes | Paper | 7,145,660 |
| Paper | cites* | Paper | 5,416,271 |
| Paper | has a topic of | Fields of study | 7,505,078 |

SeHGNN (Yang et al., 2023) was employed as baseline model for experimentation. The rationale for employing SeHGNN lies in its ability to aggregate semantics from diverse metapaths, thereby ensuring expressiveness, while also enabling fast learning due to neighbor aggregation operations being performed only during preprocessing. Each experiment was conducted four times using different random seeds. The hyperparameters and settings used in the experiments were identical to those presented by Yang et al. (Yang et al., 2023).

Preprocessing was performed on a 48 core 2X Intel Xeon Platinum 8268 CPU machine with 768GB of RAM. Training took place on a NVIDIA Tesla P100 GPU machine with 28 Intel Xeon E5-2680 V4 CPUs and 128GB of RAM.

## A.3 SETTINGS FOR EXPERIMENTS ON REAL-WORLD DATASETS.

Summary of statistics on ogbn-mag, ogbn-arxiv, and ogbn-papers100m are shown in table **??**.

Table 7: Summary of real-world graph datasets used in the experiments.

| Dataset | Type | Task | #Nodes | #Edges |
|---|---|---|---|---|
| ogbn-mag | Heterogeneous | Classify 349 venues for each paper | 1,939,743 | 21,111,007 |
| ogbn-arxiv | Homogeneous | Classify 40 primary categories for arXiv papers | 169,343 | 1,166,243 |
| ogbn-papers100m | Homogeneous | Classify 172 subject areas for arXiv papers | 111,059,956 | 1,615,685,872 |

The experiments on the ogbn-papers100m dataset, due to its large size, were conducted on a Tesla A100 GPU machine with 88 Intel Xeon 2680 CPUs and 1007 GB of RAM. On average, training one model took 5 hours and 10 minutes.

For the experiments on ogbn-mag and ogbn-arxiv, we used a Tesla P100 GPU machine with 28 Intel Xeon 2680 CPUs and 128 GB of RAM. Training on the ogbn-mag dataset, using LDHGNN as the baseline, took an average of 5 hours and 40 minutes per run. For the ogbn-arxiv dataset, experiments using the Linearized RevGAT + GIANT-XRT baseline took approximately 8 hours per run.

## A.4 FIRST AND SECOND MOMENT OF AVERAGING MESSAGE PASSING

### A.4.1 FIRST MOMENT AS APPROXIMATE OF EXPECTATION

We define the first moment of averaging message as the following steps:

(a) Take the expectation of the averaged message.
(b) Approximate $|\mathcal{N}_v(\tilde{y}, \tilde{t})|$ as $P_{yt}(\tilde{y}, \tilde{t})|\mathcal{N}_v|$ until the discrete values, i.e., the number of elements terms $|\mathcal{N}_v(\tilde{y}, \tilde{t})|$ and $|\mathcal{N}_v|$ disappear.

Because of step (b), we are defining the "approximate of expectation" as the first moment of a message. Denote the first moment of averaged message $M_v^{(k+1)}$ as $\hat{\mathbb{E}}[M_v^{(k+1)}]$. Deliberate calculations are as follows:

$$\hat{\mathbb{E}}[M_v^{(k+1)}] \overset{(a)}{=} \mathbb{E}\left[\frac{\sum_{\tilde{y}\in\mathbf{Y}}\sum_{\tilde{t}\in\mathbf{T}}\sum_{w\in\mathcal{N}_v(\tilde{y},\tilde{t})} X_w^{(k)}}{\sum_{\tilde{y}\in\mathbf{Y}}\sum_{\tilde{t}\in\mathbf{T}}|\mathcal{N}_v(\tilde{y},\tilde{t})|}\right] \tag{27}$$

$$\overset{(b)}{=} \mathbb{E}\left[\frac{\sum_{\tilde{y}\in\mathbf{Y}}\sum_{\tilde{t}\in\mathbf{T}}\sum_{w\in\mathcal{N}_v(\tilde{y},\tilde{t})} X_w^{(k)}}{\sum_{\tilde{y}\in\mathbf{Y}}\sum_{\tilde{t}\in\mathbf{T}} P_{yt}(\tilde{y},\tilde{t})|\mathcal{N}_v|}\right] \tag{28}$$

$$= \frac{1}{|\mathcal{N}_v|}\mathbb{E}\left[\sum_{\tilde{y}\in\mathbf{Y}}\sum_{\tilde{t}\in\mathbf{T}}\sum_{w\in\mathcal{N}_v(\tilde{y},\tilde{t})} X_w^{(k)}\right] \tag{29}$$

$$= \frac{1}{|\mathcal{N}_v|}\sum_{\tilde{y}\in\mathbf{Y}}\sum_{\tilde{t}\in\mathbf{T}}\mathbb{E}\left[\sum_{w\in\mathcal{N}_v(\tilde{y},\tilde{t})} X_w^{(k)}\right] \tag{30}$$

$$= \frac{1}{|\mathcal{N}_v|}\sum_{\tilde{y}\in\mathbf{Y}}\sum_{\tilde{t}\in\mathbf{T}}|\mathcal{N}_v(\tilde{y},\tilde{t})|\mu_X^{(k)}(\tilde{y}) \tag{31}$$

$$= \sum_{\tilde{y}\in\mathbf{Y}}\sum_{\tilde{t}\in\mathbf{T}}\frac{|\mathcal{N}_v(\tilde{y},\tilde{t})|}{|\mathcal{N}_v|}\mu_X^{(k)}(\tilde{y}) \tag{32}$$

$$\overset{(b)}{=} \sum_{\tilde{y}\in\mathbf{Y}}\sum_{\tilde{t}\in\mathbf{T}}\mathcal{P}_{yt}(\tilde{y},\tilde{t})\mu_X^{(k)}(\tilde{y}) \tag{33}$$

The final term of the equation above does not incorporate any discrete values $|\mathcal{N}_v(\tilde{y},\tilde{t})|$ and $|\mathcal{N}_v|$, so the step ends.

Note that we can calculate the first moment reversely as follows:

$$M_v^{(k+1)} = \frac{\sum_{\tilde{y} \in \mathbf{Y}} \sum_{\tilde{t} \in \mathbf{T}} \sum_{w \in \mathcal{N}_v(\tilde{y}, \tilde{t})} X_w^{(k)}}{\sum_{\tilde{y} \in \mathbf{Y}} \sum_{\tilde{t} \in \mathbf{T}} |\mathcal{N}_v(\tilde{y}, \tilde{t})|} \tag{34}$$

$$= \frac{\sum_{\tilde{y} \in \mathbf{Y}} \sum_{\tilde{t} \in \mathbf{T}} \frac{|\mathcal{N}_v(\tilde{y}, \tilde{t})|}{|\mathcal{N}_v|} \sum_{w \in \mathcal{N}_v(\tilde{y}, \tilde{t})} \frac{X_w^{(k)}}{|\mathcal{N}_v(\tilde{y}, \tilde{t})|}}{\sum_{\tilde{y} \in \mathbf{Y}} \sum_{\tilde{t} \in \mathbf{T}} \frac{|\mathcal{N}_v(\tilde{y}, \tilde{t})|}{|\mathcal{N}_v|}} \tag{35}$$

$$\simeq \frac{\sum_{\tilde{y} \in \mathbf{Y}} \sum_{\tilde{t} \in \mathbf{T}} \mathcal{P}_{yt}(\tilde{y}, \tilde{t}) \sum_{w \in \mathcal{N}_v(\tilde{y}, \tilde{t})} \frac{X_w^{(k)}}{|\mathcal{N}_v(\tilde{y}, \tilde{t})|}}{\sum_{\tilde{y} \in \mathbf{Y}} \sum_{\tilde{t} \in \mathbf{T}} \mathcal{P}_{yt}(\tilde{y}, \tilde{t})} \tag{36}$$

$$= \sum_{\tilde{y} \in \mathbf{Y}} \sum_{\tilde{t} \in \mathbf{T}} \left( \mathcal{P}_{yt}(\tilde{y}, \tilde{t}) \sum_{w \in \mathcal{N}_v(\tilde{y}, \tilde{t})} \frac{X_w^{(k)}}{|\mathcal{N}_v(\tilde{y}, \tilde{t})|} \right) \tag{37}$$

Take the expectation on both sides to derive

$$\mathbb{E}\left[M_v^{(k+1)}\right] \simeq \mathbb{E}\left[\sum_{\tilde{y} \in \mathbf{Y}} \sum_{\tilde{t} \in \mathbf{T}} \left( \mathcal{P}_{yt}(\tilde{y}, \tilde{t}) \sum_{w \in \mathcal{N}_v(\tilde{y}, \tilde{t})} \frac{X_w^{(k)}}{|\mathcal{N}_v(\tilde{y}, \tilde{t})|} \right)\right] \tag{38}$$

$$= \sum_{\tilde{y} \in \mathbf{Y}} \sum_{\tilde{t} \in \mathbf{T}} \left( \mathcal{P}_{yt}(\tilde{y}, \tilde{t}) \sum_{w \in \mathcal{N}_v(\tilde{y}, \tilde{t})} \frac{\mathbb{E}[X_w^{(k)}]}{|\mathcal{N}_v(\tilde{y}, \tilde{t})|} \right) \tag{39}$$

$$= \sum_{\tilde{y} \in \mathbf{Y}} \sum_{\tilde{t} \in \mathbf{T}} \left( \mathcal{P}_{yt}(\tilde{y}, \tilde{t}) \mu_X^{(k)}(\tilde{y}) \right) \tag{40}$$

### A.4.2 SECOND MOMENT AS APPROXIMATE OF VARIANCE

We define the second moment of averaging message as the following steps:

(a) Take the variance of the averaged message.
(b) Approximate $|\mathcal{N}_v(\tilde{y}, \tilde{t})|$ as $\mathcal{P}_{yt}(\tilde{y}, \tilde{t})|\mathcal{N}_v|$ until the discrete value terms $|\mathcal{N}_v(\tilde{y}, \tilde{t})|$ disappear.

Because of step (b), we are defining the "approximate of variance" as the second moment of a message. Denote the second moment of averaged message $M_v^{(k+1)}$ as $\hat{\mathrm{var}}(M_v^{(k+1)})$. Deliberate calculations are as follows:

$$\hat{\mathrm{var}}(M_v^{(k+1)}) \stackrel{(a)}{=} \mathrm{var}\left( \frac{\sum_{\tilde{y} \in \mathbf{Y}} \sum_{\tilde{t} \in \mathbf{T}} \sum_{w \in \mathcal{N}_v(\tilde{y}, \tilde{t})} X_w^{(k)}}{\sum_{\tilde{y} \in \mathbf{Y}} \sum_{\tilde{t} \in \mathbf{T}} |\mathcal{N}_v(\tilde{y}, \tilde{t})|} \right) \tag{41}$$

$$\stackrel{(b)}{=} \mathrm{var}\left( \frac{\sum_{\tilde{y} \in \mathbf{Y}} \sum_{\tilde{t} \in \mathbf{T}} \sum_{w \in \mathcal{N}_v(\tilde{y}, \tilde{t})} X_w^{(k)}}{\sum_{\tilde{y} \in \mathbf{Y}} \sum_{\tilde{t} \in \mathbf{T}} \mathcal{P}_{yt}(\tilde{y}, \tilde{t})|\mathcal{N}_v|} \right) \tag{42}$$

$$= \frac{\sum_{\tilde{y} \in \mathbf{Y}} \sum_{\tilde{t} \in \mathbf{T}} \sum_{w \in \mathcal{N}_v(\tilde{y}, \tilde{t})} \mathrm{var}(X_w^{(k)})}{\left( \sum_{\tilde{y} \in \mathbf{Y}} \sum_{\tilde{t} \in \mathbf{T}} \mathcal{P}_{yt}(\tilde{y}, \tilde{t})|\mathcal{N}_v| \right)^2} \tag{43}$$

$$= \frac{1}{|\mathcal{N}_v|^2} \sum_{\tilde{y} \in \mathbf{Y}} \sum_{\tilde{t} \in \mathbf{T}} \sum_{w \in \mathcal{N}_v(\tilde{y}, \tilde{t})} \mathrm{var}(X_w^{(k)}) \tag{44}$$

If we assume $\mathrm{var}(X_w) = \Sigma_X^{(k)}(\tilde{y})$ for $\forall w \in \mathcal{N}_v(\tilde{y}, \tilde{t})$,

$$\hat{\text{var}}(M_v^{(k+1)}) = \frac{1}{|\mathcal{N}_v|^2} \sum_{\tilde{y} \in \mathbf{Y}} \sum_{\tilde{t} \in \mathbf{T}} \sum_{w \in \mathcal{N}_v(\tilde{y},\tilde{t})} \Sigma_X^{(k)}(\tilde{y}) \tag{45}$$

$$= \frac{1}{|\mathcal{N}_v|^2} \sum_{\tilde{y} \in \mathbf{Y}} \sum_{\tilde{t} \in \mathbf{T}} |\mathcal{N}_v(\tilde{y},\tilde{t})| \Sigma_X^{(k)}(\tilde{y}) \tag{46}$$

$$\stackrel{(b)}{=} \frac{1}{|\mathcal{N}_v|^2} \sum_{\tilde{y} \in \mathbf{Y}} \sum_{\tilde{t} \in \mathbf{T}} |\mathcal{N}_v| \mathcal{P}_{yt}(\tilde{y},\tilde{t}) \Sigma_X^{(k)}(\tilde{y}) \tag{47}$$

$$= \frac{1}{|\mathcal{N}_v|} \sum_{\tilde{y} \in \mathbf{Y}} \sum_{\tilde{t} \in \mathbf{T}} \mathcal{P}_{yt}(\tilde{y},\tilde{t}) \Sigma_X^{(k)}(\tilde{y}) \tag{48}$$

## A.5  Explanation of PMP

### A.5.1  1st moment of aggregated message obtained by PMP layer.

We define the 1st moment of PMP with the identical steps of the 1st moment of averaging message passing, as in Appendix A.4.

### A.5.2  Proof of Theorem 4.1

From now on, we will denote $y$ and $t$ as the label and time belongs to target node, $v$, if there are no other specifications.

Suppose that the 1st moment of the representations from the previous layer is invariant. In other words, $\mu_X^{(k)}(y,t) = \mu_X^{(k)}(y,t_{max})$, $\forall t \in \mathbf{T}$.

Formally, when defined as $\mathcal{N}_v^{\text{single}} = \{u \in \mathcal{N}_v | u \text{ has time in } \mathbf{T}_v^{\text{single}}\}$, and $\mathcal{N}_v^{\text{double}} = \{u \in \mathcal{N}_v | u \text{ has time in } \mathbf{T}_v^{\text{double}}\}$, the message passing mechanism of PMP can be expressed as:

$$M_v^{pmp(k+1)} = \frac{\sum_{\tilde{y} \in \mathbf{Y}} \sum_{\tilde{t} \in \mathbf{T}_t^{\text{single}}} \sum_{w \in \mathcal{N}_v(\tilde{y},\tilde{t})} 2X_w^{(k)} + \sum_{\tilde{y} \in \mathbf{Y}} \sum_{\tilde{t} \in \mathbf{T}_t^{\text{double}}} \sum_{w \in \mathcal{N}_v(\tilde{y},\tilde{t})} X_w^{(k)}}{\sum_{\tilde{y} \in \mathbf{Y}} \sum_{\tilde{t} \in \mathbf{T}_t^{\text{single}}} 2|\mathcal{N}_v(\tilde{y},\tilde{t})| + \sum_{\tilde{y} \in \mathbf{Y}} \sum_{\tilde{t} \in \mathbf{T}_t^{\text{double}}} |\mathcal{N}_v(\tilde{y},\tilde{t})|} \tag{49}$$

The representations from the previous layer are invariant, i.e., $\mathbb{E}[X_w^{(k)}] = \mu_X^{(k)}(y)$. The first moment is calculated rigorously as shown in Appendix A.4 as follows.

$$\hat{\mathbb{E}}\left[M_v^{pmp(k+1)}\right] = \frac{\sum_{\tilde{y} \in \mathbf{Y}} \sum_{\tilde{t} \in \mathbf{T}_t^{\text{single}}} 2\mathcal{P}_{yt}(\tilde{y},\tilde{t})\mu_X^{(k)}(\tilde{y}) + \sum_{\tilde{y} \in \mathbf{Y}} \sum_{\tilde{t} \in \mathbf{T}_t^{\text{double}}} \mathcal{P}_{yt}(\tilde{y},\tilde{t})\mu_X^{(k)}(\tilde{y})}{\sum_{\tilde{y} \in \mathbf{Y}} \sum_{\tilde{t} \in \mathbf{T}_t^{\text{single}}} 2\mathcal{P}_{yt}(\tilde{y},\tilde{t}) + \sum_{\tilde{y} \in \mathbf{Y}} \sum_{\tilde{t} \in \mathbf{T}_t^{\text{double}}} \mathcal{P}_{yt}(\tilde{y},\tilde{t})} \tag{50}$$

$$= \frac{\sum_{\tilde{y} \in \mathbf{Y}} \left( \sum_{\tilde{t} \in \mathbf{T}_t^{\text{single}}} 2\mathcal{P}_{yt}(\tilde{y},\tilde{t}) + \sum_{\tilde{t} \in \mathbf{T}_t^{\text{double}}} \mathcal{P}_{yt}(\tilde{y},\tilde{t}) \right) \mu_X^{(k)}(\tilde{y})}{\sum_{\tilde{y} \in \mathbf{Y}} \left( \sum_{\tilde{t} \in \mathbf{T}_t^{\text{single}}} 2\mathcal{P}_{yt}(\tilde{y},\tilde{t}) + \sum_{\tilde{t} \in \mathbf{T}_t^{\text{double}}} \mathcal{P}_{yt}(\tilde{y},\tilde{t}) \right)} \tag{51}$$

By assumption 3,

$$\sum_{\tilde{t} \in \mathbf{T}_t^{\text{single}}} 2\mathcal{P}_{yt}(\tilde{y}, \tilde{t}) + \sum_{\tilde{t} \in \mathbf{T}_t^{\text{double}}} \mathcal{P}_{yt}(\tilde{y}, \tilde{t}) \tag{52}$$

$$= f(y,t) \left( \sum_{\tilde{t} \in \mathbf{T}_t^{\text{single}}} 2g(y, \tilde{y}, |\tilde{t} - t|) + \sum_{\tilde{t} \in \mathbf{T}_t^{\text{double}}} g(y, \tilde{y}, |t - t|) \right) \tag{53}$$

$$= f(y,t) \left( 2g(y, \tilde{y}, 0) + 2 \sum_{\tau > |t_{max} - t|} g(y, \tilde{y}, \tau) + \sum_{0 < \tau \le |t_{max} - t|} g(y, \tilde{y}, \tau) \right) \tag{54}$$

$$= 2f(y,t) \sum_{\tau \ge 0} g(y, \tilde{y}, \tau) \tag{55}$$

Substituting this into the previous expression yields,

$$\hat{\mathbb{E}} \left[ M_v^{pmp(k+1)} \right] = \frac{\sum_{\tilde{y} \in \mathbf{Y}} \sum_{\tau \ge 0} g(y, \tilde{y}, \tau) \mu_X^{(k)}(\tilde{y})}{\sum_{\tilde{y} \in \mathbf{Y}} \sum_{\tau \ge 0} g(y, \tilde{y}, \tau)} \tag{56}$$

Since there is no $t$ term in this expression, the mean of this aggregated message is invariant with respect to the target node's time.

---

**Algorithm 1:** Persistent Message PassingPersistent Message Passing as neighbor aggregation

---

**Input** : Undirected graph $\mathcal{G}(\mathbf{V}, \mathbf{E})$; input features $X_v, \forall v \in \mathbf{V}$; number of layers $K$; node time function $time : \mathbf{V} \to \mathrm{R}$; maximum time value $t_{\min}$; minimum time value $t_{\min}$; aggregate functions AGG; combine functions COMBINE; multisets of neighborhood $\mathcal{N}_v, \forall v \in \mathbf{V}$

**Output:** Final embeddings $\mathbf{z}_v, \forall v \in \mathbf{V}$

1   $h_v^0 \leftarrow X_v, \forall v \in \mathbf{V}$;
2   **for** $k = 0...K - 1$ **do**
3     **for** $v \in \mathbf{V}$ **do**
4       $\mathcal{N}'(v) \leftarrow \mathcal{N}(v)$;
5       **if** $|time(u) - time(v)| > \min(t_{\max} - time(v), time(v) - t_{\min})$ **then**
6        $\mathcal{N}'(v).\text{insert}(u)$ ;
7       $M_v^{(k+1)} \leftarrow \text{AGG}(\{h_u^{(k)}, \forall u \in \mathcal{N}'(v)\})$;
8       $X_v^{(k+1)} \leftarrow \text{COMBINE}(\{X_v^{(k)}, M_v^{(k+1)}\})$;
9     **end**
10 **end**
11 $z_v \leftarrow X_v^K, \forall v \in \mathbf{V}$ ;

---

**Algorithm 2:** Persistent Message PassingPersistent Message Passing as graph reconstruction

---

**Input** : Undirected graph $\mathcal{G}(\mathbf{V}, \mathbf{E})$; adjacency matrix $A^{\mathcal{G}} \in \mathbb{R}^{N \times N}$; node time function $time : \mathbf{V} \to \mathbb{R}$; maximum time value $t_{\max}$; minimum time value $t_{\min}$

**Output:** New directed graph $\mathcal{G}'(\mathbf{V}, \mathbf{E}')$; new adjacency matrix $A^{\mathcal{G}'}$

1   $A^{\mathcal{G}'} \leftarrow A^{\mathcal{G}}$ ;
2   **for** $(u, v) \in \mathbf{V}^2$ **do**
3     **if** $|time(u) - time(v)| > \min(t_{\max} - time(v), time(v) - t_{\min})$ **then**
4       $A_{uv}^{\mathcal{G}'} \leftarrow 2A_{uv}^{\mathcal{G}'}$ ;
5   **end**

---

### A.5.3 2ND MOMENT OF AGGREGATED MESSAGE OBTAINED BY PMP LAYER

We define the 2nd moment of PMP incorporating the steps of the 2nd moment of averaging message passing as in Appendix A.4, and define an additional step as:

(c) Consider $|\mathcal{N}_v|$ as a value only dependent to $y$ and $t$, namely $|\mathcal{N}_{yt}|$.

Background of step (c) is that in practice, $|\mathcal{N}_v|$ can vary for each node, but they will follow a distribution determined by the node's label $y$ and time $t$. For simplicity in our discussion, we will use the expectation of these values within each community as in step (c).

This 2nd moment is calculated rigorously as shown in Appendix A.4.

$$
\hat{\mathrm{var}}(M_v^{pmp(k+1)}) = \frac{\sum_{\tilde{y}\in\mathbf{Y}}\left(\sum_{\tilde{t}\in\mathbf{T}_t^{\text{single}}}4\mathcal{P}_{yt}\left(\tilde{y},\tilde{t}\right) + \sum_{\tilde{t}\in\mathbf{T}_t^{\text{double}}}\mathcal{P}_{yt}\left(\tilde{y},\tilde{t}\right)\right)\Sigma_X^{pmp(k)}(\tilde{y})}{\left(\sum_{\tilde{y}\in\mathbf{Y}}\sum_{\tilde{t}\in\mathbf{T}_t^{\text{single}}}2\mathcal{P}_{yt}\left(\tilde{y},\tilde{t}\right) + \sum_{\tilde{y}\in\mathbf{Y}}\sum_{\tilde{t}\in\mathbf{T}_t^{\text{double}}}\mathcal{P}_{yt}\left(\tilde{y},\tilde{t}\right)\right)^2 |\mathcal{N}_{yt}|} \tag{57}
$$

Therefore, we can write $\hat{\mathrm{var}}(M_v^{pmp(k+1)})=\Sigma_M^{pmp(k+1)}(y,t)$.

## A.6 EXPLANATION OF MMP

### A.6.1 1ST MOMENT OF AGGREGATED MESSAGE OBTAINED BY MMP LAYER.

We define the 1st moment of MMP with the identical steps of the 1st moment of averaging message passing, as in Appendix A.4.

### A.6.2 PROOF OF THEOREM 4.2

Suppose that the 1st moment of the representations from the previous layer is invariant. In other words, $\mu_X^{(k)}(y,t) = \mu_X^{(k)}(y,t_{max})$, $\forall t \in \mathbf{T}$. The message passing mechanism of PMP can be expressed as follows:

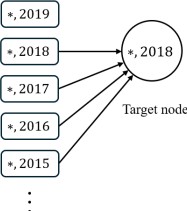

Figure 8: Graphical explanation of Mono-directional Message Passing(MMP).

$$
M_v^{mmp(k+1)} = \frac{\sum_{\tilde{y}\in\mathbf{Y}}\sum_{\tilde{t}\leq t}\sum_{v\in\mathcal{N}_v(\tilde{y},\tilde{t})}X_w}{\sum_{\tilde{y}\in\mathbf{Y}}\sum_{\tilde{t}\leq t}|\mathcal{N}_v(\tilde{y},\tilde{t})|} \tag{58}
$$

Applying assumption 3 as in PMP, the expectation is as follows. This expectation is calculated rigorously as shown in Appendix A.4.

$$
\hat{\mathbb{E}}\left[M_v^{mmp(k+1)}\right] = \frac{\sum_{\tilde{y}\in\mathbf{Y}}\sum_{\tilde{t}\leq t}\mathcal{P}_{yt}(\tilde{y},\tilde{t})\mu_X^{(k)}(\tilde{y})}{\sum_{\tilde{y}\in\mathbf{Y}}\sum_{\tilde{t}\leq t}\mathcal{P}_{yt}(\tilde{y},\tilde{t})} = \frac{\sum_{\tilde{y}\in\mathbf{Y}}\sum_{\tau\geq 0}g(y,\tilde{y},\tau)\mu_X^{(k)}(\tilde{y})}{\sum_{\tilde{y}\in\mathbf{Y}}\sum_{\tau\geq 0}g(y,\tilde{y},\tau)} \tag{59}
$$

This also lacks the $t$ term, thus it is invariant.

## A.7 MATHEMATICAL MODELING OF PMP.

Let $\mathcal{M}^{(k)}$ as the space of messages at $k$-th layer, and $\mathcal{X}^{(k)}$ as the space of representations at $k$-th layer, and let us define probability measure spaces $(\mathcal{M}^{(k)}, \sum_{\mathcal{M}^{(k)}}, m_{yt}^{(k)})$, $(\mathcal{X}^{(k)}, \sum_{\mathcal{X}^{(k)}}, x_{yt}^{(k)})$ where $\sum_{\mathcal{M}^{(k)}}$ and $\sum_{\mathcal{X}^{(k)}}$ are $\sigma$-algebras with probability measures $m_{yt}^{(k)}$ and $x_{yt}^{(k)}$, respectively.

That is, $m_{yt}^{(k)}$ is the probability measure of the message of node $v$ with label $y$ and time $t$, and $x_{yt}^{(k)}$ is the probability measure of the representation of node $v$ with label $y$ and time $t$, as defined in the main body. We are assuming that there is a "true" distribution for nodes with the same label and time. In other words, for node $v$ with label $y$ and time $t$, the assumption of this theoretical analysis is that $X_v^{(k)} \sim x_{yt}^{(k)}$ and $M_v^{(k)} \sim m_{yt}^{(k)}$.

### A.7.1 $m_{yt}^{(k)}$ TO $x_{yt}^{(k)}$

$f^{(k)}$ is the function which transfers the message $M_v^{(k)} \in \mathcal{M}^{(k)}$ to the $k$-th layer representation $X_v^{(k)} \in \mathcal{X}^{(k)}$.

Hence, $f^{(k)} : \mathcal{M}^{(k)} \to \mathcal{X}^{(k)}$ gives a pushforward of measure as $x_{yt}^{(k)} = (f_*^{(k)})(m_{yt}^{(k)}) : \sum_{\mathcal{X}^{(k)}} \to [0,1]$, given by $\left( (f_*^{(k)})(m_{yt}^{(k)}) \right)(B) = m_{yt}^{(k)} \left( (f^{(k)})^{-1}(B) \right)$, for $\forall B \in \sum_{\mathcal{X}^{(k)}}$

Here, we assume $f^{(k)}$ is G-Lipschitz for $\forall k \in \{1, 2, \ldots, K\}$.

### A.7.2 $x_{yt}^{(k)}$ TO $m_{yt}^{(k+1)}$

This is given as the message passing function of PMP. That is,

$$m_{yt}^{(k+1)} = \frac{\sum_{\tilde{y} \in \mathbf{Y}} \sum_{\tilde{t} \in \mathbf{T}_t^{single}} 2\mathcal{P}_{yt}(\tilde{y}, \tilde{t}) x_{\tilde{y}\tilde{t}}^{(k)} + \sum_{\tilde{y} \in \mathbf{Y}} \sum_{\tilde{t} \in \mathbf{T}_t^{double}} \mathcal{P}_{yt}(\tilde{y}, \tilde{t}) x_{\tilde{y}\tilde{t}}^{(k)}}{\sum_{\tilde{y} \in \mathbf{Y}} \sum_{\tilde{t} \in \mathbf{T}_t^{single}} 2\mathcal{P}_{yt}(\tilde{y}, \tilde{t}) + \sum_{\tilde{y} \in \mathbf{Y}} \sum_{\tilde{t} \in \mathbf{T}_t^{double}} \mathcal{P}_{yt}(\tilde{y}, \tilde{t})} \tag{60}$$

## A.8 THEORETICAL ANALYSIS OF PMP WHEN APPLIED IN MULTI-LAYER GNNs.

### A.8.1 LEMMAS

**Lemma 2.**
$$\forall \epsilon > 0, P(|M_v^{(k)} - M_{v'}^{(k)}| > \epsilon) \leq \frac{8V}{\epsilon^2} \text{ for } M_v^{(k)} \sim m_{yt}^{(k)}, M_{v'}^{(k)} \sim m_{yt'}^{(k)} \tag{61}$$

*Proof.* By chebyshev's inequality,

$P(|M_v^{(k)} - \mu_M^{(k)}(y)| > \frac{\epsilon}{2}) \leq \frac{4V}{\epsilon^2}, P(|M_{v'}^{(k)} - \mu_M^{(k)}(y)| > \frac{\epsilon}{2}) \leq \frac{4V}{\epsilon^2}$.

Therefore,

$$P(|M_v^{(k)} - M_{v'}^{(k)}| > \epsilon) \tag{62}$$
$$\leq P(|M_v^{(k)} - \mu_M(y)| + |M_{v'}^{(k)} - \mu_M(y)| > \epsilon) \because \text{Triangle inequality} \tag{63}$$
$$\leq P(|M_v^{(k)} - \mu_M(y)| > \frac{\epsilon}{2} \text{ or } |M_{v'}^{(k)} - \mu_M(y)| > \frac{\epsilon}{2}) \tag{64}$$
$$\leq P(|M_v^{(k)} - \mu_M(y)| > \frac{\epsilon}{2}) + P(|M_{v'}^{(k)} - \mu_M(y)| > \frac{\epsilon}{2}) \leq \frac{8V}{\epsilon^2} \tag{65}$$
$$\square$$

**Lemma 3.**
$$W_1(x_{yt}^{(k)}, x_{yt'}^{(k)}) \leq G \, W_1(m_{yt}^{(k)}, m_{yt'}^{(k)}) \tag{66}$$

*Proof.* Follows directly from G-Lipshitz property of $f^{(k)}$ and definition of pushforward measures.
$$\square$$

**Lemma 4.** $\mu_1, \ldots, \mu_n$ *are distributions with cumulative distribution functions* $F_1, \ldots, F_n$. *If* $W_1(\mu_i, \mu_j) \leq D, \ \forall i, j,$

*For arbitrary real numbers satisfying* $0 < \eta_i, \nu_i < S, \ s.t. \ \eta_1 + \cdots + \eta_n = \nu_1 + \cdots + \nu_n = S,$

$$W_1(\eta_1 \mu_1 + \cdots + \eta_n \mu_n, \nu_1 \mu_1 + \cdots + \nu_n \mu_n) < (S - \delta)D \tag{67}$$

*for some positive real number* $\delta$.

*Proof.*

$$\int_{\mathbb{R}} \Big| \sum_{i=1}^{n} (\eta_i - \nu_i) F_i(x) \Big| dx \tag{68}$$

$$= \int_{\mathbb{R}} \Big| \sum_{i=1}^{n} \delta_i F_i(x) \Big| dx, \text{ where } \delta_i = \eta_i - \nu_i \tag{69}$$

$$= \int_{\mathbb{R}} \Big| \sum_{\{i | \delta_i \geq 0\}} \delta_i F_i(x) + \sum_{\{j | \delta_j < 0\}} \delta_j F_j(x) \Big| dx \tag{70}$$

$$= \int_{\mathbb{R}} \Big| \sum_{\{i | \delta_i \geq 0\}} \delta_i \big( \delta_{i,1}(F_i(x) - F_{i,1}(x)) + \cdots + \delta_{i,n(i)}(F_i(x) - F_{i,n(i)}(x)) \big) \Big| dx \tag{71}$$

for some $\delta_{i,1}, \ldots, \delta_{i,n(i)} > 0, \ s.t. \ \delta_{i,1} + \cdots + \delta_{i,n(i)} = 1$.

$$\int_{\mathbb{R}} \Big| \sum_{\{i | \delta_i \geq 0\}} \delta_i \big( \delta_{i,1}(F_i(x) - F_{i,1}(x)) + \cdots + \delta_{i,n(i)}(F_i(x) - F_{i,n(i)}(x)) \big) \Big| dx \tag{72}$$

$$\leq \int_{\mathbb{R}} \sum_{\{i | \delta_i \geq 0\}} \delta_i \big( \delta_{i,1}|F_i(x) - F_{i,1}(x)| + \cdots + \delta_{i,n(i)}|F_i(x) - F_{i,n(i)}(x)| \big) dx \tag{73}$$

$$\leq \sum_{\{i | \delta_i \geq 0\}} \delta_i \big( \delta_{i,1} + \cdots + \delta_{i,n(i)} \big) D \tag{74}$$

$$= \sum_{\{i | \delta_i \geq 0\}} \delta_i D \tag{75}$$

$$= \sum_{\{i | \eta_i - \nu_i \geq 0\}} (\eta_i - \nu_i) D \tag{76}$$

$$< \sum_{\{i | \eta_i - \nu_i \geq 0\}} (\eta_i) D \tag{77}$$

$$< SD \tag{78}$$

$\square$

### A.8.2 PROOF OF THEOREM 4.3.

$$\mathbb{E}[|M_v^{(k)} - M_{v'}^{(k)}|] = \mathbb{E}\left[ |M_v^{(k)} - M_{v'}^{(k)}| \mathbb{1}_{\{|M_v^{(k)} - M_{v'}^{(k)}| \leq \epsilon\}} \right] + \mathbb{E}\left[ |M_v^{(k)} - M_{v'}^{(k)}| \mathbb{1}_{\{|M_v^{(k)} - M_{v'}^{(k)}| > \epsilon\}} \right] \leq \epsilon + \frac{16CV}{\epsilon^2} \tag{79}$$

since $\mathbb{E}\left[ |M_v^{(k)} - M_{v'}^{(k)}| \mathbb{1}\{|M_v^{(k)} - M_{v'}^{(k)}| \leq \epsilon\} \right] \leq \epsilon$, and

$\mathbb{E}\left[ |M_v^{(k)} - M_{v'}^{(k)}| \mathbb{1}\{|M_v^{(k)} - M_{v'}^{(k)}| > \epsilon\} \right] \leq 2C \ P(|M_v^{(k)} - M_{v'}^{(k)}| > \epsilon) \leq \frac{16CV}{\epsilon^2}$ by Lemma 2.

Plugging in $2(4CV)^{1/3}$ to $\epsilon$ gives us, $\mathbb{E}[|M_v^{(k)} - M_{v'}^{(k)}|] \leq 3(4CV)^{1/3}$.

$$\therefore W_1(m_{yt}^{(k)}, m_{yt'}^{(k)}) \leq \mathbb{E}[|M_v^{(k)} - M_{v'}^{(k)}|] \leq \mathcal{O}(C^{1/3}V^{1/3}) \tag{80}$$

### A.8.3 PROOF OF THEOREM 4.4

By Hoeffding's inequality, $P(|M_v^{(k)} - M_{v'}^{(k)}| > \frac{\epsilon}{2}) \leq 2\exp(-\frac{\epsilon^2}{8\tau^2})$.

So with the same steps of Theorem 4.3, $\mathbb{E}[|M_v^{(k)} - M_{v'}^{(k)}|] \leq \epsilon + 4C\exp(-\frac{\epsilon^2}{8\tau^2})$.

Plug in $(8\tau^2 \log C)^{1/2}$ to $\epsilon$. Then $\mathbb{E}[|M_v^{(k)} - M_{v'}^{(k)}|] \leq (8\tau^2 \log C)^{1/2} + 4$.

$$\therefore W_1(m_{yt}^{(k)}, m_{yt'}^{(k)}) \leq \mathcal{O}(\tau\sqrt{\log C}) \tag{81}$$

### A.8.4 PROOF OF THEOREM 4.5

$$m_{yt}^{(k+1)} = \frac{\sum_{\tilde{y}\in\mathbf{Y}}\sum_{\tilde{t}\in\mathbf{T}_t^{single}} 2\mathcal{P}_{yt}(\tilde{y},\tilde{t})x_{\tilde{y}\tilde{t}}^{(k)} + \sum_{\tilde{y}\in\mathbf{Y}}\sum_{\tilde{t}\in\mathbf{T}_t^{double}} \mathcal{P}_{yt}(\tilde{y},\tilde{t})x_{\tilde{y}\tilde{t}}^{(k)}}{\sum_{\tilde{y}\in\mathbf{Y}}\sum_{\tilde{t}\in\mathbf{T}_t^{single}} 2\mathcal{P}_{yt}(\tilde{y},\tilde{t}) + \sum_{\tilde{y}\in\mathbf{Y}}\sum_{\tilde{t}\in\mathbf{T}_t^{double}} \mathcal{P}_{yt}(\tilde{y},\tilde{t})} \tag{82}$$

$$= \frac{\sum_{\tilde{y}\in\mathbf{Y}}\sum_{\tilde{t}\in\mathbf{T}_t^{single}} 2f(y,t)g(y,\tilde{y},|\tilde{t}-t|)x_{\tilde{y}\tilde{t}}^{(k)} + \sum_{\tilde{y}\in\mathbf{Y}}\sum_{\tilde{t}\in\mathbf{T}_t^{double}} f(y,t)g(y,\tilde{y},|\tilde{t}-t|)x_{\tilde{y}\tilde{t}}^{(k)}}{\sum_{\tilde{y}\in\mathbf{Y}}\sum_{\tilde{t}\in\mathbf{T}_t^{single}} 2f(y,t)g(y,\tilde{y},|\tilde{t}-t|) + \sum_{\tilde{y}\in\mathbf{Y}}\sum_{\tilde{t}\in\mathbf{T}_t^{double}} f(y,t)g(y,\tilde{y},|\tilde{t}-t|)} \tag{83}$$

$$= \frac{\sum_{\tilde{y}\in\mathbf{Y}}\sum_{\tilde{t}\in\mathbf{T}_t^{single}} 2g(y,\tilde{y},|\tilde{t}-t|)x_{\tilde{y}\tilde{t}}^{(k)} + \sum_{\tilde{y}\in\mathbf{Y}}\sum_{\tilde{t}\in\mathbf{T}_t^{double}} g(y,\tilde{y},|\tilde{t}-t|)x_{\tilde{y}\tilde{t}}^{(k)}}{\sum_{\tilde{y}\in\mathbf{Y}}\sum_{\tilde{t}\in\mathbf{T}_t^{single}} 2g(y,\tilde{y},|\tilde{t}-t|) + \sum_{\tilde{y}\in\mathbf{Y}}\sum_{\tilde{t}\in\mathbf{T}_t^{double}} g(y,\tilde{y},|\tilde{t}-t|)} \tag{84}$$

$$\overset{let}{=} \sum_{\tilde{y}\in\mathbf{Y}}\sum_{\tilde{t}\in\mathbf{T}} \lambda_{yt\tilde{y}\tilde{t}}x_{\tilde{y}\tilde{t}}^{(k)} \tag{85}$$

Where $0 < \lambda_{yt\tilde{y}\tilde{t}} < 1$ is effective message passing weight in PMP, hence satisfying $\sum_{\tilde{y}\in\mathbf{Y}}\sum_{\tilde{t}\in\mathbf{T}} \lambda_{yt\tilde{y}\tilde{t}} = 1$.

Furthermore, since $\sum_{\tilde{t}\in\mathbf{T}_t^{single}} 2g(y,\tilde{y},|\tilde{t}-t|) + \sum_{\tilde{t}\in\mathbf{T}_t^{double}} g(y,\tilde{y},|\tilde{t}-t|) = 2\sum_{\tau\leq 0} g(y,\tilde{y},\tau)$, the following relation holds:

$$\sum_{\tilde{t}\in\mathbf{T}} \lambda_{yt\tilde{y}\tilde{t}} = \frac{\sum_{\tau\geq 0} g(y,\tilde{y},\tau)}{\sum_{y'\in\mathbf{Y}}\sum_{\tau\geq 0} g(y,y',\tau)} \tag{86}$$

Thus, $\sum_{\tilde{t}\in\mathbf{T}} \lambda_{yt\tilde{y}\tilde{t}} = \sum_{\tilde{t}\in\mathbf{T}} \lambda_{yt'\tilde{y}\tilde{t}}, \forall t,t' \in \mathbf{T}$. We can let $\sum_{\tilde{t}\in\mathbf{T}} \lambda_{yt\tilde{y}\tilde{t}} = \rho_{y\tilde{y}}$.

$$W_1(m_{yt}^{(k+1)}, m_{yt_{max}}^{(k+1)}) = W_1\left(\sum_{\tilde{y}\in\mathbf{Y}}\sum_{\tilde{t}\in\mathbf{T}} \lambda_{yt\tilde{y}\tilde{t}}x_{\tilde{y}\tilde{t}}^{(k)}, \sum_{\tilde{y}\in\mathbf{Y}}\sum_{\tilde{t}\in\mathbf{T}} \lambda_{yt'\tilde{y}\tilde{t}}x_{\tilde{y}\tilde{t}}^{(k)}\right) \tag{87}$$

$$= \int_{\mathbb{R}} \left| \sum_{\tilde{y}\in\mathbf{Y}}\sum_{\tilde{t}\in\mathbf{T}} \lambda_{yt\tilde{y}\tilde{t}}F_{\tilde{y}\tilde{t}}^{(k)}(x) - \sum_{\tilde{y}\in\mathbf{Y}}\sum_{\tilde{t}\in\mathbf{T}} \lambda_{yt'\tilde{y}\tilde{t}}F_{\tilde{y}\tilde{t}}^{(k)}(x)\right| dx \tag{88}$$

$$= \int_{\mathbb{R}} \left| \sum_{\tilde{y}\in\mathbf{Y}}\sum_{\tilde{t}\in\mathbf{T}} (\lambda_{yt\tilde{y}\tilde{t}} - \lambda_{yt'\tilde{y}\tilde{t}})F_{\tilde{y}\tilde{t}}^{(k)}(x)\right| dx \tag{89}$$

Where $F_{\tilde{y}\tilde{t}}^{(k)}$ is the cumulative distribution function of $x_{\tilde{y}\tilde{t}}^{(k)}$.

By Lemma 3 and Lemma 4,

$$\int_{\mathbb{R}} \left| \sum_{\tilde{t}\in\mathbf{T}} (\lambda_{yt\tilde{y}\tilde{t}} - \lambda_{yt'\tilde{y}\tilde{t}})F_{\tilde{y}\tilde{t}}^{(k)}(x)\right| dx \leq (\rho_{y\tilde{y}} - \epsilon_{y\tilde{y}tt'})GW \tag{90}$$

For some $0 < \epsilon_{y\tilde{y}tt'} < \rho_{y\tilde{y}}$.

$$\therefore W_1(m_{yt}^{(k+1)}, m_{yt_{max}}^{(k+1)}) \leq \int_{\mathbb{R}} \sum_{\tilde{y} \in \mathbf{Y}} \sum_{\tilde{t} \in \mathbf{T}} \left| (\lambda_{yt\tilde{y}\tilde{t}} - \lambda_{yt'\tilde{y}\tilde{t}}) F_{\tilde{y}\tilde{t}}^{(k)}(x) \right| dx \tag{91}$$

$$\leq \sum_{\tilde{y} \in \mathbf{Y}} (\rho_{y\tilde{y}} - \epsilon_{y\tilde{y}tt'}) GW \tag{92}$$

$$= G(1 - \sum_{\tilde{y} \in \mathbf{Y}} \epsilon_{y\tilde{y}tt'}) W \tag{93}$$

Let $\epsilon_{ytt'} = \sum_{\tilde{y} \in \mathbf{Y}} \epsilon_{y\tilde{y}tt'}$ and $\min_{y \in \mathbf{Y}, t, t' \in \mathbf{T}} \epsilon_{ytt'} = \epsilon$.

Then, $W_1(m_{yt}^{(k+1)}, m_{yt'}^{(k+1)}) \leq G(1 - \epsilon) W$.

Let $G^{(k)} = \frac{1}{1-\epsilon} > 1$.

Then, $\forall y, t, t', \ W_1(m_{yt}^{(k+1)}, m_{yt_{max}}^{(k+1)}) \leq \frac{G}{G^{(k)}} W$

## A.9 ESTIMATION OF RELATIVE CONNECTIVITY

When $t \neq t_{max}$ and $\tilde{t} \neq t_{max}$, $\mathcal{P}_{yt}(\tilde{y}, \tilde{t})$ has the following best unbiased estimator:

$$\hat{\mathcal{P}}_{yt}(\tilde{y}, \tilde{t}) = \frac{\sum_{u \in \{u' \in \mathbf{V} | u' \text{ has label } y, u' \text{ has time } t\}} |\mathcal{N}_u(\tilde{y}, \tilde{t})|}{\sum_{u \in \{u' \in \mathbf{V} | u' \text{ has label } y, u' \text{ has time } t\}} |\mathcal{N}_u|}, \ \forall t, \tilde{t} \neq t_{max} \tag{94}$$

We can regard this problem as a nonlinear overdetermined system $\hat{\mathcal{P}}_{yt}(\tilde{y}, \tilde{t}) = f(y, t)g(y, \tilde{y}, |\tilde{t} - t|), \ \forall y, \tilde{y} \in \mathbf{Y}, \forall t, \tilde{t} \in \mathbf{T}$, with the constraint of $\sum_{\tilde{y} \in \mathbf{Y}} \sum_{\tilde{t} \in \mathbf{T}} \hat{\mathcal{P}}_{yt}(\tilde{y}, \tilde{t}) = 1$.

When $t = t_{max}$ or $\tilde{t} = t_{max}$ is not feasible due to the unavailability of labels in the test set, we utilize assumption 3 to compute $\hat{\mathcal{P}}_{yt}(\tilde{y}, \tilde{t})$ for this cases. Let's first consider the following equation:

$$\sum_{\tilde{y} \in \mathbf{Y}} \mathcal{P}_{yt}(\tilde{y}, t) = \sum_{\tilde{y} \in \mathbf{Y}} f(y, t)g(y, \tilde{y}, 0) = f(y, t) \sum_{\tilde{y} \in \mathbf{Y}} g(y, \tilde{y}, 0) \tag{95}$$

Earlier, when introducing assumption 3, we defined $\sum_{\tilde{y} \in \mathbf{Y}} g(y, \tilde{y}, 0) = 1$. Therefore, when $t < t_{max}$, we can express $f(y, t)$ as follows:

$$f(y, t) = \sum_{\tilde{y} \in \mathbf{Y}} \mathcal{P}_{yt}(\tilde{y}, t) \tag{96}$$

For any $\Delta \in \{|\tilde{t} - t| \mid t, \tilde{t} \in \mathbf{T}\}$, we have:

$$\sum_{t < t_{max} - \Delta} \mathcal{P}_{yt}(\tilde{y}, t + \Delta) = \sum_{t < t_{max} - \Delta} f(y, t)g(y, \tilde{y}, \Delta) \tag{97}$$

$$\sum_{t < t_{max}} \mathcal{P}_{yt}(\tilde{y}, t - \Delta) = \sum_{t < t_{max}} f(y, t)g(y, \tilde{y}, \Delta) \tag{98}$$

The reason we consider up to $t = t_{max} - 1 - \Delta$ in the first equation and up to $t = t_{max} - 1$ in the second equation is because we assume situations where $\mathcal{P}_{yt}(\tilde{y}, \tilde{t})$ cannot be estimated when $t = t_{max}$ or $\tilde{t} = t_{max}$. Utilizing both equations aims to construct an estimator using as many measured values as possible when $t \neq t_{max}$.

Thus,

$$g(y, \tilde{y}, \Delta) = \frac{\sum_{t < t_{max} - \Delta} \mathcal{P}_{yt}(\tilde{y}, t + \Delta) + \sum_{t < t_{max}} \mathcal{P}_{yt}(\tilde{y}, t - \Delta)}{\sum_{t < t_{max} - \Delta} f(y, t) + \sum_{t < t_{max}} f(y, t)} \tag{99}$$

Since $f(y, t) = \sum_{\tilde{y} \in \mathbf{Y}} \mathcal{P}_{yt}(\tilde{y}, t)$,

$$g(y, \tilde{y}, \Delta) = \frac{\sum_{t < t_{max} - \Delta} \mathcal{P}_{yt}(\tilde{y}, t + \Delta) + \sum_{t < t_{max}} \mathcal{P}_{yt}(\tilde{y}, t - \Delta)}{\sum_{t < t_{max} - \Delta} \sum_{y' \in \mathbf{Y}} \mathcal{P}_{yt}(y', t) + \sum_{t < t_{max}} \sum_{y' \in \mathbf{Y}} \mathcal{P}_{yt}(y', t)} \quad (100)$$

For any $y, \tilde{y} \in \mathbf{Y}$ and $\Delta \in \{|\tilde{t} - t| \mid t, \tilde{t} \in \mathbf{T}\}$, we can construct an estimator $\hat{g}(y, \tilde{y}, \Delta)$ for $g(y, \tilde{y}, \Delta)$ as follows:

$$\hat{g}(y, \tilde{y}, \Delta) = \frac{\sum_{t < t_{max} - \Delta} \hat{\mathcal{P}}_{yt}(\tilde{y}, t + \Delta) + \sum_{t < t_{max}} \hat{\mathcal{P}}_{yt}(\tilde{y}, t - \Delta)}{\sum_{t < t_{max} - \Delta} \sum_{y' \in \mathbf{Y}} \hat{\mathcal{P}}_{yt}(y', t) + \sum_{t < t_{max}} \sum_{y' \in \mathbf{Y}} \hat{\mathcal{P}}_{yt}(y', t)} \quad (101)$$

This estimator is designed to utilize as many measured values $\hat{\mathcal{P}}_{yt}(\tilde{y}, \tilde{t})$ as possible, excluding cases where $t = t_{max}$ or $\tilde{t} = t_{max}$.

$$\mathcal{P}_{yt}(\tilde{y}, \tilde{t}) = \frac{\mathcal{P}_{yt}(\tilde{y}, \tilde{t})}{\sum_{y' \in \mathbf{Y}} \sum_{t' \in \mathbf{T}} \mathcal{P}_{yt}(y', t')} = \frac{g(y, \tilde{y}, |\tilde{t} - t|)}{\sum_{y' \in \mathbf{Y}} \sum_{t' \in \mathbf{T}} g(y, y', |t' - t|)} \quad (102)$$

Therefore, for all $y, \tilde{y} \in \mathbf{Y}$ and $|\tilde{t} - t| \in \{|\tilde{t} - t| \mid t, \tilde{t} \in \mathbf{T}\}$, we can define the estimator $\hat{\mathcal{P}}_{yt}(\tilde{y}, \tilde{t})$ of $\mathcal{P}_{yt}(\tilde{y}, \tilde{t})$ as follows:

$$\hat{\mathcal{P}}_{yt}(\tilde{y}, \tilde{t}) = \frac{\hat{g}(y, \tilde{y}, |\tilde{t} - t|)}{\sum_{y' \in \mathbf{Y}} \sum_{t' \in \mathbf{T}} \hat{g}(y, y', |t' - t|)} \quad (103)$$

## A.10 EXPLANATION OF PNY

### A.10.1 1ST AND 2ND MOMENT OF AGGREGATED MESSAGE OBTAINED THROUGH PNY TRANSFORM.

We define the 1st and 2nd moment of PNY with the identical steps of the 1st and 2nd moment of averaging message passing, as in Appendix A.4.2.

### A.10.2 PROOF OF THEOREM 5.1

$$\hat{\mathbb{E}}[M_v^{PNY(k+1)}] \stackrel{(a)}{=} \mathbb{E}[A_t(M_v^{pmp(k+1)} - \mu_M^{pmp(k+1)})] + \mathbb{E}[M_v^{pmp(k+1)}] \quad (104)$$

$$= A_t(\mathbb{E}[M_v^{pmp(k+1)}] - \mu_M^{pmp(k+1)}) + \mu_M^{pmp(k+1)} \quad (105)$$

$$\stackrel{(b)}{=} A_t(\mu_M^{pmp(k+1)} - \mu_M^{pmp(k+1)}) + \mu_M^{pmp(k+1)} \quad (106)$$

$$= \mu_M^{pmp(k+1)} \quad (107)$$

---

**Algorithm 3:** Persistent Message PassingEstimation of relative connectivity.

---

**Input** : Neighboring node sets $\mathcal{N}_u$, $\forall u \in \mathbf{V}$; node time function $time : V \to \mathbf{T}$; train, test split $V^{tr} = \{v \mid v \in V, time(v) < t_{\max}\}$ and $V^{te} = \{v \mid v \in \mathbf{V}, time(v) = t_{\max}\}$; node label function $label : \mathbf{V}^{tr} \to \mathbf{Y}$.

**Output:** Estimated relative connectivity $\hat{\mathcal{P}}_{y,t}(\tilde{y},\tilde{t})$, $\forall y, \tilde{y} \in \mathbf{Y}$, $t, \tilde{t} \in \mathbf{T}$.

---

1 **Estimate $\hat{\mathcal{P}}_{y,t}(\tilde{y},\tilde{t})$ when $t \neq t_{\max}$ and $\tilde{t} \neq t_{\max}$.**

2 **for** $t \in \mathbf{T} \setminus \{t_{\max}\}$ **do**

3     **for** $\tilde{t} \in \mathbf{T} \setminus \{t_{\max}\}$ **do**

4        $\hat{\mathcal{P}}_{y,t}(\tilde{y},\tilde{t}) \leftarrow \frac{\sum_{u \in \{v \in \mathbf{V} \mid v \text{ has label } y, v \text{ has time } t\}} |\{v \in \mathcal{N}_u \mid v \text{ has label } \tilde{y}, v \text{ has time } \tilde{t}\}|}{\sum_{u \in \{v \in \mathbf{V} \mid v \text{ has label } y, v \text{ has time } t\}} |\mathcal{N}_u|}$;

5     **end**

6 **end**

7 **Estimate $g$ function.**

8 **for** $y \in \mathbf{Y}$ **do**

9     **for** $\tilde{y} \in \mathbf{Y}$ **do**

10        **for** $\Delta \in \{|\tilde{t} - t| \mid t, \tilde{t} \in \mathbf{T}\}$ **do**

11           $\hat{g}(y,\tilde{y},\Delta) \leftarrow \frac{\sum_{t < t_{\max} - \Delta} \hat{\mathcal{P}}_{y,t}(\tilde{y}, t+\Delta) + \sum_{t < t_{\max}} \hat{\mathcal{P}}_{y,t}(\tilde{y}, t-\Delta)}{\sum_{t < t_{\max} - \Delta} \sum_{y' \in \mathbf{Y}} \hat{\mathcal{P}}_{y,t}(y', t) + \sum_{t < t_{\max}} \sum_{y' \in \mathbf{Y}} \hat{\mathcal{P}}_{y,t}(y', t)}$;

12        **end**

13     **end**

14 **end**

15 **Estimate $\hat{\mathcal{P}}_{y,t}(\tilde{y},\tilde{t})$ when $t = t_{\max}$ or $\tilde{t} = t_{\max}$.**

16 **for** $y \in \mathbf{Y}$ **do**

17     **for** $\tilde{y} \in \mathbf{Y}$ **do**

18        **for** $t \in \mathbf{T}$ **do**

19           $\hat{\mathcal{P}}_{y,t}(\tilde{y}, t_{\max}) \leftarrow \frac{\hat{g}(y,\tilde{y},|t_{\max}-t|)}{\sum_{y' \in \mathbf{Y}} \sum_{t' \in \mathbf{T}} \hat{g}(y,y',|t'-t|)}$;

20        **end**

21     **end**

22 **end**

23 **for** $y \in \mathbf{Y}$ **do**

24     **for** $\tilde{y} \in \mathbf{Y}$ **do**

25        **for** $\tilde{t} \in \mathbf{T}$ **do**

26           $\hat{\mathcal{P}}_{y,t_{\max}}(\tilde{y},\tilde{t}) \leftarrow \frac{\hat{g}(y,\tilde{y},|\tilde{t}-t_{\max}|)}{\sum_{y' \in \mathbf{Y}} \sum_{t' \in \mathbf{T}} \hat{g}(y,y',|t'-t_{\max}|)}$;

27        **end**

28     **end**

29 **end**

---

$$\hat{\text{var}}[M_v^{PNY(k+1)}] \stackrel{(a)}{=} \text{var}\left( A_t(M_v^{pmp(k+1)} - \mu_M^{pmp(k+1)}(y)) + \mu_M^{pmp(k+1)}(y) \right) \tag{108}$$

$$\stackrel{(b)}{=} \mathbb{E}[A_t(M_v^{pmp(k+1)} - \mu_M^{pmp(k+1)}(y))(M_v^{pmp(k+1)} - \mu_M^{pmp(k+1)}(y))^\top A_t^\top] \tag{109}$$

$$= A_t\mathbb{E}[(M_v^{pmp(k+1)} - \mu_M^{pmp(k+1)}(y))(M_v^{pmp(k+1)} - \mu_M^{pmp(k+1)}(y))^\top]A_t^\top \tag{110}$$

$$\stackrel{(b)}{=} A_t\hat{\text{var}}(M_v^{pmp(k+1)})A_t^\top \tag{111}$$

$$= (U_{yt_{max}}\Lambda_{yt_{max}}^{1/2}\Lambda_{yt}^{-1/2}U_{yt}^\top)\Sigma_M^{pmp(k+1)}(U_{yt}\Lambda_{yt}^{-1/2}\Lambda_{yt_{max}}^{1/2}U_{yt_{max}}^\top) \tag{112}$$

$$= (U_{yt_{max}}\Lambda_{yt_{max}}^{1/2}\Lambda_{yt}^{-1/2}U_{yt}^\top)(U_{yt}\Lambda_{yt}U_{yt}^{-1})(U_{yt}\Lambda_{yt}^{-1/2}\Lambda_{yt_{max}}^{1/2}U_{yt_{max}}^\top) \tag{113}$$

$$= (U_{yt_{max}}\Lambda_{yt_{max}}^{1/2}\Lambda_{yt}^{-1/2})\Lambda_{yt}(\Lambda_{yt}^{-1/2}\Lambda_{yt_{max}}^{1/2}U_{yt_{max}}^\top) \tag{114}$$

$$= (U_{yt_{max}}\Lambda_{yt_{max}}^{1/2})(\Lambda_{yt_{max}}^{1/2}U_{yt_{max}}^\top) \tag{115}$$

$$= U_{yt_{max}}\Lambda_{yt_{max}}U_{yt_{max}}^\top \tag{116}$$

$$= \Sigma_M^{pmp(k+1)}(y, t_{max}) \tag{117}$$

## A.11 EXPLANATION OF JJNORM

### A.11.1 1ST AND 2ND MOMENT OF AGGREGATED MESSAGE OBTAINED THROUGH JJNORM.

We define the first moment of JJNORM message as the following steps:

**1st moment of aggregated message obtained through JJNORM.**

(a) Take the expectation of the averaged message.

(b) Approximate the expectation of every PMP message to the 1st moment of PMP message.

$$\hat{\mathbb{E}}[M_v^{JJ}] \stackrel{(a)}{=} \mathbb{E}[\alpha_t(M_v^{pmp(K)} - \mu_M^{JJ}(y, t)) + \mu_M^{JJ}(y, t)] \tag{118}$$

$$= \alpha_t\mathbb{E}[M_v^{pmp(K)}] + (1 - \alpha_t)\mathbb{E}[\mu_M^{JJ}(y, t)] \tag{119}$$

$$= \alpha_t\mathbb{E}[M_v^{pmp(K)}] + (1 - \alpha_t)\frac{1}{|\mathbf{V}_{y,t}|}\mathbb{E}\left[\sum_{x \in \mathbf{V}_{y,t}} M_w^{pmp(K)}\right] \tag{120}$$

$$= \alpha_t\mathbb{E}[M^{pmp(K)}] + (1 - \alpha_t)\frac{1}{|\mathbf{V}_{y,t}|}\sum_{x \in \mathbf{V}_{y,t}}\mathbb{E}\left[M_w^{pmp(K)}\right] \tag{121}$$

$$\stackrel{(b)}{=} \alpha_t\mathbb{E}[M^{pmp(K)}] + (1 - \alpha_t)\frac{1}{|\mathbf{V}_{y,t}|}\sum_{x \in \mathbf{V}_{y,t}}\hat{\mathbb{E}}\left[M_w^{pmp(K)}\right] \tag{122}$$

$$= \alpha_t\mathbb{E}[M^{pmp(K)}] + (1 - \alpha_t)\frac{1}{|\mathbf{V}_{y,t}|}\sum_{w \in \mathbf{V}_{y,t}}\mu_M^{pmp(K)}(y) \tag{123}$$

$$= \alpha_t\mu_M^{pmp(K)}(y) + (1 - \alpha_t)\mu_M^{pmp(K)}(y) \tag{124}$$

$$= \mu_M^{pmp(K)}(y) \tag{125}$$

**2nd moment of aggregated message obtained through JJNORM.**

We define the second moment of the JJNORM message as the following steps:

(a) Take the variance of the averaged message.

(b) Consider $\mu_M^{JJ}(y, t)$ as a constant.

(c) Approximate the variance of the PMP message to the 2nd moment of the PMP message.

---

**Algorithm 4:** Persistent Message PassingPNY transformation

**Input** : Previous layer's representation $X_v, \forall v \in \mathbf{V}$; Aggregated message $M_v, \forall v \in \mathbf{V}$, obtained from 1st moment alignment message passing; node time function $time : \mathbf{V} \to \mathbf{T}$; train, test split $\mathbf{V}^{tr} = \{v \mid v \in \mathbf{V}, time(v) < t_{\max}\}$ and $\mathbf{V}^{te} = \{v \mid v \in \mathbf{V}, time(v) = t_{\max}\}$; node label funtion $label : \mathbf{V}^{tr} \to \mathbf{Y}$; Estimated relative connectivity $\hat{\mathcal{P}}_{y,t}(\tilde{y}, \tilde{t}), \forall y, \tilde{y} \in \mathbf{Y}, \ t, \tilde{t} \in \mathbf{T}$.

**Output:** Modified aggregated message $M'_v, \forall v \in \mathbf{V}$

**1** Let $\mathbf{V}_{y,t} = \{u \in \mathbf{V} \mid label(u) = y, time(u) = t\}$;
**2** Let $\mathbf{V}_{\cdot,t} = \{u \in \mathbf{V} \mid time(u) = t\}$;
**3** Let $\mathbf{T}_\tau^{\text{single}} = \{t \in \mathbf{T} \mid t = \tau \text{ or } t < 2\tau - t_{\max}\}$;
**4** Let $\mathbf{T}_\tau^{\text{double}} = \{t \in \mathbf{T} \mid |t - \tau| \le |t_{\max} - \tau|, t \ne \tau\}$;
**5** Let $|\mathcal{N}_{yt}| = \frac{1}{|\mathbf{V}_{y,t}|} \sum_{u \in \mathbf{V}_{y,t}} |\mathcal{N}_u|$;
**6** **Estimate covariance matrices of previous layer's representation.**
**7** **for** $t \in \mathbf{T}$ **do**
**8** $\quad \hat{\mu}_X(\cdot, t) \leftarrow \hat{\mu}_M(\cdot, t) = \frac{1}{|\mathbf{V}_{\cdot,t}|} \sum_{v \in \mathbf{V}_{\cdot,t}} X_v$;
**9** $\quad \hat{\Sigma}_{XX}(y) \leftarrow \frac{1}{|\mathbf{V}_{\cdot,t}|-1} \sum_{v \in \mathbf{V}_{\cdot,t}} (X_v - \hat{\mu}_X(\cdot, t))(X_v - \hat{\mu}_X(\cdot, t))^\top$;
**10** **end**
**11** **Estimate covariance matrices of aggregated message.**
**12** **for** $y \in \mathbf{Y}$ **do**
**13** $\quad$ **for** $t \in \mathbf{T}$ **do**
**14** $\qquad \hat{\Sigma}_{MM}(y, t) \leftarrow \dfrac{\sum_{\tilde{y} \in \mathbf{Y}} \left( \sum_{\tilde{t} \in \mathbf{T}_t^{\text{single}}} 4\hat{\mathcal{P}}_{y,t}(\tilde{y}, \tilde{t}) + \sum_{\tilde{t} \in \mathbf{T}_t^{\text{double}}} \hat{\mathcal{P}}_{y,t}(\tilde{y}, \tilde{t}) \right) \hat{\Sigma}_{XX}(\tilde{y})}{\left( \sum_{\tilde{y} \in \mathbf{Y}} \sum_{\tilde{t} \in \mathbf{T}_t^{\text{single}}} 2\hat{\mathcal{P}}_{y,t}(\tilde{y}, \tilde{t}) + \sum_{\tilde{y} \in \mathbf{Y}} \sum_{\tilde{t} \in \mathbf{T}_t^{\text{double}}} \hat{\mathcal{P}}_{y,t}(\tilde{y}, \tilde{t}) \right)^2 |\mathcal{N}_{yt}|}$;
**15** $\quad$ **end**
**16** **end**
**17** **Orthogonal diagonalization.**
**18** **for** $y \in \mathbf{Y}$ **do**
**19** $\quad$ **for** $t \in \mathbf{T}$ **do**
**20** $\qquad$ Find $\hat{P}_{y,t}, \ \hat{D}_{y,t}$ s.t. $\hat{\Sigma}_{MM}(y, t) = \hat{P}_{y,t} \hat{D}_{y,t} \hat{P}_{y,t}^{-1}$ and $\hat{P}_{y,t}^{-1} = \hat{P}_{y,t}^\top$;
**21** $\quad$ **end**
**22** **end**
**23** **Update aggregated message.**
**24** **for** $v \in \mathbf{V} \setminus \mathbf{V}_{\cdot, t_{\max}}$ **do**
**25** $\quad$ Let $y = label(v)$;
**26** $\quad$ Let $t = time(v)$;
**27** $\quad M'_v \leftarrow \hat{P}_{y,t_{\max}} \hat{D}_{y,t_{\max}}^{1/2} \hat{D}_{y,t}^{-1/2} \hat{P}_{y,t}^\top (M_v - \hat{\mu}_M(y)) + \hat{\mu}_M(y)$;
**28** **end**

---

$$\hat{\text{var}}(M_v^{JJ}) \overset{(a)}{=} \text{var}\left(\alpha_t(M_v^{pmp(k)} - \mu_M^{JJ}) + \mu_M^{JJ}(y,t)\right) \tag{126}$$

$$\overset{(b)}{=} \text{var}\left(\alpha_t M_v^{pmp(K)}\right) \tag{127}$$

$$= \alpha_t^2 \text{var}\left(M_v^{pmp(K)}\right) \tag{128}$$

$$\overset{(c)}{=} \alpha_t^2 \hat{\text{var}}\left(M_v^{pmp(K)}\right) \tag{129}$$

$$= \alpha_t^2 \Sigma_M^{pmp(K)}(y,t) \tag{130}$$

### A.11.2 Proof of Lemma 1

Consider GNNs with linear semantic aggregation functions.

$$M_v^{pmp(k+1)} \leftarrow \text{PMP}(X_w^{pmp(k)}, w \in \mathcal{N}_v) \tag{131}$$

$$X_v^{pmp(k+1)} \leftarrow A^{(k+1)} M_v^{pmp(k+1)}, \forall k < K, v \in \mathbf{V} \tag{132}$$

Let's use mathematical induction. First, for initial features, $\Sigma_X^{pmp(0)}(y, t_{max}) = \Sigma_X^{pmp(0)}(y, t)$ holds. Suppose that in the $k$-th layer, representation $X^{(k)}$ satisfies $\beta_t^{(k)} \Sigma_X^{pmp(k)}(y, t_{max}) = \Sigma_X^{pmp(k)}(y, t)$. This assumes that the expected covariance matrix of representations of nodes with identical labels but differing time information only differs by a constant factor.

$$\Sigma_M^{pmp(k+1)}(y,t) = \sum_{\tilde{y} \in \mathbf{Y}} \left( \sum_{\tilde{t} \in \mathbf{T}_t^{\text{single}}} 4\mathcal{P}_{yt}(\tilde{y}, \tilde{t}) + \sum_{\tilde{t} \in \mathbf{T}_t^{\text{double}}} \mathcal{P}_{yt}(\tilde{y}, \tilde{t}) \right) \Sigma_X^{pmp(k)}(\tilde{y}) \tag{133}$$

$$\left/ \left( \sum_{\tilde{y} \in \mathbf{Y}} \sum_{\tilde{t} \in \mathbf{T}_t^{\text{single}}} 2\mathcal{P}_{yt}(\tilde{y}, \tilde{t}) + \sum_{\tilde{y} \in \mathbf{Y}} \sum_{\tilde{t} \in \mathbf{T}_t^{\text{double}}} \mathcal{P}_{yt}(\tilde{y}, \tilde{t}) \right)^2 |\mathcal{N}_v| \right. \tag{134}$$

$$\Sigma_M^{pmp(k+1)}(y,t) = \frac{\sum_{\tilde{y} \in \mathbf{Y}} \left( \sum_{\tilde{t} \in \mathbf{T}_t^{\text{single}}} 4\mathcal{P}_{yt}(\tilde{y}, \tilde{t}) \Sigma_X^{pmp(k)}(\tilde{y}, \tilde{t}) + \sum_{\tilde{t} \in \mathbf{T}_t^{\text{double}}} \mathcal{P}_{yt}(\tilde{y}, \tilde{t}) \Sigma_X^{pmp(k)}(\tilde{y}, \tilde{t}) \right)}{\left( \sum_{\tilde{y} \in \mathbf{Y}} \left( \sum_{\tilde{t} \in \mathbf{T}_t^{\text{single}}} 2\mathcal{P}_{yt}(\tilde{y}, \tilde{t}) + \sum_{\tilde{t} \in \mathbf{T}_t^{\text{double}}} \mathcal{P}_{yt}(\tilde{y}, \tilde{t}) \right) \right)^2 |\mathcal{N}_{yt}|} \tag{135}$$

$$= \frac{\sum_{\tilde{y} \in \mathbf{Y}} \left( \sum_{\tilde{t} \in \mathbf{T}_t^{\text{single}}} 4\mathcal{P}_{yt}(\tilde{y}, \tilde{t}) \beta_{\tilde{t}}^{(k)} + \sum_{\tilde{t} \in \mathbf{T}_t^{\text{double}}} \mathcal{P}_{yt}(\tilde{y}, \tilde{t}) \beta_{\tilde{t}}^{(k)} \right) \Sigma_X^{pmp(k)}(y, t_{max})}{\left( \sum_{\tilde{y} \in \mathbf{Y}} \left( \sum_{\tilde{t} \in \mathbf{T}_t^{\text{single}}} 2\mathcal{P}_{yt}(\tilde{y}, \tilde{t}) + \sum_{\tilde{t} \in \mathbf{T}_t^{\text{double}}} \mathcal{P}_{yt}(\tilde{y}, \tilde{t}) \right) \right)^2 |\mathcal{N}_{yt}|} \tag{136}$$

$$\frac{\sum_{\tilde{t} \in \mathbf{T}_t^{\text{single}}} 4\mathcal{P}_{yt}(\tilde{y}, \tilde{t}) \beta_{\tilde{t}}^{(k)} + \sum_{\tilde{t} \in \mathbf{T}_t^{\text{double}}} \mathcal{P}_{yt}(\tilde{y}, \tilde{t}) \beta_{\tilde{t}}^{(k)}}{\sum_{\tilde{t} \in \mathbf{T}} 4\mathcal{P}_{yt_{max}}(\tilde{y}, \tilde{t}) \beta_{\tilde{t}}^{(k)}} \tag{137}$$

$$= \frac{\sum_{\tilde{t} \in \mathbf{T}_t^{\text{single}}} 4g(y, \tilde{y}, |\tilde{t} - t|) \beta_{\tilde{t}}^{(k)} + \sum_{\tilde{t} \in \mathbf{T}_t^{\text{double}}} g(y, \tilde{y}, |\tilde{t} - t|) \beta_{\tilde{t}}^{(k)}}{\sum_{\tilde{t} \in \mathbf{T}} 4g(y, \tilde{y}, |\tilde{t} - t_{max}|) \beta_{\tilde{t}}^{(k)}} \tag{138}$$

Since it is unrelated to $y$ by Assumption 4, we can define it as $\gamma_t^{(k)}$.

$$\frac{\sqrt{|\mathcal{N}_{yt}|}}{\sqrt{|\mathcal{N}_{yt_{max}}|}} \frac{\sum_{\tilde{t}\in\mathbf{T}_t^{\text{single}}} 2\mathcal{P}_{yt}(\tilde{y},\tilde{t}) + \sum_{\tilde{t}\in\mathbf{T}_t^{\text{double}}} \mathcal{P}_{yt}(\tilde{y},\tilde{t})}{\sum_{\tilde{t}\in\mathbf{T}} 2\mathcal{P}_{yt_{max}}(\tilde{y},\tilde{t})} \tag{139}$$

$$\stackrel{(c)}{=} \frac{\sqrt{P(t)}}{\sqrt{P(t_{max})}} \frac{\sum_{\tilde{t}\in\mathbf{T}_t^{\text{single}}} 2\mathcal{P}_{yt}(\tilde{y},\tilde{t}) + \sum_{\tilde{t}\in\mathbf{T}_t^{\text{double}}} \mathcal{P}_{yt}(\tilde{y},\tilde{t})}{\sum_{\tilde{t}\in\mathbf{T}} 2\mathcal{P}_{yt_{max}}(\tilde{y},\tilde{t})} \tag{140}$$

$$= \frac{\sqrt{P(t)}}{\sqrt{P(t_{max})}} \frac{\sum_{\tilde{t}\in\mathbf{T}_t^{\text{single}}} 2g(y,\tilde{y},|\tilde{t}-t|) + \sum_{\tilde{t}\in\mathbf{T}_t^{\text{double}}} g(y,\tilde{y},|\tilde{t}-t|)}{\sum_{\tilde{t}\in\mathbf{T}} 2g(y,\tilde{y},|\tilde{t}-t_{max}|)(\tilde{y},\tilde{t})} \tag{141}$$

Since it is unrelated to $y$ by assumption 4, we can define it as $\lambda_t$.

1st equality holds by step (c) of 2nd moment of PMP, as defined in Appendix A.4.2

$$\Sigma_M^{pmp(k+1)}(y,t) = \frac{\gamma_t^{(k)}}{\lambda_t^2} \frac{\sum_{\tilde{y}\in\mathbf{Y}} \sum_{\tilde{t}\in\mathbf{T}} 4\mathcal{P}_{yt}(\tilde{y},\tilde{t})\beta_t^{(k)}\Sigma_X^{pmp(k)}(\tilde{y},t_{max})}{\left(\sum_{\tilde{y}\in\mathbf{Y}} \sum_{\tilde{t}\in\mathbf{T}} 2\mathcal{P}_{yt}(\tilde{y},\tilde{t})\right)^2} \tag{142}$$

Using $T_{t_{max}}^{\text{double}} = \phi$,

$$\Sigma_M^{pmp(k+1)}(y,t) = \frac{\gamma_t^{(k)}}{\lambda_t^2} \Sigma_M^{pmp(k+1)}(y,t_{max}) \tag{143}$$

Since $X_v^{(k+1)} = A^{(k+1)} M_v^{(k+1)}$, the following equation holds.

$$\Sigma_X^{pmp(k+1)}(y,t) = A^{(k+1)} \Sigma_M^{pmp(k+1)}(y,t) A^{(k+1)\top} \tag{144}$$

$$= A^{(k+1)} \frac{\gamma_t^{(k)}}{\lambda_t^2} \Sigma_M^{pmp(k+1)}(y,t_{max}) A^{(k+1)\top} \tag{145}$$

$$= \frac{\gamma_t^{(k)}}{\lambda_t^2} \Sigma_X^{pmp(k+1)}(y,t_{max}) \tag{146}$$

Therefore, we proved that if $\beta_t^{(k)}\Sigma_X^{pmp(k)}(y,t_{max}) = \Sigma_X^{pmp(k)}(y,t)$ holds for $k$, then for constants $\gamma_t^{(k)}, \lambda_t, \beta_t^{(k+1)}$ which depends only on time and layer, $\Sigma_M^{pmp(k+1)}(y,t) = \frac{\gamma_t^{(k)}}{\lambda_t^2} \Sigma_M^{pmp(k+1)}(y,t_{max})$ and $\beta_t^{(k+1)}\Sigma_X^{pmp(k+1)}(y,t_{max}) = \Sigma_X^{pmp(k+1)}(y,t)$ holds. By induction, lemma is proved.

### A.11.3 PROOF OF THEOREM 5.2

In this discussion, we will regard $\mu_M^{JJ}(\cdot,t)$ and $\mu_M^{JJ}(y,t)$ as constant, since generally there are sufficient number of samples in each community, especially for large-scale graphs.

As shown earlier, when passing through PMP, the covariance matrix of the aggregated message is as follows.

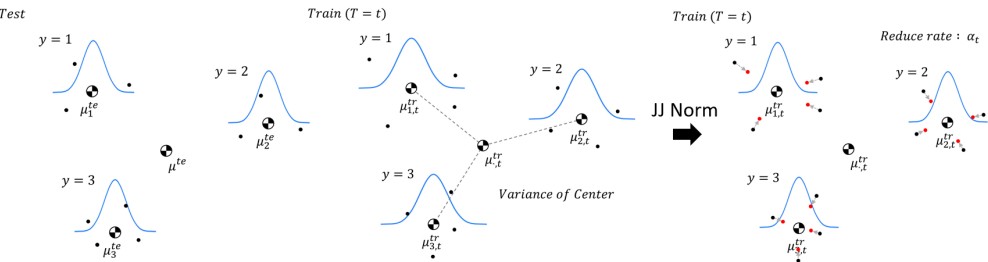

Figure 9: Graphical explanation of JJNORM. Under assumption 4, covariance matrices of aggregated message on each community differs only by a constant factor $\alpha_t$.

Unlike PNY, which estimates an affine transformation using $\hat{\mathcal{P}}_{yt}(\tilde{y}, \tilde{t})$ to align the covariance matrix to be invariant, JJNORM provides a more direct method to obtain an estimate $\hat{\alpha}_t$ of $\alpha_t$. Since the objective of this section is to get a sufficiently good estimator for implementation, the equations here may be heuristic but are proceeded with intuitive reasons.

Since we know that the covariance matrix differs only by a constant factor, we can simply use norms in multidimensional space rather than the covariance matrix to estimate $\alpha_t$.

Firstly, let's define $\mathbf{V}_{y,t} = \{u \in \mathbf{V} \mid u \text{ has label } y, u \text{ has time } t\}$, $\mathbf{V}_{\cdot,t} = \{u \in \mathbf{V} \mid u \text{ has time } t\}$.

Let us define

$$\sigma_{y,t}^2 = \mathbb{E}_{v \sim \mathbf{V}_{y,t}}[(M_v - \mu_M(y, t))^2] = \frac{1}{|\mathbf{V}_{y,t}|} \sum_{v \in \mathbf{V}_{y,t}} (M_v - \mu_M(y, t))^2 \tag{147}$$

$$\sigma_{\cdot,t}^2 = \mathbb{E}_{v \sim \mathbf{V}_{\cdot,t}}[(M_v - \mu_M(t))^2] = \frac{1}{|\mathbf{V}_{\cdot,t}|} \sum_{v \in \mathbf{V}_{\cdot,t}} (M_v - \mu_M(t))^2 \tag{148}$$

$$\mu_{y,t} = \mathbb{E}_{v \sim \mathbf{V}_{y,t}}[M_v] = \frac{1}{|\mathbf{V}_{y,t}|} \sum_{v \in \mathbf{V}_{y,t}} M_v \tag{149}$$

$$\mu_{y,t} = \mathbb{E}_{v \sim \mathbf{V}_{\cdot,t}}[M_v] = \frac{1}{|\mathbf{V}_{y,t}|} \sum_{v \in \mathbf{V}_{\cdot,t}} M_v \tag{150}$$

Note that definition of mean and variance here, are different with the definitions stated in A.11.1. Here, JJNORM is a process of transforming the aggregated message, which is aggregated through PMP, into a time-invariant representation. Hence, we can suppose that $\mu_M(y, t)$ is invariant to $t$. That is, for all $t \in \mathbf{T}$, $\mu_M(y, t) = \mu_M(y, t_{max})$. Additionally, we can define the variance of distances as follows: $\sigma_{y,t}^2 = \mathbb{E}_{v \in \mathbf{V}_{y,t}} \left[ (M_v - \mu_M(y, t))^2 \right]$ and $\sigma_{\cdot,t}^2 = \mathbb{E}_{v \in \mathbf{V}_{\cdot,t}} \left[ (M_v - \mu_M(t))^2 \right]$. Here, the square operation denotes the L2-norm.

$$\mathbb{E}_{v \in \mathbf{V}_{\cdot,t}} \left[ (M_v - \mu_M(t))^2 \right] = \sum_{y \in \mathbf{Y}} P(y) \mathbb{E}_{v \in \mathbf{V}_{y,t}} \left[ (M_v - \mu_M(y, t) + \mu_M(y, t) - \mu_M(t))^2 \right] \tag{151}$$

$$= \sum_{y \in \mathbf{Y}} P(y) \Big( \mathbb{E}_{v \in \mathbf{V}_{y,t}} \left[ (M_v - \mu_M(y, t))^2 \right] + (\mu_M(y, t) - \mu_M(t))^2 \Big) \tag{152}$$

Since $\mathbb{E}_{v \in \mathbf{V}_{y,t}} \left[ (M_v - \mu_M(y, t))^\top (\mu_M(y, t) - \mu_M(t)) \right] = 0$.

Here, mean of the aggregated messages during training and testing times satisfies the following equation: $\mu_M(t) = \mu_M(t_{max})$

$$\mu_M(t) = \sum_{y \in \mathbf{Y}} P(y) \mu_M(y, t) = \sum_{y \in \mathbf{Y}} P(y) \mu_M(y, t_{max}) = \mu_M(t_{max}) \tag{153}$$

This equation is derived from the assumption that $\mu_M(y, t)$ is invariant to $t$ and from Assumption 1 regarding $P(y)$. Furthermore, by using Assumption 1 again, we can show that the variance of the mean computed for each label is also invariant to $t$:

$$\sum_{y \in \mathbf{Y}} P(y) \mathbb{E}_{v \in \mathbf{V}_{y,t}} \left[ (\mu_M(y, t) - \mu_M(t))^2 \right] = \sum_{y \in \mathbf{Y}} P(y) \mathbb{E}_{v \in \mathbf{V}_{y,t_{max}}} \left[ (\mu_M(y, t_{max}) - \mu_M(t_{max}))^2 \right]$$
$$\tag{154}$$

$$\mathbb{E}_{v \in \mathbf{V}_{y,t}} \left[ (\mu_M(y, t) - \mu_M(t))^2 \right] = \mathbb{E}_{v \in \mathbf{V}_{y,t_{max}}} \left[ (\mu_M(y, t_{max}) - \mu_M(t_{max}))^2 \right] = \nu^2, \ t \in \mathbf{T} \tag{155}$$

Here, $\nu^2$ can be interpreted as the variance of the mean of messages from nodes with the same $t \in \mathbf{T}$ for each label. According to the above equality, this is a value invariant to $t$.

Meanwhile, from Assumption 4,

$$\alpha_t \mathbb{E}_{v \in \mathbf{V}_{y,t}} \left[ (M - \mu_M(y,t))^2 \right] = \mathbb{E}_{v \in \mathbf{V}_{y,t_{max}}} \left[ (M - \mu_M(y,t_{max}))^2 \right], \forall t \in \mathbf{T} \tag{156}$$

$$\alpha_t \sum_{y \in \mathbf{Y}} P(y) \mathbb{E}_{v \in \mathbf{V}_{y,t}} \left[ (M_v - \mu_M(y,t))^2 \right] = \sum_{y \in \mathbf{Y}} P(y) \mathbb{E}_{v \in \mathbf{V}_{y,t_{max}}} \left[ (M_v - \mu_M(y,t_{max}))^2 \right] \tag{157}$$

Adding $\nu^2$ to both sides,

$$\alpha_t \sum_{y \in \mathbf{Y}} P(y) \mathbb{E}_{v \in \mathbf{V}_{y,t}} \left[ (M_v - \mu_M(y,t))^2 \right] + \sum_{y \in \mathbf{Y}} P(y) \mathbb{E}_{v \in \mathbf{V}_{y,t}} \left[ (\mu_M(y,t) - \mu_M(t))^2 \right] = \sigma^2_{\cdot,t_{max}} \tag{158}$$

Thus,

$$\alpha_t = \frac{\sigma^2_{\cdot,t_{max}} - \nu^2}{\sum_{y \in \mathbf{Y}} P(y) \mathbb{E}_{v \in \mathbf{V}_{y,t}} \left[ (M_v - \mu_M(y,t))^2 \right]} \tag{159}$$

Here, $\hat{\alpha}_t$ is an unbiased estimator of $\alpha_t$.

$$\hat{\nu}^2 = \frac{1}{|\mathbf{V}_{\cdot,t}| - 1} \sum_{y \in \mathbf{Y}} \sum_{v \in \mathbf{V}_{y,t}} (\hat{\mu}_M(y,t) - \hat{\mu}_M(t))^2 \tag{160}$$

$$\hat{\alpha}_t = \frac{\left( \frac{1}{|\mathbf{V}_{\cdot,t_{max}}| - 1} \sum_{v \in \mathbf{V}_{\cdot,t_{max}}} (M_v - \hat{\mu}_M(t_{max}))^2 - \hat{\nu}^2 \right)}{\frac{1}{|\mathbf{V}_{\cdot,t}| - 1} \sum_{y \in \mathbf{Y}} \sum_{v \in \mathbf{V}_{y,t}} (M_v - \hat{\mu}_M(y,t))^2} \tag{161}$$

Where $\hat{\mu}_M(y,t) = \frac{1}{|\mathbf{V}_{y,t}|} \sum_{v \in \mathbf{V}_{y,t}} M_v$ and $\hat{\mu}_M(t) = \frac{1}{|\mathbf{V}_{\cdot,t}|} \sum_{v \in \mathbf{V}_{\cdot,t}} M_v$.

Note that all three terms in the above equation can be directly computed without requiring test labels.

By using $\hat{\alpha}_t$, we can update the aggregated message from PMP to align the second-order statistics.

$$M_v^{JJnorm} \leftarrow \hat{\mu}_M(y,t) + \hat{\alpha}_t(M_v - \hat{\mu}_M(y,t)), \ \forall i \in \mathbf{V} \setminus \mathbf{V}_{\cdot,t_{max}} \tag{162}$$

A.12 DETAILED EXPERIMENTAL SETUP FOR SYNTHETIC GRAPH EXPERIMENTS.

In our experiments, we set $f = 5$, $k_y$ was sampled from a uniform distribution in $[0, 8]$, and the center of features for each label $\mu(y) \in \mathbb{R}^f$ was sampled from a standard normal distribution. Each graph consisted of 2000 nodes, with a possible set of times $\mathbf{T} = \{0, 1, \ldots, 9\}$ and a set of labels $\mathbf{Y} = \{0, 1, \ldots, 9\}$, with time and label uniformly distributed. Therefore, the number of communities is 100, each comprising 20 nodes. Additionally, we defined $\mathbf{V}_{te} = \{u \in \mathbf{V} \mid u \text{ has time } \geq 8\}$ and $\mathbf{V}_{tr} = \{u \in \mathbf{V} \mid u \text{ has time } < 8\}$. When communities have an equal number of nodes, the following relationship holds:

$$\mathbf{P}_{t\tilde{t}y\tilde{y}} = \gamma_{y,\tilde{y}}^{|t-\tilde{t}|} \mathbf{P}_{tty\tilde{y}}, \ \forall |t - \tilde{t}| > 0 \tag{163}$$

To fully determine the tensor $\mathbf{P}_{t\tilde{t}y\tilde{y}}$, we needed to specify the values when $t = \tilde{t}$. In order to imbue the graph with topological information, we defined two hyperparameters, $\mathcal{K}$ and $\mathcal{G}$, such that $\mathcal{K} < \mathcal{G}$. For any $y, \tilde{y} \in \mathbf{Y}$, if $y = \tilde{y}$, we sampled $\mathcal{P}_{y,t,\tilde{y},t}$ from a uniform distribution in $[0, \mathcal{K}]$, and if $y \neq \tilde{y}$, we sampled $\mathcal{P}_{y,t,\tilde{y},t}$ from a uniform distribution in $[0, \mathcal{G}]$. In our experiments, we used $\mathcal{K} = 0.6$ and $\mathcal{G} = 0.24$.

For cases where Assumption 4 was not satisfied, $\gamma_{y,\tilde{y}}$ was sampled from a uniform distribution $[0.4, 0.7]$. For cases where Assumption 4 was satisfied, all decay factors were the same, i.e., $\gamma_{y,\tilde{y}} = \gamma, \ \forall y, \tilde{y} \in \mathbf{Y}$. In this case, $\gamma$ indicates the extent to which the connection probability varies with the time difference between two nodes. A smaller $\gamma$ corresponds to a graph where the connection probability decreases drastically. We also compared the trends in the performance of each IMPaCT method by varying the value of $\gamma$. The baseline SGC consisted of 2 layers of message passing and

---

**Algorithm 5:** Persistent Message PassingJJ normalization

---

**Input** : Aggregated message $M_v, \forall v \in \mathbf{V}$, obtained from 1st moment alignment message passing; node time function $time : \mathbf{V} \to \mathbf{T}$; train, test split $\mathbf{V}^{tr} = \{v \mid v \in \mathbf{V}, time(v) < t_{\max}\}$ and $\mathbf{V}^{te} = \{v \mid v \in \mathbf{V}, time(v) = t_{\max}\}$; node label funtion $label : \mathbf{V}^{tr} \to \mathbf{Y}$.

**Output:** Modified aggregated message $M'_v, \forall v \in \mathbf{V}$.

1 Let $\mathbf{V}_{y,t} = \{u \in \mathbf{V} \mid label(u) = y, time(u) = t\}$;
2 Let $\mathbf{V}_{\cdot,t} = \{u \in \mathbf{V} \mid time(u) = t\}$;

3 **Estimate mean and variance for each community.**
4 **for** $t \in \mathbf{T}$ **do**
5     $\hat{\mu}_M(\cdot, t) \leftarrow \hat{\mu}_M(\cdot, t) = \frac{1}{|\mathbf{V}_{\cdot,t}|} \sum_{v \in \mathbf{V}_{\cdot,t}} M_v$;
6 **end**
7 **for** $y \in \mathbf{Y}$ **do**
8     **for** $t \in \{\ldots, t_{\max} - 1\}$ **do**
9        $\hat{\nu}_t^2 \leftarrow \frac{1}{|\mathbf{V}_{\cdot,t}|-1} \sum_{y \in \mathbf{Y}} \sum_{v \in \mathbf{V}_{y,t}} (\hat{\mu}_M(y,t) - \hat{\mu}_M(\cdot,t))^2$;
10     **end**
11 **end**
12 **for** $y \in \mathbf{Y}$ **do**
13     **for** $t \in \{\ldots, t_{\max} - 1\}$ **do**
14        $\hat{\mu}_M(y,t) \leftarrow \frac{1}{|\mathbf{V}_{y,t}|} \sum_{v \in \mathbf{V}_{y,t}} M_v$;
15        $\hat{\sigma}_{y,t}^2 \leftarrow \frac{1}{|\mathbf{V}_{\cdot,t}|-1} \sum_{y \in \mathbf{Y}} \sum_{v \in \mathbf{V}_{y,t}} (M_v - \hat{\mu}_M(y,t))^2$;
16     **end**
17 **end**
18 $\hat{\sigma}_{t_{\max}}^2 \leftarrow \frac{1}{|\mathbf{V}_{\cdot,t_{\max}}|-1} \sum_{v \in \mathbf{V}_{\cdot,t_{\max}}} (M_v - \hat{\mu}_M(\cdot, t_{\max}))^2 - \frac{1}{|\mathbf{V}_{\cdot,t}|-1} \sum_{y \in \mathbf{Y}} \sum_{v \in \mathbf{V}_{y,t}} (\hat{\mu}_M(y,t) - \hat{\mu}_M(\cdot,t))^2$;

19 **Estimate $\hat{\alpha}_t$ for $t < t_{\max}$.**
20 **for** $t \in \{\ldots, t_{\max} - 1\}$ **do**
21     $\hat{\alpha}_t^2 \leftarrow \frac{\hat{\sigma}_{t_{\max}}^2 - \hat{\nu}_t^2}{\hat{\sigma}_{y,t}^2}$;
22 **end**

23 **Update aggregated message.**
24 **for** $v \in \mathbf{V} \setminus \mathbf{V}_{\cdot,t_{\max}}$ **do**
25     Let $y = label(i)$;
26     Let $t = time(i)$;
27     $M'_v \leftarrow \hat{\mu}_M(y,t) + \hat{\alpha}_t(M_v - \hat{\mu}_M(y,t)), \forall v \in \mathbf{V} \setminus \mathbf{V}_{\cdot,t_{\max}}$;
28 **end**

---

2 layers of MLP, with the hidden layer dimension set to 16. The baseline GCN also consisted of 2 layers with the hidden layer dimension set to 16. Adam optimizer was used for training with a learning rate of 0.01 and a weight decay of 0.0005. Each model was trained for 200 epochs, and each data was obtained by repeating experiments on 200 random graph datasets generated through TSBM. The training of both models were conducted on a 2X Intel Xeon Platinum 8268 CPU with 48 cores and 192GB RAM.

### A.13 SCALABILITY OF INVARIANT MESSAGE PASSING METHODS

First moment alignment methods such as MMP and PMP have the same complexity and can be easily applied by modifying the graph. By adding or removing edges according to the conditions, only $\mathcal{O}(|E|)$ additional preprocessing time is required, which is necessary only once throughout the entire training process. If the graph cannot be modified and the message passing function needs to be modified instead, it would require $\mathcal{O}(|E|fK)$, which is equivalent to the traditional averaging message passing. Similarly, the memory complexity remains $\mathcal{O}(|E|fK)$, consistent with traditional averaging message passing. Despite having the same complexity, PMP is much more expressive than MMP. Unless there are specific circumstances, PMP is recommended for first moment alignment.

In PNY, estimating the relative connectivity $\hat{\mathcal{P}}_{y,t}(\tilde{y}, \tilde{t})$ requires careful consideration. If both $t \neq t_{max}$ and $\tilde{t} \neq t_{max}$, calculating the relative connectivity for all pairs involves $\mathcal{O}((N + |E|)f)$ operations, while computing for cases where either time is $t_{max}$ requires $\mathcal{O}(|Y|^2|T|^2)$ computations. Therefore, the total time complexity becomes $\mathcal{O}(|Y|^2|T|^2 + (N + |E|)f)$. Additionally, for each message passing step, the covariance matrix of the previous layer's representation and the aggregated message needs to be computed for each label-time pair. Calculating the covariance matrix of the representation from the previous layer requires $\mathcal{O}((|Y||T| + N)f^2)$ operations. Subsequently, computing the covariance matrix of the aggregated message obtained through PMP via relative connectivity requires $\mathcal{O}(|Y|^2|T|^2f^2)$ operations. Diagonalizing each of them to create affine transforms requires $\mathcal{O}(|Y||T|f^3)$, and transforming all representations requires $\mathcal{O}(Nf^2)$. Thus, with a total of $K$ layers of topological aggregation, the time complexity for applying PNY becomes $\mathcal{O}(K(|Y||T|f^3 + |Y|^2|T|^2f^2 + Nf^2) + |E|f)$. Additionally, the memory complexity includes storing covariance matrices based on relative connectivity and label-time information, which is $\mathcal{O}(|Y||T|f^2 + |Y|^2|T|^2)$.

Now, let's consider applying PNY to real-world massive graph data. For instance, in the ogbn-mag dataset, $|Y| = 349$, $|T| = 11$, $N = 629571$, and $|E| = 21111007$. Assuming a representation dimension of $f = 512$, it becomes apparent that performing at least several trillion floating-point operations is necessary. Without approximation or transformations, applying PNY to large graphs becomes challenging in terms of scalability.

Lastly, for JJNORM, computing the sample mean of aggregated messages for each label and time pair requires $\mathcal{O}(Nf)$ operations. Based on this, computing the total variance, variance of the mean, and mean of representations with each time requires $\mathcal{O}(Nf)$ operations. Calculating each $\hat{\alpha}_t$ requires $O(|T|)$ operations, and modifying the aggregated message based on this requires $\mathcal{O}(Nf)$ operations, resulting in a total of $\mathcal{O}(Nf + |T|) \simeq \mathcal{O}(Nf)$ operations. For GNNs with nonlinear node-wise semantic aggregation function with a total of $K$ layers, layer-wise JJNORM have to be applied, which results in $\mathcal{O}(NfK)$ operations. Additionally, the memory complexity becomes $\mathcal{O}(|Y||T|f)$. Considering that most operations in JJNORM can be parallelized, it exhibits excellent scalability.

In experiments with synthetic graphs, it was shown that invariant message passing methods can be applied to general spatial GNNs, not just decoupled GNNs. For 1st moment alignment methods such as PMP and MMP, which can be applied by reconstructing the graph, they have the same time and memory complexity as calculated above. However, for 2nd moment alignment methods such as JJNORM or PNY, transformation is required for each message passing step, resulting in a time complexity multiplied by the number of epochs as calculated above. Therefore, when using general spatial GNNs on real-world graphs, only 1st moment alignment methods may be realistically applicable.

**Guidelines for deciding which IMPaCT method to use.** Based on these findings, we propose guidelines for deciding which invariant message passing method to use. If the graph exhibits differences in environments due to temporal information, we recommend starting with PMP to make the representation's 1st moment invariant during training. MMP is generally not recommended. Next,

if using Decoupled GNNs, PNY and JJNORM should be compared. If the graph is too large to apply PNY, compare the results of using PMP alone with using both PMP and JJNORM. In cases where there are no nonlinear operations in the message passing stage, JJNORM needs to be applied only once at the end. Using 2nd moment alignment methods with General Spatial GNNs may be challenging unless scalability is improved.

Caution is warranted when applying invariant message passing methods to real-world data. If Assumptions do not hold or if the semantic aggregation functions between layers exhibit loose Lipschitz continuity, the differences in the distribution of final representations over time cannot be ignored. Therefore, rather than relying on a single method, exploring various combinations of the proposed invariant message passing methods to find the best-performing approach is recommended.

