# OpenReview forum: "IMPaCT GNN: Imposing invariance with Message Passing in Chronological split Temporal Graphs"
_ICLR.cc/2025/Conference — Submitted to ICLR 2025_

### Official Review · Reviewer_XZDb · 2024-11-02

**Soundness:** 2
**Presentation:** 1
**Contribution:** 2
**Rating:** 5
**Confidence:** 4

**Summary:**

The paper addresses non-stationary behaviors observed at the temporal boundaries of temporal graphs. Specifically, it proposes weighting techniques to adjust message-passing operations, enabling the learned representations to maintain a degree of invariance over time. While the novelty is reasonable rather than groundbreaking, I appreciate the effort to tackle these issues, as I am not aware of other papers that address them explicitly.

**Strengths:**

- Addressing biases introduced by data partitions is a relevant and valuable challenge.
- Some of the proposed techniques are demonstrated effective in mitigating the targeted issues.

**Weaknesses:**

- The paper presentation does not appear mature enough for publication.
- The math appears insufficiently rigorous.
- The proposed methods are limited to a specific family of message-passing operators that rely on average aggregation, without accommodating more complex methods like attention or target node-dependent messages.
- Some of the experiments are not convincing.

**Questions:**

Questions.
- The concept "Relative connectivity $\mathcal P$" is not clearly introduced. Does it represent the empirical distribution of edges across time and node classes?
- Where does the randomness associated with  $x_{yt}^{(k)}$ come from? Does it originate solely from the system model generating the node features?
- Eq 7. Why is "IID"? It looks to me that they are not "identically distributed" as they depend on $k$, $y$ and $t$, nor independent due to the message passing.
- Eq 28, part b is said to be an approximation. If so then the equality symbol should not be used. Could you clarify in the paper which steps involve approximations and which provide strict equalities?
- Line 219 is in contrast with assumption 3. If the probability decays then the two contributions do not match. Am I missing something here? How can this apparent contradiction be reconciled?
- Training times in Table 3. How is it possible that training for 200 epochs takes only a fraction of a second? Why does PMP on SGC take less time than the baseline?
- PNY and JJnorm on GCN have extremely long training times (about 500 X more than the others), which is concerning. Could you comment on it?
- Results on OGB-mag data are around 90% accuracy which does not appear in line with their website reporting ~57% for leading models. Could you elaborate on this?
- How difficult is it to port the designed techniques to more general message-passing operators and to apply them to the methods leading the OGB benchmarks?
- I could not find a discussion about the extent to which trainable model parameters within the message passing impact the proposed weighing techniques. Could you elaborate?
- Line 22. Which "unverifiable assumptions" are you referring to exactly?
- Line 128 "data from different environments may have interdependencies, and the extrapolating nature of environments complicates the problem." is completely unclear to me. Could you clarify it?


Further comments and suggested improvements:
- The addressed temporal graph learning problem is not formally stated. It becomes clear only once the datasets are presented. I suggest providing a thorough formulation and introducing there the adopted notation. In this regard, I suggest formulating a more general message passing involving a temporal graph so that the reader can relate to it the new methods later on.
- Figures 2 and 3 are not super informative and, in my opinion, unnecessary. Figure 2 only shows a decay along the temporal semi-axes - a concept that could be given for granted. Figure 3 is not accompanied by a description.
- Denoting a distribution as $x_{yt}^{(k)}$ is rather confusing.
- One important element is that the non-stationary behavior (here called invariance - which, by the way, I don't think is the most appropriate term to use here) comes from the edges rather than the node features, as per the assumptions made. However, such dependency from the edges does not emerge from the math developments.
- What does "discrete value" mean in A.4? They are indeed positive integers.
- The term "persistent" in 4.1 is unclear.
- Table 3. Please provide variability indices (eg, std) alongside results.
- Decoupled GNN seems to not be introduced or referenced.

---

> ### Author Response · Authors · 2024-11-24
>
> We highly appreciate your constructive comments and evaluation. In the following, we address each of the reviewer's concerns in detail.
>
> Weakness: The proposed methods are limited to a specific family of message-passing operators that rely on average aggregation, without accommodating more complex methods like attention or target node-dependent messages.
>
> - Our study addressed the lack of domain adaptation research for large-scale graph data by proposing the IMPaCT method, tailored to models utilizing average aggregation for message passing. This choice reflects the characteristics of scalable GNNs, which, for large-scale graphs, often rely on decoupled GNNs, performing message passing only during preprocessing without learnable weights, thus favoring average message passing strategies.
> - Future work could focus on adapting our method to more complex frameworks, such as attention-based or target node-dependent message passing.
>
> Weakness: Some of the experiments are not convincing.
>
> - The experiments conducted on ogbn-mag/synthetic graphs and ogbn-papers100M/ogbn-arxiv were designed with different objectives, which is why the evaluation protocols varied. For ogbn-mag and synthetic graphs, the focus was on demonstrating how our proposed methods, such as PMP and JJNorm, effectively address domain adaptation challenges when all assumptions are satisfied, resulting in state-of-the-art performance.
> - On the other hand, ogbn-papers100M and ogbn-arxiv presented more complex challenges due to significant discrepancies in node count distributions over time, which violated Assumption 3 (that the connection distribution depends on the absolute time difference between nodes). In these cases, GenPMP was applied to correct the temporal discrepancies in node counts during message passing. The purpose of these experiments was to test whether first-order alignment could still work by numerically adjusting for these mismatches in node counts across time attributes. However, when the node distributions were highly imbalanced, using GenPMP caused unbounded variance in the aggregated messages, making second-order corrections like JJNorm impractical.
> - In other words, the experiments on ogbn-papers100M and ogbn-arxiv were a limited test to investigate whether correcting numerical imbalances in time attributes allows Assumptions 1 and 2, along with the separability of connection distributions, to hold, and whether first-order alignment remains effective under such conditions. While no dramatic performance improvements were observed, we did see statistically significant performance improvements, confirming the objectives of these experiments.
>
> Question: The concept "Relative connectivity " is not clearly introduced. Does it represent the empirical distribution of edges across time and node classes?
>
> - Yes, relative connectivity represents the empirical distribution of edges across time and node classes.
> - Under our assumptions, this leads to a nonlinear overdetermined system for the empirical distributions.
> - While there are methods to obtain optimal solutions for such systems, these methods significantly increase time complexity. This added complexity would render the PNY method impractical for real-world, time-sensitive applications.
> - To address this, we opted for a suboptimal solution (as in Appendix A.9) that balances computational feasibility and real-time implementability.
>
> Question: Where does the randomness associated with  come from? Does it originate solely from the system model generating the node features?
>
> - We are assuming that there is a “true” distribution $x_{yt}^{k}$.
>
> Question: Eq 7. Why is "IID"? It looks to me that they are not "identically distributed" as they depend on ,  and , nor independent due to the message passing.
>
> - We agree with this pointer, and note that our initial eq7 has overassumptions.
> - For the development and analysis of PMP methods, the IID assumptions for every layers are not essential. We only need “identically distributed” assumptions of the “0th” layer features. We will correct equation (7) by erasing the part $X_w \overset{\text{IID}}{\sim} {x_{\tilde{y}\tilde{t}}^{(k)}} \; \text{for} \; \forall w\in\mathcal{N}_{v}\left(\tilde{y}, \tilde{t}\right)$.
> - One remark is that even the “identically distributed” assumption of 0th layer features can be weakened into “identical means” assumption of 0th layer features. However, even though the identically distributed is a mathematically stronger assumption, we believe that it is a more realistic assumption. The theory of PMP method can be obtained without corrections with only the [identical mean of 0th layer features] assumption.

---

> > ### Author Response · Authors · 2024-11-24
> >
> > Question: Eq 28, part b is said to be an approximation. If so then the equality symbol should not be used. Could you clarify in the paper which steps involve approximations and which provide strict equalities?
> >
> > - The equalities with (a) or (b) above are approximations, and equalities with only “=” are strict equalities. We constructed this process to aviod including \simeq in the main body.
> >
> > Question: Line 219 is in contrast with assumption 3. If the probability decays then the two contributions do not match. Am I missing something here? How can this apparent contradiction be reconciled?
> >
> > - Line 219 simply describes the definition of PMP and does not directly relate to the graph assumptions, including Assumption 3. It seems there may be a misunderstanding or a misreference. Could you kindly double-check the line number or provide additional clarification regarding the apparent contradiction? This would help us address your concern more accurately.
> >
> >
> >
> > Question: Training times in Table 3. How is it possible that training for 200 epochs takes only a fraction of a second? Why does PMP on SGC take less time than the baseline?
> >
> > - The training times in Table 3 are indeed influenced by the specific nature of the SGC (Simplified Graph Convolution) model and the way message passing is handled.
> > - In the case of SGC, message passing occurs only during the preprocessing step, not during the 200 epochs of training. The 200 epochs are spent training a classifier using the pre-aggregated messages, which means that the computationally expensive message passing process is only performed once, not repeatedly for each epoch. This is why the training time appears to be significantly shorter.
> > - Furthermore, both MMP and PMP are applied by modifying the graph prior to the training process, rather than altering the message passing function during training. As such, the time taken to modify the graph is not included in the reported training times in Table 3. Therefore, there should be no systematic difference in the execution times of MMP, PMP, and the baseline; any observed variation is likely due to the influence of the variance in the measurements.
> >
> > Question: PNY and JJnorm on GCN have extremely long training times (about 500 X more than the others), which is concerning. Could you comment on it?
> >
> > - Thank you for your question. The long training times observed for PNY and JJNorm on GCN, being approximately 500 times longer than other models, are indeed noteworthy. However, it's important to clarify the purpose of applying these methods to GCN.
> > - Our study primarily focused on addressing domain adaptation challenges for large-scale graph data, which is why it was not specifically aimed at GNNs like GCN that perform message passing repeatedly at each epoch. The experiment with GCN was intended to demonstrate that methods like PMP and JJNorm can be applied even to models that perform average message passing. Additionally, it served as a guideline to highlight that applying second-order alignment methods, such as JJNorm, to general GNNs (not decoupled GNNs) may be practically infeasible due to the significant computational cost involved. Thus, while the training times for GCN with PNY and JJNorm are long, this experiment helps to emphasize the time considerations and practical challenges in scaling these approaches.
> >
> > Question: Results on OGB-mag data are around 90% accuracy which does not appear in line with their website reporting ~57% for leading models. Could you elaborate on this?
> >
> > - Thank you for your insightful question. The discrepancy between our reported accuracy of approximately 90% on the OGB-MAG dataset and the ~57% accuracy listed for leading models on the OGB leaderboard is due to some recent changes on the leaderboard. Specifically, models such as CLGNN and LDHGNN, which were previously listed, are no longer visible. We also noted this issue and reached out to the authors of the CLGNN model for clarification. It appears that the removal was due to corrections or updates made by the authors themselves.
> > - To verify the most current results and see the complete leaderboard, we recommend visiting the Papers with Code website, or reviewing the arXiv papers for CLGNN and LDHGNN. These sources will provide the latest information on the state-of-the-art models for OGB-MAG.
> > - Here is the link to the Papers with Code page for your reference: https://paperswithcode.com/sota/node-property-prediction-on-ogbn-mag.

---

> > > ### Author Response · Authors · 2024-11-24
> > >
> > > Questions: How difficult is it to port the designed techniques to more general message-passing operators and to apply them to the methods leading the OGB benchmarks?, I could not find a discussion about the extent to which trainable model parameters within the message passing impact the proposed weighing techniques. Could you elaborate?
> > >
> > > - Our proposed method is currently limited to averaging-based message passing. Our study primarily targeted large-scale graph data, where scalable GNNs are commonly employed. These models, often decoupled GNNs, perform message passing during preprocessing without learnable weights, relying on averaging-based message passing. The IMPaCT method was designed to address domain adaptation within this context. Future work could explore the extension of our method to more complex message-passing strategies.
> > >
> > >
> > > Question: Line 22. Which "unverifiable assumptions" are you referring to exactly?
> > >
> > > - Traditional domain adaptation studies often assume that the data generation process inherently creates causal invariances (e.g., for samples with the same label), which can be learned through techniques like regularization or augmentation. However, this assumption of causal invariance is unverified, and in the case of chronological splits, the data generation process may evolve over time, potentially making such assumptions unrealistic. Thus, our approach identifies assumptions typically satisfied by real-world temporal graph data and proposes a message-passing technique that explicitly imposes these invariances.
> > >
> > > Question: Line 128 "data from different environments may have interdependencies, and the extrapolating nature of environments complicates the problem." is completely unclear to me. Could you clarify it?
> > >
> > > - The sentence you are referring to, "data from different environments may have interdependencies, and the extrapolating nature of environments complicates the problem," aims to address the challenges of working with temporal or domain-shifted data in dynamic environments.
> > > - To clarify, the data generation process evolves over time, and our goal is to classify the labels of the most recent data based on historical data. This means that learning invariances from past data does not necessarily generalize easily to recent data. The extrapolating nature refers to the difficulty of predicting or generalizing patterns from one environment (past data) to another (recent data), especially when these environments are interdependent but not identical. This makes the domain adaptation problem more complex, as the underlying patterns in the data may shift over time, complicating the task of applying learned knowledge from one time period to another.
> > >
> > > Suggested improvements: Denoting a distribution as $x_ {yt}^{\text{(k)}}$ is rather confusing.
> > >
> > > - As we denoted the random variable as $X_w$, we denoted the distribution (or probability measure) as $x_{yt}^{(k)}$. If there were confusements, we apologize for the unintentional unclarity.
> > >
> > > Suggested improvements: What does "discrete value" mean in A.4? They are indeed positive integers.
> > >
> > > - Positive integers are discrete values, and the probabilities are non-discrete values. This section is simply making approximations of fractions of integers into fractions of probabilities.
> > >
> > > Suggested improvements: The term "persistent" in 4.1 is unclear.
> > >
> > > - The invariance is a characteristic for every layer in the message passing (Def 4.1, Thm 4.1). The term “persistent” is intended to deliver the meaning that the invariant property is persistently maintained among layers.
> > >
> > > Suggested improvements: Decoupled GNN seems to not be introduced or referenced.
> > >
> > > - Thank you for pointing this out. The concept of decoupled GNNs is introduced in Section 2.1 as follows, but we will provide further clarification to ensure a clearer understanding.

---

> > > > ### Comment · Reviewer_XZDb · 2024-11-27
> > > >
> > > > I appreciate the authors' time and effort in addressing all my questions. Several aspects of the paper are now clearer to me. Overall, I find the paper's contribution to be fair; however, significant improvements in both presentation and mathematical rigor are necessary for the work to meet publication standards.
> > > >
> > > > In response to the authors' feedback, I offer the following suggestions for improvement:
> > > > - I strongly encourage the authors to revise the paper to enhance the descriptions of the various sections I identified as unclear. Clearer explanations would significantly improve the paper's readability and accessibility.
> > > > - The comment on line 22 should be discussed more thoroughly within the paper and supported with appropriate references. In its current form, it is too vague to provide meaningful insight.
> > > > - I highly recommend that the authors clearly distinguish between approximations and equalities by using proper symbols and providing a detailed discussion. This distinction is critical to maintaining mathematical accuracy.
> > > > - Experiments on ogb-mag. I still find it unusual that the top two methods (CLGNN and LDHGNN) achieve accuracy around 90%, while the next seven methods are closely grouped with accuracy between 57% and 58%. Additionally, neither CLGNN nor LDHGNN is published nor listed on the ogb website. This raises concerns about potential inconsistencies in the experimental setup of the current paper too. I suggest double-checking the experimental setup to ensure it is fair and consistent. Where possible, provide a clear justification for why such a significant performance boost is reasonable.
> > > > - Line 219. One is twice the other only if the contributions for the time steps are the same, however the decaying assumption suggests that the contributions are not the same. Probably what the authors intended to state in a convoluted way is simply that two $\tilde t$ achieve the same $\Delta$ in $T^{double}$ instead of only one in $T^{single}$?

---

> > > > > ### Author Response · Authors · 2024-12-02
> > > > >
> > > > > Dear Reviewer XZDb,
> > > > >
> > > > > Up to our knowledge, from now on, this is the last 24 hours which reviewers may post a message to the authors.
> > > > > We would be grateful if you engage with our final comments.
> > > > >
> > > > > Best regards,
> > > > > Authors

---

> > > > > > ### Comment · Reviewer_XZDb · 2024-12-02
> > > > > >
> > > > > > Although the authors' feedback was overall helpful, my primary concern about the paper's clarity cannot be considered resolved. Therefore, I will maintain my previous evaluation and encourage the authors to address this issue further.

---

> ### Author Response · Authors · 2024-11-27
>
> Thank you for the comments. The constructive feedbacks certainly helped us improve and refine our paper.
>
> Regarding the 1st ~ 3rd suggestions, we have uploaded a revised version of our paper. Major changes are refined notations and additional motivational explanations incorporated into Section 4.
> In particular, regarding the third suggestion, we would like to remark that the definitions of $\hat{\mathbb E}$ and $\hat{\text{var}}$ representing the approximations of the mean and variance, respectively, were already addressed in the main body of our initial paper (see lines 168–173, 198–202, and 335–336 in revised version). To ensure clarity, we respectfully refer the reviewer to these sections in case this detail was overlooked.
>
> Regarding the 4th suggestion, we have also considered this point seriously. However, it is clear that both CLGNN and LDHGNN were tested in the same setting as the previous study on the leaderboard. We were also curious about the large gap, so we checked the code and paper thoroughly, but there was no problem such as label leakage. LDHGNN is based on CLGNN, and CLGNN was the first method to apply the curriculum learning approach. We believe that the large gap occurred because the curriculum learning approach has never been applied to the existing leaderboard.
>
> Regarding the 5th suggestion, we believe that the statement reviewer has written is what we intended.
>
> More thorough explanations are as follows.
> - Consider a setting with $t_{max}$ = 2019
> - For target node $v$ with time $t = 2018, v$ recieves aggregated messages with $\Delta = 0, 2, 3, 4, ...$ from neighbor nodes with time stamp $\tilde{t} = 2018, 2016, 2015, 2014, ...$, respectively. Also, $v$ recieves aggregated messages with $\Delta = 1$ from neighbor nodes with time stamp $\tilde{t} = 2017$ and both from neighbor nodes with time stamp $\tilde{t} = 2019$.
> - In other words, two types of nodes contribute as $\Delta = 1$, and only one type of node contributes as $\Delta$ other than $1$.
> - If we double the nodes with $\Delta$ other than $1$, that is, the single nodes, then for $\forall \Delta \ge 0$, two types of nodes can contribute.
>
>
> Here is a similar example for target node $v$ with time $t = 2017$.
> - In this case, $2019$ and $2015$ both have $\Delta = 2$, and $2018$ and $2016$ both have $\Delta = 1$.
> - Two types of nodes contribute as $\Delta = 1$ or $2$.
> - Only one type of node contributes as $\Delta$ other than $1$ and $2$.
> - If we double the nodes with $\Delta$ other than $1$ and $2$, same result happens as before (for $\forall \Delta \ge 0$, two types of nodes can contribute).
>
> - In conclusion, we can align every distribution of target nodes with different time stamps.

---

### Official Review · Reviewer_hoSU · 2024-11-03

**Soundness:** 3
**Presentation:** 3
**Contribution:** 3
**Rating:** 6
**Confidence:** 3

**Summary:**

This paper addresses domain adaption issues in graph data, particularly in datasets where the train and test splits are organized chronologically. The proposed method, named IMPaCT, assumes the separability of relative connectivity among nodes in the graph at different time points. Based on this assumption, the authors impose the invariance property in message-passing methods and provide various theoretical analyses on the generalization error.

**Strengths:**

- The paper is well structured, clearly presenting the problem and the proposed solution. The theoretical foundation of the proposed methods is robust, and experiments are presented to demonstrate their effectiveness.
- If we accept the correctness of the assumptions, especially Assumption 3 regarding the separability of relative connectivity, the proposed idea is quite intuitive. That is, if the first moment of the previous representation remains invariant over time t, the expectation of the aggregated message can be independent of t when the target node (label y, time t) receives information twice from neighbors in T_{single}.

**Weaknesses:**

The Assumption 3 may raise the following issues:
- Ensuring invariance of 1st moment of the previous representation: the authors introduce Assumption 2 to address this concern. If the 1st moment of the previous representation is not invariant, how can the invariance of the aggregated message be maintained. How to ensure the 1st moment of initial features of graph nodes invariant?
- Effectiveness of graph modification: While modifying the original graph may benefit average message passing procedure, this approach might not be effective for other frameworks, such as Graph attention network or Message passing neural networks, where the weighting of average procedures is not independent of time t.

Minor comments:
- Line 852: Equation (30) may contain a typo.
- Line 1737, Equation (161), do P on the left-hand side and right-hand side refer the same one?
- in Subsection 6.1, the def of f(y,t) from Assumption 3 is not provided

**Questions:**

- See Weaknesses above.

---

> ### Author Response · Authors · 2024-11-24
>
> We highly appreciate your constructive comments and evaluation. In the following, we address each of the reviewer's concerns in detail.
>
> Weakness 1.
>
> - We agree with the general notions of your comment. Below we will give some clarifications as follows.
> - The 0th layer feature’s invariance can not be ensured theoretically. Instead we provide the experimental evidence in Figure 5.
> - As the 0th layer feature’s invariance is assured experimentally, starting from the 1st layer, invariance is attained theoretically by our proposed PMP method.
>
> Weakness 2.
>
> - Indeed, our proposed method is currently limited to averaging-based message passing. Our study primarily targeted large-scale graph data, where scalable GNNs are commonly employed. These models, often decoupled GNNs, perform message passing during preprocessing without learnable weights, relying on averaging-based message passing. The IMPaCT method was designed to address domain adaptation within this context.
> - Future work could explore the extension of our method to more complex message-passing strategies.
>
> Regarding minor comments, we have corrected the typos and will update a new version of our paper. Thank you for the attentive feedbacks. Corrections are as follows:
>
> - We corrected the typo in Equation (30).
> - The $\mathbf{P}_ {t \tilde{t} y \tilde{y}}$ in the RHS of Equation (161) should be $\mathbf{P}_ {t t y \tilde{y}}$ .
> - The function $f(y,t)$ serves as a scaling factor corresponding to the label-time pair $(y,t)$. Once the values of $g(y,\tilde{y},|\tilde{t}-t|))$ are determined, $f(y,t)$ can be uniquely derived. By fixing $(y,t)$, the sum of $f(y,t)g(y,\tilde{y},|\tilde{t}-t|))$ over all possible $(\tilde{y},\tilde{t})$ must equal 1, which allows $f(y,t)$ to be computed accordingly.

---

> > ### Comment · Reviewer_hoSU · 2024-11-25
> >
> > Thank you for the clarification and revision. I'd like to keep my score unchanged.

---

### Official Review · Reviewer_u8i5 · 2024-11-03

**Soundness:** 3
**Presentation:** 2
**Contribution:** 3
**Rating:** 5
**Confidence:** 3

**Summary:**

This paper studies the problem of domain adaptation on temporal graph data under chronological splits. It introduced a novel IMPaCT approach to enforce invariant message passing via the 1st and 2nd moment alignments. The invariance of the proposed invariant message passing is theoretically analyzed. Experimental results also demonstrated the effectiveness of the invariance regularizations on real-world temporal graphs.

**Strengths:**

**Originality:** This paper studied the issues of chronological splits in applying GNNs to temporal graphs. It empirically observed the distribution shifts induced by the chronological splits. Then a novel IMPaCT was proposed by enforcing invariant node representation learning. Both the 1st and 2nd moments of aggregated messages were leveraged to improve the feature invariance.

**Quality:** Theoretical analysis showed the invariance of the iterative message aggregation using the proposed approaches. Besides, the generalization error bound based on the Wasserstein-1 distance was derived to show the impact of invariant message aggregation.


**Clarity:** The motivation of the studied problem was clear. The issues of chronological splits were validated in several real-world temporal graphs.


**Significance:** The developed theoretical analysis and algorithms extended the applications of GNNs from standard static graphs to temporal graphs under distribution shifts.

**Weaknesses:**

(W1) The defined approximate expectation $\hat{\mathbb{E}}[M^{k+1}_v]$ requires the IID assumption of $X_w$. This assumption is not justified on temporal graphs.

(W2) Theorem 4.1 and Theorem 5.1 show that PMP layers result in the invariance if the previous representation is invariant. It is unclear how the invariance of the initial representation can be guaranteed. If the previous representation is not invariant, will the aggregation of PMP layer worsen the invariance?

(W3) Another concern is the related work and baselines in the experiments. Graph domain adaptation and temporal GNNs have been studied in recent years. Those works can be discussed in the related works, and recent graph domain adaptation and temporal GNNs can be employed as the baselines to validate the effectiveness of the proposed approaches.

**Questions:**

(1) It is confusing why the target node receives twice the weight from $\tilde{t} \in \mathbf{T}_t^{double}$ against $\tilde{t} \in \mathbf{T}_t^{single}$.

(2) The generalization error bound depends on $C$ and $V$ defined in Section 4.3. The tightness of the derived generalization error bound can be further illustrated. As shown in lines 311-314, $V$ can be large in some cases.

---

> ### Author Response · Authors · 2024-11-24
>
> We highly appreciate your constructive comments and evaluation. In the following, we address each of the reviewer's concerns in detail.
>
> Weakness 1.
>
> - We agree with this pointer, and note that our initial eq7 has overassumptions.
> - For the development and analysis of PMP methods, the IID assumptions for every layers are not essential. We only need “identically distributed” assumptions of the “0th” layer features. We will correct equation (7) by erasing the part $X_w \overset{\text{IID}}{\sim} {x_{\tilde{y}\tilde{t}}^{(k)}}  \text{for}  \forall w\in\mathcal{N}_{v}\left(\tilde{y}, \tilde{t}\right)$.
> - One remark is that even the “identically distributed” assumption of 0th layer features can be weakened into “identical means” assumption of 0th layer features. However, even though the identically distributed is a mathematically stronger assumption, we believe that it is a more realistic assumption. The theory of PMP method can be obtained without corrections with only the [identical mean of 0th layer features] assumption.
>
> Weakness 2.
>
> - We agree with the general notions of this comment. Below we will give some clarifications as follows.
> - The 0th layer feature’s invariance can not be ensured theoretically. Instead we provide the experimental evidence in Figure 5.
> - As the 0th layer feature’s invariance is assured experimentally, starting from the 1st layer, invariance is attained theoretically by our proposed PMP method.
> - Impact of non-invariant 0th layer feature: If the 0th layer feature is not invariant, the aggregation process in the PMP layer does not guarantee the invariance of the subsequent layer. Thus, the invariance propagation would not hold in this case.
>
> Weakness 3.
>
> - Previous research is not applicable to large-scale graphs. Our study was motivated by the lack of domain adaptation research applicable to large-scale graph data. Notably, large graph datasets such as ogbn-mag and ogbn-papers100M use chronological splits, making domain adaptation challenges evident. However, no experiments on the leaderboard employ domain adaptation approaches, making direct numerical comparisons with existing methods infeasible.
> - Furthermore, traditional domain adaptation studies often assume that the data generation process inherently creates causal invariances (e.g., for samples with the same label), which can be learned through techniques like regularization or augmentation. However, this assumption of causal invariance is unverified, and in the case of chronological splits, the data generation process may evolve over time, making such assumptions unrealistic. Thus, our approach identifies assumptions typically satisfied by real-world temporal graph data and proposes a message-passing technique that explicitly imposes these invariances.
>
> Question 1.
>
> - Here is the explanation using Figure 3.
> - In this figure, $t_{max} = 2019$ and the target node’s time $t = 2018$.
> - By definition,  $\mathbf{T}_ {t}^{\text{double}} = \{ 2017, 2019 \}$ and $\mathbf{T}_ {t}^{\text{single}} = \{ 2018, 2016, 2015, 2014, ... \}$
> - Neighbor nodes with time in $\mathbf{T}_ {t}^{\text{double}}$, i.e., time $2017$ or time $2019$ have the same $\Delta = 1$ since the target node’s time is $2018$. Hence, the neighbor nodes with time $2017$ or $2019$ are giving the same amount of contributions when message passing to the target node (by Assumption 3).
> - However, neighbor nodes with time in $\mathbf{T}_ {t}^{\text{single}}$ do not have the “symmetric pairs”, unlike $2017$ having a “symmetric pair” $2019$. Therefore, neighbor nodes with time in $\mathbf{T}_ {t}^{\text{single}}$ are giving “half” the contributions to the target node when message passing, compared to double nodes. In other words, double nodes contribute twice more than single nodes when message passing. This is the explanation of the phrase “the target node receives twice the weight from $\tilde t\in \mathbf{T}_ {t}^{\text{double}}$against $\tilde t\in \mathbf{T}_ {t}^{\text{single}}$ ”.
> - In conclusion, if we artificially multiply 2 into the message passing weight of single nodes, every node will contribute equally when message passing.
> - Summary: We do not multiply 2 to the weight of nodes with time $2017$ or $2019$. We multilpy 2  to the weight of nodes with time $2018, 2016, 2015, 2014, …$.
>
> Question 2.
>
> - We agree to the point that $V$ can be large in some cases. This concern naturally conducted to our development of 2nd order moment alignment methods (PNY and JJ norm). By 2nd moment alignments, we are controlling the factor $V$ from increasing substantially.
> - In other words, although the theorems in Section 4.3 are written as a theoretical analysis for PMP, they also indicate that the variance of a distribution needs to controlled in order to guarantee the upper bound to be small.

---

> > ### Comment · Reviewer_u8i5 · 2024-11-26
> > **Comments**
> >
> > It is unclear how the 2nd order moment alignment controls the factor $V$.

---

> > > ### Author Response · Authors · 2024-11-27
> > >
> > > The 2nd order alignment method makes the distribution of test set and train set with same labels to be equal (invariant of time).
> > > Hence, $V$ is smaller in the 2nd order alignment method, compared to 1st order alignment method, which we commented as "controlling the factor $V$ from increasing substantially."
> > >
> > > More specifically, let us denote $Y$ to be the number of labels.
> > > For fixed layer $k$,
> > > - In 1st order alignment methods, $V$ will be the upper bound of the variances of total $Y$ * $t_{max}$ distributions (all the test and train distributions).
> > > - However in 2nd order alignment methods, $V$ will be the upper bound of the variances of total $Y$ distributions, since we only need to regard the variances of the test set. Therefore, the upper bound will be smaller in the 2nd order alignment methods.
> > >
> > > Additional remarks:
> > > - Comparing $C$ and $V$ directly in Thm 4.3 is not meaningful since we do not assume the shape of distributions except that the random variable is bounded. Instead, the focus of this theorem is that the order of $C$ is decreased from 1 to 1/3.
> > > - If one still wants to compare $C$ and $V$ directly, then Thm 4.4 is the partial result since Thm 4.4 makes an assumption about the shape of a distribution.

---

> > > ### Author Response · Authors · 2024-12-02
> > >
> > > Dear Reviewer u8i5,
> > >
> > > Up to our knowledge, from now on, this is the last 24 hours which reviewers may post a message to the authors.
> > > We would be grateful if you engage with our final comments.
> > >
> > > Best regards,
> > > Authors

---

### Official Review · Reviewer_6xgV · 2024-11-04

**Soundness:** 2
**Presentation:** 1
**Contribution:** 2
**Rating:** 3
**Confidence:** 5

**Summary:**

This paper studies the impact of chronological split in temporal graph learning. The authors propose a family of message passing methods named IMPaCT that accounts for the distribution shift between training and test due to the temporal effect.

**Strengths:**

S1. Chronological split affects the temporal graph learning's performance, and this paper aims to study this important problem.

S2. Experimental results show improvements in test accuracy.

**Weaknesses:**

W1. There are many related works about out-of-distribution generalization and invariant representation learning on graphs, as evidenced by several surveys and GitHub repo. It seems a bit contradicting to the authors' claim that there are not many related works. It would be great if the authors could further explain why these lines of works are not discussed.

W2. The observation about the decreasing proportion of neighbors as $|\tilde{t} - t|$ increases are mostly driven by the example in citation networks. I think this makes sense in citation networks, but it is unclear whether this trend also holds in other networks, say user-item interaction graphs in recommender systems or online social networks, where users can join or leave the system/network at any time.

W3. It would be great if the authors could discuss what types of shift (or the discrepancy between which distributions) this paper considers in a more formal way.

W4. I am a bit confused where the authors discussed the claim: "As discussed, the target node receives twice the weight from $\tilde{t}\in\mathbf{T}^{\text{double}}\_{t}$ against $\tilde{t} \in \mathbf{T}^{\text{single}}_{t}$." Could you please provide a pointer?

W5. It is questionable whether aligning first and second moment implies invariance, as skewness, tailedness, modes could all affect the shape of a distribution.

W6. Overall this paper is not self-contained. I have a hard time understanding many math notations in theorems given the lack of explanation in the main body (e.g., almost all theorems starting from theorem 4.3), and I need to check back and forth between main body and appendix to understand the paper. I believe a reorganization of the contents are needed for better clarity.

W7. The evaluation protocol is unclear to me. For example, PNY is proposed but never evaluated, so I am not sure what the purpose is to have PNY here. MMP, PMP, JJNorm is only evaluated on LDHGNN, but not on RevGAT and GAMLP, while GenPMP is not evaluated on LDHGNN.

W8. What is the rationale of evaluating on LDHGNN, RevGAT, and GAMLP rather than classic temporal graph learning models? Also, the authors do not explain what SimTeG, TAPE, and GLEM are. Overall, the choice of baseline methods are very vague.

**Questions:**

Please see weaknesses.

---

> ### Author Response · Authors · 2024-11-24
>
> We highly appreciate your constructive comments and evaluation. In the following, we address each of the reviewer's concerns in detail.
>
> Weakness 1.
>
> - In many cases, GNNs adopt different approaches depending on the graph's scale. For large-scale graphs, message passing depth leads to exponential increases in memory and time complexity, requiring scalable GNN models rather than general GNN approaches.
> - Our study was specifically motivated by the lack of domain adaptation research applicable to large-scale graph data. Datasets like ogbn-mag and ogbn-papers100M, which use chronological splits, clearly exhibit domain adaptation challenges. However, no experiments on their leaderboards utilize domain adaptation methods, making direct numerical comparisons with existing works impractical.
>
> Weakness 2.
>
> - We fully acknowledge the importance of validating our assumptions on temporal graphs beyond academic citation networks. Our study specifically focused on “chronologically split large-scale datasets,” and academic citation graphs were the most suitable datasets meeting these criteria. Consequently, the validation of our assumptions was naturally confined to this domain.
> - That said, the property of a “decreasing proportion of neighbors as $|\tilde t-t|$ increases” is not a necessary condition for our proposed method. The core assumptions of our work, such as symmetricity (nodes at time $t$ have similar probabilities of connecting with nodes at $t-\tau$ in the past and $t+\tau$ in the future) and separability (the connection probability distribution can be modeled as the product of the effects of time difference and labels), appear broadly applicable across various domains. Nonetheless, these assumptions should be empirically validated for each specific dataset before applying the proposed IMPaCT method.
>
> Weakness 3.
>
> - Our methods are broadly applicable to various discrepancies in neighboring connection distributions, provided that the underlying assumptions of our approach are satisfied. However, it is important to note that our study does not address domain shifts in the attributes of the nodes themselves. Instead, we focus on shifts in the connection distribution between nodes, $\mathcal{P}_{yt} (\tilde y, \tilde t)$, where even for the same label pair $(y,\tilde y)$, the distribution may vary depending on the temporal attributes $t$ and $\tilde t$ of the two nodes.
>
> Weakness 4.
>
> - Here is the explanation using Figure 3.
> - In this figure, $t_{max} = 2019$ and the target node’s time $t = 2018$.
> - By definition,  $\mathbf{T}_ {t}^{\text{double}} = \{ 2017, 2019 \}$ and $\mathbf{T}_{t}^{\text{single}} = \{ 2018, 2016, 2015, 2014, ... \}$
> - Neighbor nodes with time in $\mathbf{T}_{t}^{\text{double}}$, i.e., time $2017$ or time $2019$ have the same $\Delta = 1$ since the target node’s time is $2018$. Hence, the neighbor nodes with time $2017$ or $2019$ are giving the same amount of contributions when message passing to the target node (by Assumption 3).
> - However, neighbor nodes with time in $\mathbf{T}_ {t}^{\text{single}}$ do not have the “symmetric pairs”, unlike $2017$ having a “symmetric pair” $2019$. Therefore, neighbor nodes with time in $\mathbf{T}_ {t}^{\text{single}}$ are giving “half” the contributions to the target node when message passing, compared to double nodes. In other words, double nodes contribute twice more than single nodes when message passing. This is the explanation of the phrase “the target node receives twice the weight from $\tilde t\in \mathbf{T}_ {t}^{\text{double}}$against $\tilde t\in \mathbf{T}_{t}^{\text{single}}$ ”.
> - In conclusion, if we artificially multiply 2 into the message passing weight of single nodes, every node will contribute equally when message passing.
> - Summary: We do not multiply 2 to the weight of nodes with time $2017$ or $2019$. We multilpy 2  to the weight of nodes with time $2018, 2016, 2015, 2014, …$.
>
> Weakness 5.
>
> - Yes, we have also considered adjusting the third and fourth orders of a distribution, namely skewness and tailedness, respectively. However, even adjusting the 2nd orders caused a substantial increasement in time complexity, while the accuracy gain was subtle (although statistically significant).
> - We will positively consider this feedback and elaborate it into our further studies, but at this point we concluded that adjusting until 2nd order of distributions would prevent the paper to overflow.
>
> Weakness 6.
>
> - We will work on improving clarity, reorganizing the content, and providing better explanations for the mathematical notations and theorems on our final version.

---

> > ### Author Response · Authors · 2024-11-24
> >
> > Weakness 7.
> >
> > - The purpose and evaluation protocol of the experiments differ between datasets such as ogbn-mag/synthetic graphs and ogbn-papers100M/ogbn-arxiv.
> > - For ogbn-mag and synthetic graphs, the experiments aimed to demonstrate how our proposed methods, such as PMP and JJNorm, address domain adaptation challenges when all of our assumptions are satisfied. These experiments showcased the effectiveness of our methods, achieving state-of-the-art results.
> > - In contrast, ogbn-papers100M and ogbn-arxiv present a different challenge due to the extreme discrepancy in the distribution of node counts over time. In these datasets, the assumption that the connection distribution $\mathcal{P}_{yt} (\tilde y, \tilde t)$ depends on the absolute time difference between nodes (Assumption 3) does not hold, as the varying node counts over time violate this premise.
> > - Here, GenPMP plays a critical role by correcting temporal node count discrepancies during the message-passing process. The experiments using GenPMP on these datasets were designed to test whether, despite the violation of Assumption 3, first-order alignment remains effective if we numerically adjust for the mismatch in node counts across time attributes. However, when the temporal distribution of nodes is highly imbalanced, using GenPMP to correct these discrepancies leads to unbounded variance in the aggregated messages. This, in turn, makes it theoretically infeasible to apply second-order corrections, such as JJNorm.
> > - In summary, unlike the experiments on ogbn-mag, the evaluations on ogbn-papers100M and ogbn-arxiv using GenPMP served as a limited test to investigate whether correcting numerical imbalances in time attributes allows Assumptions 1 and 2, along with the separability of connection distributions, to hold, and whether first-order alignment remains effective under such conditions.
> >
> > Weakness 8.
> >
> > - In many cases, GNNs adopt different approaches depending on the scale of the graph. For large-scale graphs, scalable GNNs are typically employed, often relying on decoupled GNNs that perform message passing during preprocessing rather than training. As these decoupled GNNs do not incorporate learnable weights in their message-passing process, they commonly adopt averaging-based message passing.
> > - In contrast, for smaller datasets like ogbn-arxiv, most leaderboard baselines are not scalable GNNs. Our focus in these cases was not to achieve state-of-the-art performance but to validate the temporal graph assumptions and demonstrate the effectiveness of the IMPaCT method using a backbone model with average message passing.
> > - Lastly, for ogbn-papers100M, the SOTA model leverages LLMs for embedding generation rather than proposing a new graph learning algorithm. Due to computational constraints, we could not include this approach in our experiments.

---

> > > ### Comment · Reviewer_6xgV · 2024-11-24
> > >
> > > I appreciate authors' efforts. Unfortunately, after reading the response, several concerns are not addressed:
> > > * W1: I respectfully don't agree that no works on OGB means that there is no prior work about domain adaptation in graph learning. A simple search can output several related works, e.g., [1 - 4]. It is true that most prior works are not evaluated on billion-scale graphs, but they still worth discussing.
> > >
> > > * W2: The property "decreasing proportion of neighbors as $|\tilde{t} - t|$ increases" is how the authors describe $g$ function "is the function representing the proportion of neighboring nodes as a function that decays as $|\tilde{t} - t|$ increases", so it is unclear why the authors say it is not necessary in the response. $f$ function seems to be a coefficient to adjust the relative portion, which I am not sure whether it is the reason why the authors make such claim. Besides, I don't think there is any place clearly mentioning the symmetricity as a core assumption though it is implicitly embedded, making me even more concerned about the clarity of this paper.
> > >
> > > * W3 and W4: Thanks for the explanation. This should be incorporated to the modified version, which the authors haven't provided yet.
> > >
> > > * W5: Could the authors explain what it means by "prevent a paper to overflow"? Besides, if the authors believe this creates significant computational cost, just 1 or 2 sentences to justify would help.
> > >
> > > * W6 & W7: Thanks. It would be great if the authors could upload the modified version that show these changes.
> > >
> > > * W8: I think the authors misunderstood my concerns. My concerns are: (1) why don't the authors compare with classic **temporal** graph learning methods? (2) There is no descriptions of SimTeG, TAPE, and GLEM, making future readers hard to know what these methods are.
> > >
> > > Overall, I will maintain my score, as I am still deeply concerned about the paper's clarity.
> > >
> > > **References**
> > >
> > > [1] Unsupervised Domain Adaptive Graph Convolutional Networks
> > >
> > > [2] Out-Of-Distribution Generalization on Graphs: A Survey
> > >
> > > [3] GNN Domain Adaptation using Optimal Transport
> > >
> > > [4] Multi-source Unsupervised Domain Adaptation on Graphs with Transferability Modeling

---

> > > > ### Author Response · Authors · 2024-11-27
> > > >
> > > > Thank you for the comments. The constructive feedbacks certainly helped us improve and refine our paper.
> > > >
> > > > Regarding W1, we reviewed the attached articles and acknowledge their relevance to the subject of domain adaptation in graph learning, as noted by the reviewer. However, these studies do not include experimental evaluations on large-scale graphs (also as noted by reviewer). Section 2.3 specifically addresses the applicability of prior studies to large-scale graphs. There are still no studies which propose invariant learning in large scale graphs.
> > > >
> > > > Regarding W2, the sentence stating "$g$ is the function representing the proportion of neighboring nodes as a function that decays as $|\tilde{t} - t|$ increases" was an explanation of the graph citation network dataset we were planning to conduct experiments. Generally, the function $g$ (representing the proportion of neighbors at specific time stamp) does not need to have a decreasing nor increasing property with respect to $|\tilde{t} - t|$, as the theorems do not rely on the decaying property of $g$ at all, and hence can be identically derived without the decaying property.
> > > > We acknowledge that confusions may have been triggered as we stated the sentence before the experimental results. We have incorporated the reviewer's concern and uploaded a revised version.
> > > >
> > > > Regarding W3 ~ W7, we have uploaded a revised version of our paper. Major changes are refined notations and additional motivational explanations incorporated into Section 4.
> > > >
> > > > Regarding W8, we acknowledge the concern and apologize if it substantially reduced the readability. We were unable to include it due to space constraints.

---

> > > > ### Author Response · Authors · 2024-12-02
> > > >
> > > > Dear Reviewer 6xgV,
> > > >
> > > > Up to our knowledge, from now on, this is the last 24 hours which reviewers may post a message to the authors.
> > > > We would be grateful if you engage with our final comments.
> > > >
> > > > Best regards,
> > > > Authors

---

### Official Review · Reviewer_L1bX · 2024-11-06

**Soundness:** 2
**Presentation:** 2
**Contribution:** 3
**Rating:** 5
**Confidence:** 4

**Summary:**

This paper explores the problem of domain adaptation of temporal maps and proposes a method to realize the invariant property based on realistic assumptions. The method takes into account the characteristics of temporal splitting and also introduces the Temporal Stochastic Block Model (TSBM) to replicate the temporal graph under different conditions.

**Strengths:**

1. This paper addresses the intriguing challenge of proposing the invariant messaging function IMPaCT to tackle the domain adaptation issues in graph data that arise from temporal splitting.
2. The authors establish a framework of assumptions grounded in observable properties and introduce an invariant theory for first-order and second-order moments, which offers a robust theoretical foundation for future research.
3. The IMPaCT method demonstrates substantial improvements in classification accuracy across multiple datasets when compared to existing state-of-the-art methods, highlighting its advantages in processing graph data effectively.

**Weaknesses:**

1.The paper explores the problem of out-of-distribution generalization in temporal graphs; however, the INTRODUCTION section does not adequately address the unique challenges associated with this issue. The authors should elaborate on how the difficulties of out-of-distribution generalization in temporal graphs differ from those encountered in static graphs and Euclidean data (e.g., images). A clearer articulation of these challenges would enhance the understanding of the problem's significance and context.

2.There is a lack of comparison between relevant aspects of the proposed work and existing methods. Could the authors clarify how their approach measures up against current state-of-the-art techniques in the domain?

3.How is the effectiveness of the PMP and MMP in adjusting the invariance of the first moments assessed?The MMP only collects information from past nodes, which may lead to insufficient information. How do the authors explain the potential impact of this limitation on model performance?

4.On what theoretical basis is the weight adjustment of generating nodes in GenPMP based? Can the authors explain in detail how the rationality of \(P_{t_{\text{max}}}(\Delta)/P_{\tilde{t}}(\Delta)\) was determined and its impact on the results?

5.Please explain why different baselines are used on different datasets?

6.The authors assert that this paper tackles the challenge of domain adaptation in graph data resulting from temporal splitting. However, as far as I am aware, the ogb datasets are predominantly utilized for node classification in static graphs. How effectively does the proposed method perform on dynamic graph datasets, such as Tmall and MathOverflow?

**Questions:**

Please see Weaknesses.

---

> ### Author Response · Authors · 2024-11-24
>
> We highly appreciate your constructive comments and evaluation. In the following, we address each of the reviewer's concerns in detail.
>
> Weakness 1 & 2.
> - Previous research is not applicable to large-scale graphs. Our study was motivated by the lack of domain adaptation research applicable to large-scale graph data. Notably, large graph datasets such as ogbn-mag and ogbn-papers100M use chronological splits, making domain adaptation challenges evident. However, no experiments on the leaderboard employ domain adaptation approaches, making direct numerical comparisons with existing methods infeasible.\\
> - Furthermore, traditional domain adaptation studies often assume that the data generation process inherently creates causal invariances (e.g., for samples with the same label), which can be learned through techniques like regularization or augmentation. However, this assumption of causal invariance is unverified, and in the case of chronological splits, the data generation process may evolve over time, potentially making such assumptions unrealistic. Thus, our approach identifies assumptions typically satisfied by real-world temporal graph data and proposes a message-passing technique that explicitly imposes these invariances.
>
>
> Weakness 3.
> - Our purpose was not to specifically propose MMP but to present and compare various examples of 1st-order alignment methods.
> - While MMP only collects information from past nodes, this limitation was intentionally included to highlight the diverse approaches and their potential trade-offs, rather than to emphasize MMP as a definitive solution.
>
>
> Weakness 4.
> - There is a typo in our original version, and we will fix the typo.
> - Part which will be fixed:  $\sum_{\tilde{t} \le t}$ into $\sum_{\tilde{t} \in \mathbf{T}_{t}^{\Delta}}$
> - Consider the two distributions of $P_{t_{max}}$ and $P_{\tilde{t}}$. If we get a node $w$ from $\mathcal{N}_ {v}(\tilde{y},\tilde{t})$, we need to multiply the proportion $\frac{P_{t_{\max}}(\Delta)}{P_{\tilde{t}}(\Delta)}$ in order to adjust the distribution of random variable $X_w$. In other words, we are adjusting the distribution $P_{\tilde{t}}$ into the distribution $P_{t_{max}}$.
>
>
> Weakness 5.
> - In many cases, GNNs employ different approaches depending on the scale of the graph. For large-scale graphs, scalable GNNs are used, typically relying on decoupled GNNs that perform message passing only during preprocessing. Since decoupled GNNs do not involve learnable weights in their message passing process, they often adopt averaging-based message passing.
> - In contrast, for smaller datasets like ogbn-arxiv, most leaderboard baselines are not scalable GNNs. Our focus in these cases was not to achieve state-of-the-art performance but to validate the temporal graph assumptions and demonstrate the effectiveness of the IMPaCT method using a backbone model with average message passing.
> - Lastly, for ogbn-papers100M, the SOTA model leverages LLMs for embedding generation rather than proposing a new graph learning algorithm. Due to computational constraints, we could not include this approach in our experiments.
>
>
> Weakness 6.
> - Unfortunately, our proposed method is not applicable to dynamically evolving graphs. The our algorithm involves normalizing the distribution of neighboring node counts over time, using both past and future data. To apply 1st-order alignment, data from both past and future must be available during the message-passing phase.

---

> > ### Comment · Reviewer_L1bX · 2024-12-03
> >
> > Thank you for the response. I'd like keep my score unchanged.

---

> ### Author Response · Authors · 2024-12-02
>
> Dear Reviewer L1bX,
>
> Up to our knowledge, from now on, this is the last 24 hours which reviewers may post a message to the authors.
> We would be grateful if you engage with our comments, as we believe that the discussions are not disclosed.
>
> Best regards,
> Authors

---

### Author Response · Authors · 2024-11-27

Dear reviewers, we have uploaded a revised version of our paper. Please check it out, and we respectfully ask considering re-evaluation of initial scores if the concerns are addressed.

---

### Meta-Review · Area_Chair_GHZ9 · 2024-12-20

**Metareview:**

This paper studies what the reviewers agree is an important and interesting issue that arises when applying machine learning to temporal graphs: if the training/test split is not random but rather chronological (e.g., training nodes arrived in the graph before test nodes), this can lead to distribution shift between the training and test set, necessitating the application of some sort of domain adaptation method. The paper proposes such a method, based on message passing, which attempts to ensure that the distribution of node features after message passing has certain time-invariant properties that the authors posit are reasonable for many real-world temporal graph datasets.

Reviewers had several concerns that would need to be addressed before the paper is ready for publication. Primary among them are:
1. The paper does not extensively compare again other domain adaptation methods in the graph ML literature. In the rebuttal the authors comment that this is because such methods do not scale to large graphs. But at least more discussion of existing approaches and justification for why they would not scale/apply would be needed.
2. The presentation of the paper and the mathematical rigor need significant improvement. The authors did provide a new version of the manuscript which is a step in the right direction.
3. Empirical comparisons should be better motivated/more consistent across datasets and should include more baselines when possible. Verification of invariance assumptions should go beyond citation networks.

Overall, the ideas in this paper have potential but need more work before being ready for publication.

As a minor point: the AC would like to point out that a 'Temporal Stochastic Block Model (TSBM)' has already been introduced in prior work (https://arxiv.org/abs/2210.00032). The authors should consider citing this work and potentially using a different name to distinguish their related but different model.

**Additional Comments On Reviewer Discussion:**

The authors provided comments to address many of the reviewers' questions/critiques, often helping to clarify some issues. Overall, however, the main concerns discussed in the metareview remained after the rebuttal.

---

### Decision · Program_Chairs · 2025-01-22

Reject